# EARLINET evaluation of the CATS L2 aerosol backscatter coefficient product

Emmanouil Proestakis[1], Vassilis Amiridis[1], Eleni Marinou[2], Ioannis Binietoglou[3], Albert Ansmann[4], Ulla Wandinger[4], Julian Hofer[4], John Yorks[5], Edward Nowottnick[6], Abduvosit Makhmudov[7], Alexandros Papayannis[8], Aleksander Pietruczuk[9], Anna Gialitaki[1], Arnoud Apituley[10], Artur Szkop[9], Constantino Muñoz Porcar[11], Daniele Bortoli[12], Davide Dionisi[13], Dietrich Althausen[4], Dimitra Mamali[14], Dimitris Balis[15], Doina Nicolae[3], Eleni Tetoni[2], Gian Luigi Liberti[13], Holger Baars[4], Ina Mattis[16], Iwona S. Stachlewska[17], Kalliopi Artemis Voudouri[15], Lucia Mona[18], Maria Mylonaki[8], Maria Rita Perrone[19,20], Maria João Costa[12], Michael Sicard[11,21], Nikolaos Papagiannopoulos[11,18], Nikolaos Siomos[15], Pasquale Burlizzi[19,20], Rebecca Pauly[22], Ronny Engelmann[4], Sabur F. Abdullaev[7], Gelsomina Pappalardo[18]

[1]IAASARS, National Observatory of Athens, Athens, 15236, Greece
[2]Institut für Physik der Atmosphäre, Deutsches Zentrum für Luft und Raumfahrt (DLR), Oberpfaffenhofen, Germany
[3]National Institute of R&D for Optoelectronics, Magurele, Romania
[4]Leibniz Institute for Tropospheric Research, Leipzig, Germany
[5]NASA Goddard Space Flight Center, Greenbelt, Maryland, 20771, United States
[6]GESTAR, Universities Space Research Association, Columbia, Maryland, United States
[7]Physical Technical Institute of the Academy of Sciences of Tajikistan, Dushanbe, Tajikistan
[8]Laser Remote Sensing Unit (LRSU), National Technical University of Athens Physics Department, Zografou, Greece
[9]Institute of Geophysics, Polish Academy of Sciences, 01-452 Warsaw, Poland
[10]Royal Netherlands Meteorological Institute (KNMI), De Bilt, the Netherlands
[11]CommSensLab, Dept. of Signal Theory and Communications, Universitat Politècnica de Catalunya, Barcelona, Spain
[12]Departamento de Física, Instituto de Ciências da Terra, Escola de Ciências e Tecnologia, Universidade de Évora, Évora, Portugal
[13]Consiglio Nazionale delle Ricerche, Istituto Scienze Marine (CNR-ISMAR), Rome-Tor Vergata, Italy
[14]Department of Geoscience and Remote Sensing, TU Delft, Delft, The Netherlands
[15]Laboratory of atmospheric physics, Physics Department, Aristotle University of Thessaloniki, Greece
Observatory Hohenpeissenberg, German Weather Service, Hohenpeissenberg, Germany
[17]Institute of Geophysics, Faculty of Physics, University of Warsaw (IGFUW), 02-093 Warsaw, Poland
[18]Consiglio Nazionale delle Ricerche, Istituto di Metodologie per l'Analisi Ambientale (CNR-IMAA), C.da S. Loja, Tito Scalo (PZ), 85050, Italy
[19]Dipartimento di Matematica e Fisica, Università del Salento, Lecce, Italy
[20]CNISM-Consorzio Nazionale Interuniversitario per le Scienze Fisiche della Materia, Lecce, Italy
[21]Ciències i Tecnologies de l'Espai - Centre de Recerca de l'Aeronàutica i de l'Espai/Institut d'Estudis Espacials de Catalunya (CTE-CRAE/IEEC),
[22]Science Systems and Applications Inc., Lanham, 20706, United States

*Correspondence to:* Emmanouil Proestakis (proestakis@noa.gr)

**Abstract.** We present the evaluation activity of the European Aerosol Research Lidar Network (EARLINET) for the quantitative assessment of the Level 2 aerosol backscatter coefficient product derived by the Cloud-Aerosol Transport System (CATS) onboard the International Space Station (ISS). The study employs correlative CATS and EARLINET backscatter measurements within 50 km distance between the ground station and the ISS overpass and as close in time as possible, typically with starting time or stop time of the EARLINET performed measurements time window within 90 minutes of the ISS overpass, from February 2015 to September 2016. The results demonstrate the good agreement of CATS Level 2 backscatter coefficient and EARLINET. Three ISS overpasses close to the EARLINET stations of Leipzig-Germany, Évora-Portugal and Dushanbe-Tajikistan are analysed here to demonstrate the

performance of CATS lidar system under different conditions. The results show that under cloud-free, relative homogeneous aerosol conditions, CATS is in good agreement with EARLINET, independently of daytime/nighttime conditions. CATS low negative biases, partially attributed to the deficiency of lidar systems to detect tenuous aerosol layers of backscatter signal below the minimum detection thresholds, may lead to systematic deviations and slight underestimations of the total Aerosol Optical Depth (AOD) in climate studies. In addition, CATS misclassification of aerosol layers as clouds, and vice versa, in cases of coexistent and/or adjacent aerosol and cloud features, may lead to non-representative, unrealistic and cloud contaminated aerosol profiles. Regarding solar illumination conditions, low negative biases in CATS backscatter coefficient profiles, of the order of 6.1%, indicate the good nighttime performance of CATS. During daytime, reduced signal-to-noise ratio by solar background illumination prevents retrievals of weakly scattering atmospheric layers that would otherwise be detectable during nighttime, leading to higher negative biases, of the order of 22.3%, in CATS daytime performance.

## 1 Introduction

The Cloud-Aerosol Transport System (CATS) is a satellite-based elastic backscatter lidar developed to provide near-real time, vertically resolved information on the vertical distribution of aerosols and clouds in the Earth's atmosphere (McGill et al., 2015). Developed at the NASA's Goddard Space Flight Center, CATS is based on the Cloud Physics Lidar (CPL; McGill et al., 2002) and the Airborne Cloud-Aerosol Transport System (ACATS; Yorks et al., 2014), designed to operate onboard the high-altitude NASA ER-2 aircraft. CATS operated as a scientific payload onboard the Japanese Experiment Module - Exposed Facility (JEM-EF), utilizing the International Space Station (ISS) as a space science platform (Yorks et al., 2016). Starting from 10 February 2015, CATS provided aerosol and cloud profile observations along the ISS flight track for more than 33 months until 30 October 2017 when the system suffered an unrecoverable fault.

CATS was developed to meet three main science goals. The primary objective was to measure and characterize aerosols and clouds on a global scale. The space-borne lidar orbited the Earth at an altitude of approximately 405 km and 51-degree inclination. The use of the ISS as an observation platform facilitated for the first time global lidar-based climatic studies of aerosols and clouds at various local times (Noel et al., 2018, Lee et al., 2018). In addition, near-real-time data acquisition of the CATS observations was developed towards the improvement of aerosol forecast models (Hughes et al, 2016). A secondary objective was related to the need of long-term and continuous satellite-based lidar observations to be available for climatic studies. The first spaceborne lidar mission, the Lidar In-space Technology Experiment (LITE; McCormick et al., 1993) in 1994, was succeeded by the joint NASA and Centre National d'Études Spatiales (CNES) Cloud-Aerosol Lidar and Infrared Pathfinder Satellite Observation (CALIPSO) mission in June, 2006 (Winker et al., 2007). Since 2009 the Cloud-Aerosol Lidar with Orthogonal Polarization (CALIOP) instrument (Winker et al., 2009) onboard CALIPSO operates on the secondary backup laser. The launch of the post-CALIPSO missions, the joint European Space Agency (ESA) and JAXA satellite Earth Cloud Aerosol and Radiation Explorer (EarthCARE; Illingworth et al., 2015) and the NASA's Aerosols, Clouds, and Ecosystems

(ACE) are planned for 2021 and post-2020 respectively. The CATS project was partially intended to fill a potential gap on global lidar observations of vertical aerosols and clouds profiling. The third scientific objective of CATS was to serve as a low-cost technological demonstration for future satellite lidar missions (McGill et al., 2015). Its science goal to explore different technologies was fulfilled through the use of photon-counting detectors and of two low energy (1-2 mJ) and high repetition rate (4-5 kHz) Nd:YVO4 lasers (Multi-Beam and HSRL - UV demonstrations), aiming to provide simultaneous multiwavelength observations (355, 532 and 1064 nm). Additional gains of the CATS project were related to the exploitation and risk reduction of newly applied laser technologies, to pave the way for future spaceborne lidar missions (high repetition rate, injection seeding, wavelength tripling at 355 nm).

CATS performance has been validated against ground-based AErosol RObotic NETwork (AERONET; Holben et al., 1998) measurements and evaluated against satellite-based Atmospheric Optical Depth (AOD) retrievals of Aqua and Terra Moderate Imaging Spectroradiometer (MODIS; Levy et al., 2013) and active CPL (McGill et al., 2002) and CALIPSO CALIOP (Winker et al., 2009) profiles of extinction coefficient and AOD at 1064 nm. Lee et al. (2018) compared daytime quality-assured CATS V2-01 vertically integrated extinction coefficient profiles (1064 nm) and AERONET AOD (1020 nm) values, spatially (within 0.4° Longitude and Latitude) and temporally (±30 minutes) collocated, and found a reasonable agreement with a correlation of 0.64. A comparative analysis of CATS and MODIS C6.1 Dark Target (DT) AOD retrievals, through spectral interpolation between 0.87 and 1.24 μm channels, reported correlation of 0.75 and slope of 0.79, over ocean. In addition, Lee et al., (2019) evaluated AOD and extinction coefficient profiles from CATS through intercomparison with CALIOP. Regarding AOD, analysis of 2681 CATS and CALIOP collocated observation cases (within 0.4° Longitude/Latitude and ±30 minutes ISS and CALIPSO overpass difference), showed correlation of 0.62 and 0.52 over land and ocean respectively during daytime (1342 cases), and 0.84 and 0.81 over land and ocean respectively during nighttime (1339 cases). Comparison of CATS and CALIOP collocated extinction coefficient profiles based on the closest Euclidian distance on the earth's surface shows also good shape agreement, despite an apparent CALIOP underestimation in the lowest 2 km height. CATS and CALIOP observations were used by Rajapakshe et al. (2017) to study the seasonally transported aerosol layers over the SE Atlantic Ocean. The performed comparative analysis reported on similar geographical patterns regarding Above Cloud Aerosols (ACA), Cloud Fraction (CF) and ACA occurrence frequency (ACA_F) between CATS and CALIOP retrievals. However, the authors reported also on differences between CATS and CALIOP vertical aerosol distributions, with ACA bottom height identified by CATS lower than the respective of CALIOP. Noel et al. (2018), implemented measurements from CATS to investigate the diurnal cycle and variations of clouds over land and ocean. The authors showed that both CATS and CALIOP profiles and CF agree well on both the vertical patterns and values at 01:30 and 13:30 LT, over both land and ocean, with minor differences of the order of 2-7% throughout the entire cloud profiles. CATS depolarization measurements, which are critical in the processing algorithms of aerosol subtype classification, were investigated in the case of desert dust, smoke from biomass burning and cirrus clouds (Yorks et al., 2016), and were found consistent and in good agreement with depolarization measurements from previous studies and historical datasets implementing CPL (Yorks et al., 2011) and CALIOP (Liu et al., 2015).

Overall, CATS retrievals have been evaluated and found in reasonable agreement with ground-based AERONET, airborne CPL and satellite-based MODIS and CALIOP measurements. However, for the quality assessment of CATS backscatter coefficient profiles, a large-scale and dense network of ground-based lidar systems is needed, in order to facilitate high-quality collocated and concurrent measurements. This necessity is largely related to the ISS orbital characteristics, the CATS near-nadir viewing (0.5° off nadir), the lidar narrow footprint (14.38 m diameter), and the limited number of ISS overpasses. The European Aerosol Research Lidar Network (EARLINET) consists of a unique infrastructure for assessing the validation needs for spaceborne lidar missions. EARLINET operates in the framework of Aerosols, Clouds and Trace Gases Research Infrastructure (ACTRIS) as a pan-European effort to develop a coordinated lidar research infrastructure (Pappalardo et al., 2014) of advanced Raman lidar systems and is characterized by extensive geographical coverage.

In this paper, we utilize EARLINET for the evaluation of CATS Level 2 aerosol backscatter coefficient product at 1064 nm. The paper is structured as follows: in section 2 we introduce aspects of CATS and EARLINET relevant to the study and additionally the comparison methodology is presented and discussed. Specific study cases are evaluated and discussed in section 3. Section 4 presents the generic intercomparison results between CATS and EARLINET, while the concluding remarks on the CATS-EARLINET backscatter coefficient evaluation are summarized in section 5.

## 2 Data and methodology

### 2.1 CATS

The CATS elastic backscatter lidar was designed to provide near-real-time measurements of the vertical profiles of aerosol and cloud optical properties at three wavelengths (355, 532 and 1064 nm). As a payload of the JEM-EF on the ISS, CATS was designed to operate two high repetition rate lasers in three different Modes and at four instantaneous fields of view (iFOV). Mode 1 was designed as multi-beam backscatter and depolarization configuration at 532 and 1064 nm, where a beam-splitter would produce two footprints of 14.38 m diameter on the Earth's surface, to the left side FOV (LSFOV) and the right side FOV (RSFOV) of the ISS orbit track, separated by approximately a distance of 7 km. Mode 2 was designed as a demonstration of HSRL, to provide backscatter profiles at 532 nm and backscatter and depolarization ratio profiles at 1064 nm (Forward FOV). Mode 3 was designed to operate and provide backscatter at 355, 532 and 1064 nm, and depolarization ratio at 532 and 1064 nm. CATS was a technology demonstration designed to operate on-orbit between six months and three years. Due to a failure in the CATS optics at the 355 nm wavelength, CATS did not operate in Mode 3, while the use of Mode 1 was limited between 10/02/2015 and 21/03/2015 due to a failure in the electronics of laser 1. Nevertheless, the successful long-term operation of Mode 2, between 02/2015 and 10/2017, allowed CATS to fulfil its science objectives.

CATS processing algorithms (Pauly et al., 2019) rely heavily on the processing algorithms developed in the framework of the CPL, ACATS and CALIPSO lidar systems (Palm et al., 2002; Yorks et al., 2011; Hlavka et al., 2012), while CATS products are provided in different levels of processing. CATS Level 1B data include vertical

profiles of total and perpendicular attenuated backscatter signals, range-corrected, calibrated and annotated with ancillary meteorological parameters based on previous work using CPL and CALIPSO (McGill et al., 2007; Powell et al., 2009; Vaughan et al., 2010). Level 2 products provide the vertical distribution of aerosol and cloud properties (depolarization ratio, backscatter and extinction coefficient profiles at 1064 nm – FFOV), with a horizontal and vertical resolution of 5 km and 60 m respectively. In addition, Level 2 data include geophysical parameters of the identified atmospheric layers (vertical feature mask - feature type, aerosol subtype), the required horizontal averaging and information on the feature type classification confidence (Yorks et al., 2019). In addition to CATS Level 2 Feature Type (namely: clear air, cloud, aerosol and totally attenuated), the algorithm provides the confidence level of the Feature Type classification, similar to the CALIOP Cloud-Aerosol-Discrimination (CAD) algorithm (Liu et al., 2004; Liu et al., 2009). CATS Feature Type Score is a multidimensional probability density function (PDF) developed based on multiyear CPL observations, that discriminates cloud and aerosol features, assigning an integer between -10 and 10 for each detected atmospheric layer.

In this study, we used CATS Level 2 v2.01 profiles (Palm et al., 2016). A comprehensive overview of the CATS instrument and CATS science goals is given by McGill et al. (2015) and Yorks et al. (2016), while detailed information about CATS datasets and an images browser can be found in the CATS Data Release Notes, Quality Statements and Theoretical Basis, available at https://cats.gsfc.nasa.gov/ (last access: 20 December 2018).

## 2.2 EARLINET

EARLINET (EARLINET; https://www.earlinet.org/index.php?id=earlinet_homepage, last access: 20 December 2018) was founded by the European Commission (Bösenberg et al., 2001) as a research project within the framework of the Fifth Framework Programme (FP5). Currently the network activity is integrated and constitutes a major component of the ACTRIS research infrastructure (ACTRIS; https://www.actris.eu/, last access: 20 December 2018). The main objective of EARLINET is to establish an extended, coordinated and continental wide network of sophisticated ground-based Raman lidar systems. The vertical distribution of aerosols in the atmosphere, as well as their temporal evolution, are provided by high-resolution EARLINET measurements over Europe. The long-term continuous operation of EARLINET infrastructure has fostered a quantitative, comprehensive, and statistically significant database of the distribution of aerosol on a continental scale (Bösenberg et al., 2003; Pappalardo et al., 2014).

Since the beginning of the initiative in 2000, EARLINET has significantly increased its observing and operational capacity. Currently, EARLINET is composed of twenty-nine operating lidar stations distributed over Europe (Fig. 1), including seven admitted or joining stations. EARLINET stations are classified between "active", "not permanent", "joining" and "not active". An EARLINET station is classified as active when on condition of performing regularly and simultaneously measurements with the other stations composing the lidar network, and accordingly, contributing with uploading the performed measurements to the EARLINET database (https://www.earlinet.org/, last access: 20 December 2018). Lidar observations in the framework of EARLINET are performed according to a common schedule - on preselected dates. The schedule involves three measurements per

week, one during daytime around local noon (Monday, 14:00 ± 1h) and two during nighttime (Monday/Thursday, sunset + 2/3h), to enable Raman extinction retrievals. In addition to the preselected dates of the operation schedule, dedicated measurements are performed to monitor special events such as major volcanic activity (Ansmann et al., 2010; Ansmann et al., 2011; Pappalardo et al., 2013; Perrone et al., 2012; Sicard et al., 2012; Wang et al., 2008),

long-range transport of Saharan dust (Ansmann et al., 2003, Solomos et al., 2017, 2018) and smoke particles (Ortiz-Amezcua et al., 2017, Janicka et al. 2017, Stachlewska et al. 2018). Some of the EARLINET systems perform meanwhile 24/7 continuous measurements as for example the PollyXT systems (Engelmann et al., 2016, Baars et al., 2016). The quality assurance and improvement of the performance of the EARLINET systems is tested through the intercomparison of both the infrastructure (Wandinger et al., 2015) and the optical products (Böckmann et al., 2004;

Pappalardo et al., 2004). In addition, the homogenization of the lidar data in a standardized output format is facilitated and an automatic algorithm is developed to further address the quality assurance of the lidar measurements (the Single Calculus Chain (SCC), D'Amico et al, 2015; D'Amico et al, 2016; Mattis et al., 2016). The SCC has been used in near-real time to shown the potential operationality of the network in a 72-hr continuous measurement exercise in 2012 (Sicard et al., 2015).

Due to its implicit characteristics, EARLINET is an optimum tool to support satellite-based lidar missions with extensive experience to satellite calibration and validation activities. EARLINET and Cloud-Aerosol Lidar and Infrared Pathfinder Satellite Observation (CALIPSO; Winker et al., 2009) correlative measurements are regularly performed in order to investigate the quality of the Cloud-Aerosol Lidar with Orthogonal Polarization (CALIOP) observations, to test the presence of possible biases, and to assess the aspects of spaceborne lidar measurements (e.g.

Pappalardo et al., 2010; Mamouri et al., 2009, Mona et al., 2009; Perrone et al., 2011; Wandinger et al., 2011; Amiridis et al., 2013; Grigas et al., 2015; Papagiannopoulos et al., 2016). Similarly, ESA validation programs of the -Atmospheric Laser Doppler Instrument (ALADIN) onboard Aeolus (Stoffelen et al., 2005; Ansmann et al., 2007) and the ESA-JAXA EarthCARE (Illingworth et al., 2015) are highly-dependent on ground-based EARLINET correlative measurements. In addition, EARLINET supports the homogenization of the different satellite missions.

CALIOP, is a two-wavelength polarization-sensitive lidar that operates at 532 and 1064 nm, while the ESA's ALADIN onboard Aeolus and the ESA-JAXA ATLID onboard EarthCARE operate at 355 nm and NASA's CATS lidar at 532 and 1064 nm in Mode 1 and 1064 nm in Mode 2 (Yorks et al., 2014). EARLINET supports the continuity of satellite lidar missions through the calculation of aerosol-dependent spectral conversion factors between different wavelengths, to homogenize different missions at different operating wavelengths in order to provide a long-term 3D

climatic record from space (Amiridis et al., 2015; Marinou et al., 2017; Proestakis et al., 2018).

## 2.3 EARLINET-CATS correlative measurements
### 2.3.1 Comparison methodology

To obtain a significant number of collocated and concurrent EARLINET-CATS cases, a large number of EARLINET stations contributed to the CATS evaluation activity. Figure 1 shows the geographical distribution of the active EARLINET stations during the study over Europe and Asia, including the daytime/nighttime overpasses of ISS

within the evaluation period, between 02/2015 and 09/2016, encompassing the first twenty months of CATS operation. The green circles denote the stations participating in the EARLINET-CATS inter-comparison activity (namely - in alphabetical order: Athens-NOA, Athens-NTUA, Barcelona, Belsk, Bucharest, Cabauw, Dushanbe, Évora, Hohenpeissenberg, Lecce, Leipzig, Potenza, Thessaloniki and Warsaw). All participating stations operate high performance multiwavelength lidar systems. Six of the contributing stations (Athens-NOA, Cabauw, Dushanbe, Évora, Leipzig and Warsaw) are part of the PollyNET subnetwork (http://polly.tropos.de/), operating 24/7 portable, remote-controlled multiwavelength-polarization-Raman lidar systems (PollyXT; Baars et al., 2016; Engelmann et al., 2016). Due to the geographical distribution of EARLINET stations, the evaluation activity accounts for a large variety of aerosol types (marine, urban, desert dust, smoke). Table 1 provides the locations of the EARLINET stations contributing to this analysis along with the surface elevation and the respective identification codes.

In order to quantitatively address the accuracy and representativeness of CATS retrievals, we follow the methodology introduced by EARLINET for CALIOP validation, which is based on correlative independent measurements (Pappalardo et al., 2010). For the validation of spaceborne lidar observations, of fundamental significance is the spatial and temporal variability of the atmospheric scene. The effect of distance between ground-based lidar measurements and space-based lidar measurements was investigated in the framework of the CALIPSO validation. In particular, EARLINET-based studies attribute an introduced discrepancy of the order of 5 % to the intercompared signal analysis, when the horizontal distance between the EARLINET stations and the spaceborne lidar footprint is below 100 km (Mamouri et al., 2009; Mona et al., 2009; Pappalardo et al., 2010; Papagiannopoulos et al., 2016). In the context of the applied validation criteria, we selected CATS measurements within 50 km horizontal distance between the EARLINET stations and the ISS subsatellite overpass position. In addition, the correlative measurements should be as close in time as possible. EARLINET contributed with performed measurements as close in time as possible, typically with starting time or stop time of the performed measurements window within 90 minutes of the ISS station overpass. The EARLINET-CATS cases considered to the assessment of the accuracy and representativeness of CATS backscatter coefficient profiles are provided in Table 2, including the name of the EARLINET station, the EARLINET measurements window, the ISS overpass time and ISS minimum distance between the corresponding EARLINET station and the lidar footprint of CATS and the Daytime/Nighttime information.

The number of available cases for the intercomparison is subject to a certain number of constraints. First and foremost, the orbital inclination of the ISS does not allow to overpass close to EARLINET stations northern of 52.2° latitude. Second, the ISS crossing-time and ground-track over an area is highly variable, enhancing the probability of the overpass time to fall outside the predefined common and fixed schedule of EARLINET measurements. In addition, to account for contamination effects of multiple-scattering and specular reflection in the intercomparison process, only cloud-free atmospheric scenes are used. Cases with detected cirrus either at the EARLINET Range-Corrected-Signal quicklooks or at the ISS-CATS backscatter coefficient profiles or the feature type profiles are not considered in the study. Initially, the presence of clouds is investigated through the implementation of CATS backscatter coefficient and depolarization time-height images and EARLINET range-corrected-signal. Cases for

which the retrieval of EARLINET temporally-averaged profile is not feasible due to the presence of clouds, and/or CATS cases that the presence of clouds propagated into the CATS spatial-averaged profile are discarded from the analysis. Regarding CATS, the "Sky_Condition" flag is used to screen cloudy (no aerosols) and hazy/cloudy (both clouds/aerosols) profiles from the analysis. The "Feature_Type_Score" parameter stored in the Level 2 data was additionally used to remove aerosol cases of medium/low confidence in the comparison process ("Feature_Type_Score" ≥ -1). Applying all match-up selection criteria resulted in a total of 47 correlative EARLINET-CATS cases suitable to quantitatively address the accuracy and representativeness of CATS Level 2 backscatter coefficient product at 1064 nm. CATS requirements applied in the methodology are summarized in Table 3.

## 2.3.2 Particle backscatter coefficient retrievals from ground based lidars at 1064 nm

In order to evaluate the CATS Level 2 aerosol backscatter product at 1064 nm we utilized backscatter coefficient profiles calculated either with the SCC algorithm or, in case of PollyXT lidar systems, with independently developed user assisted retrieval algorithms (Baars 2016). The EARLINET backscatter coefficient profiles used in this study are calculated with the SCC version 4 algorithm (for the stations that are not part of PollyNET) and with the methodology described in Haarig et al., 2017 (for the stations that are part of PollyNET). The SCC algorithm (D'Amico et al., 2015; D'Amico et al., 2016; Mattis et al., 2016) is developed in the concept of sustaining the homogeneity of aerosol products derived from different EARLINET lidar systems while satisfying the need for coordinated, quality assured measurements. It consists of five different modules, including one for handling the pre-processing of raw lidar data by applying all the necessary instrumental corrections to the signal and a module for providing the final aerosol optical products, namely the particle backscatter and extinction coefficient. In particular, SCC algorithm calculates the backscatter coefficient with the iterative method (Di Girolamo et al., 1995), using only the elastic lidar channels. To calculate the $b_{1064nm}$ with these methods, an assumption of the lidar ratio value is required (as a profile or a height independent value, representative of the corresponding atmospheric scene) and the selection/determination of a reference height ($R_0$), usually chosen at an altitude range with the minimum aerosol contribution. All methods applied within the SCC, have been tested against synthetic (Mattis et al., 2016) and real lidar data (D'Amico et al., 2015). The comparison showed that by using only the signal from the elastic channels, the mean relative deviation in the calculation of the aerosol backscatter coefficient at 1064 nm is less than 30 % (Althausen et al., 2009; Baars et al., 2012; Engelmann et al., 2016; Hänel et al., 2012), thus meeting the quality assurance requirements of EARLINET. None of the lidar systems participating in the present study, is equipped with a rotational-vibrational Raman channel excited by the 1064 nm as for example recently reported by Haarig et al (2017). In the case of PollyXT lidars, for the daytime backscatter coefficient calculations, the Fernald-Klett method (Klett, 1981; Fernald, 1984) is implemented assuming a height independent lidar ratio. For the nighttime calculations, the Raman channel at 607 nm is additionally used (Baars et al., 2016). Specifically, the basic lidar equation at 1064 nm can be described by:

$$P^{1064}(R) = C^{1064} \frac{O(R)}{R^2} (\beta_{par}{}^{1064}(R) + \beta_{mol}{}^{1064}(R)) \exp(-2\int_0^R [a_{mol}{}^{1064}(r) + a_{par}{}^{1064}(r)]dr) \qquad (1)$$

And the corresponding lidar equation at 607 nm by:

$$P^{607}(R) = C^{607} \frac{O(R)}{R^2} (\beta_{mol}{}^{607}(R)) \exp(-\int_0^R [a_{mol}{}^{532}(r) + a_{par}{}^{532}(r) + a_{mol}{}^{607}(r) + a_{par}{}^{607}(r)]dr) \qquad (2)$$

a solution for the particle backscatter coefficient at 1064 nm is obtained, using the ratio:

$$\frac{P^{607}(R_0)P^{1064}(R)}{P^{1064}(R_0)P_{607}(R)} \qquad (3)$$

where $P^{607}$ and $P^{1064}$ stand for the power received from a distance R, with respect to the lidar system, at 607 nm and 1064 nm respectively. The constant C at 607 or 1064 nm contains all range independent system parameters. The overlap function O(R), which is less than unity for the altitude range where the laser beam is not completely inside the receiving telescope field of view (Wandinger et al., 2002), is assumed identical between the two channels, which is the case for PollyXT systems which use one beam expander for all three emitted wavelengths. $\beta_{mol}$ and $\beta_{par}$ represent molecular and particle backscattering respectively, whereas $\alpha_{mol}$ and $\alpha_{par}$ are the molecular and particle extinction coefficients.

Finally, in order to perform the intercomparison between CATS and EARLINET profiles, the high resolution of EARLINET profiles was lowered to match the vertical resolution of CATS profiles (i.e. 60m). The objective of obtaining profiles of similar vertical resolution was addressed through computing the EARLINET mean backscatter coefficient value from all EARLINET bins within each CATS 60m backscatter coefficient height range. The computed EARLINET profiles of similar vertical resolution with CATS followed with high accuracy the characterizes and tendencies, both qualitative and quantitative, of the initial EARLINET profiles, despite the loss of vertical resolution (Iarlori et al., 2015).

## 2.4 Demonstration of the comparison methodology for a case study over Athens

To illustrate the evaluation methodology for the CATS Level 2 aerosol backscatter coefficient at 1064 nm, a pair of collocated and concurrent CATS and EARLINET lidar observations is shown in Figure 2. The example refers to a nighttime ISS overpass of the coastal city of Athens-Greece on the 1st of February, 2016. During that period, the PollyXT-NOA system was operating in a 24/7 mode in Athens, at the premises of the National Observatory of Athens, to fulfill the needs of an ACTRIS Joint Research Activity (JRA) for aerosol absorption (Tsekeri et al., 2018). At the same time, on Monday 1st of February 2016, the lidar station operating at the National Technical University of Athens

(NTUA) was performing nighttime measurements according to the EARLINET schedule of regular and simultaneous measurements, in order to enable Raman extinction retrievals. The closest distances between the CATS footprint of the ISS overpass and the locations of the EARLINET-at (NTUA) and EARLINET-no (NOA) stations were approximately 18.58 and 23.3 km at 17:24 UTC (Fig. 2a). The vertical distribution of aerosols and clouds is shown in the CATS 1064 nm backscatter coefficient quicklook (Fig. 2b) and the PollyXT-NOA lidar range-corrected signal at 1064 nm, between 01/02/2016 at 12:00 UTC and 02/02/2016 00:00 UTC (Fig. 2c). The temporal averaging window of the ground-based lidar signal is shifted a few minutes after the ISS overpass (17:45-19:30 UTC), due to routinely/automatic depolarization calibration measurements conducted with PollyXT-NOA system at the exact time of the overpass (Engelmann et al. 2016), while for the EARLINET-at system the temporal averaging window between 18:20:51 and 19:57:41 UTC was used. Both CATS and PollyXT-NOA quicklooks advocate the horizontal and vertical homogeneity of the scene. For the comparison of CATS and EARLINET observations, the latest are regridded to the CATS Level 2 vertical resolution (60 m). Accordingly, CATS spatial averaged and the EARLINET systems of NOA and NTUA temporal averaged backscatter coefficient profiles are qualitative compared (Fig. 2d). The observed disagreements between the two EARLINET profiles are related to differences between the two system, to the different surface elevation of the locations of the two stations (86m for EARLINET-no and 212 for EARLIENT-at), and the different overlap regions. The horizontal-bars in the CATS profile (Fig. 2d) correspond to the standard deviation of the spatially averaged backscatter coefficient profiles.

The comparison of the mean backscatter coefficient profiles retrieved by CATS and the two corresponding EARLINET NOA and NTUA profiles presented in Figure 2 is an initial demonstration of the good agreement between the two products. The CATS instrument reproduces the observed aerosol features, in terms of aerosol load as well as their vertical distribution (Fig. 2d). The assessment of CATS backscatter coefficient is performed in the region between 0.5 km above ground-level of the EARLINET sites, to account for overlap effects between the laser beam and the telescope (Wandinger and Ansmann, 2002), topographic effects, surface returns, and differences of atmospheric samples within the Planetary Boundary Layer (Fig. 2d - shaded area iii), and 10 km height (a.s.l.). An upper limit of $2 \, Mm^{-1}sr^{-1}$ is applied to the aerosol backscatter coefficient values, in order to account for cloud features possible misclassified as aerosols (Fig. 2d - shaded area ii). Finally, cases of EARLINET backscatter coefficient values below the CATS minimum detectable backscatter limit at 1064 nm are not included in the comparison, when the corresponding CATS backscatter coefficient is reported to be zero (Fig. 2d - shaded area i). The latter constrain is applied to account for very thin detected layers from ground-based Lidar systems with backscatter values below the CATS minimum detection limit due to the low Signal-to-Noise Ratio values (SNR). The discussed constrains are employed because of our basic idea to quantitatively assess the representativeness and accuracy of the detected by CATS aerosol features, while preventing possible contaminations (e.g. presence of clouds) to propagate into the CATS-EARLINET dataset.

**3 Results and discussion**

**3.1 EARLINET-CATS Correlative Cases**

To illustrate strengths and limitations of CATS products, we discuss in detail three selected cases of collocated and concurrent CATS-EARLINET observations close to the (EARLINET) stations of Leipzig, Évora and Dushanbe. The three study cases represent different atmospheric conditions with increasing degree of difficulty in the detection of representative aerosol layers by CATS.

**3.1.1 Case I: ISS-CATS over Leipzig - 13/09/2016 03:37 UTC**

The first overpass considered here shows a representative case study of a nighttime ISS orbit, on September 13, 2016 (blue line), at a minimum distance of 3.78 km from the EARLINET Leipzig – Germany PollyXT lidar system (indicated by a white dot), at 03:37 UTC (Fig. 3a). CATS particulate backscatter coefficient cross section at 1064 nm (Fig. 3b) shows the presence of aerosols up to 2.6 km (a.s.l.). CATS feature mask algorithm classifies all of the detected layers as aerosols (not shown). The ground based lidar measurements at Leipzig station between 00:00 and 12:00 UTC did not report any cloud features either, including cirrus clouds. CATS spatial-averaged and Leipzig temporal-averaged profiles were derived from CATS profiles within horizontal distance of 50 km, between the Leipzig station and the ISS footprint, and Leipzig measurements within 90 minutes of the ISS overpass, respectively (Fig. 3c). The direct comparison of the backscatter coefficient profiles, measured from the EARLINET Leipzig station (red line) and CATS (blue line), along with their standard deviations (horizontal error bars), indicate also the presence of aerosol up to 2.6 km height (a.s.l.). The intercompared profiles between ISS-CATS and EARLINET-Leipzig station are characterized by high agreement, although discrepancies are also present. To the uppermost part of the profiles, between 2.5 and 3 km (a.s.l.), due to the higher SNR, Leipzig lidar is capable to detect tenuous atmospheric features of low backscatter coefficient values. Although the case presented and discussed in Figure 3 corresponds to a nighttime ISS overpass, the case is representative for cloud free and relative homogeneous atmospheric scenes in terms of aerosols, for both daytime and nighttime solar background illumination, demonstrating the overall high performance of CATS under such conditions.

Small biases between EARLINET and CATS backscatter coefficient are also identified in specific cases. CATS particulate backscatter coefficient profiles are available for the identified atmospheric features and not as full profiles as in the case of the attenuated backscatter profiles. The feature classification algorithm, assuming no cloud or aerosol layers are detected and no overlaying opaque layers are present, classifies the atmospheric layers as clear-air. Clear-air segments though are not pristine and aerosol-free, as they frequently contain tenuous particulate layers (Kim et al., 2018). Layers of atmospheric features that are not detected, contain either fill values ($0.0 \ km^{-1}sr^{-1}$), or are marked as invalid in cases when the calculation of the particulate backscatter coefficients was not possible (-999.9). This scheme of assigning appropriate backscatter coefficients to the detected atmospheric features (e.g., aerosol and clouds) propagates through many of the Level 2 products in the comparison of CATS Level 2 data, thus in the assessment of the representativeness of CATS observations. Consequently, the comparison of CATS Level 2

backscatter coefficient profiles against EARLINET observations is only possible over the detected atmospheric features. In addition, the identification of the atmospheric features strongly depends on the calibrations of CATS lidar system and to the level of the background signal - solar illumination conditions, due to the different SNR between daytime and nighttime.

### 3.1.2 Case II: ISS-CATS over Évora - 31/05/2016 19:43 UTC

Figure 4 shows a daytime ISS match-up, on May 31, 2016 (red line), at a minimum distance of 39.4 km from the EARLINET station of Évora - Portugal (indicated by a white dot), at 19:43:41 UTC, during a time window of cloud free atmospheric conditions (Fig. 4a). CATS particulate backscatter coefficient cross section at 1064 nm (Fig. 4b) shows the absence of aerosol and/or cloud features, while the Évora temporal-averaged profile during the cloud free window (Fig. 4c) indicates the presence of thin aerosol layers in the altitude range between 1 and 2.5 km height (a.s.l.). The aerosol layer detected by the Évora PollyXT lidar system is characterized by backscatter coefficient values lower than 0.3 Mm$^{-1}$sr$^{-1}$. Although CATS is characterized by relatively low Minimum Detection Thresholds (Yorks et al., 2016), CATS capabilities are limited in terms of detecting similarly tenuous aerosol layers at levels that lie below the detection thresholds (e.g. CATS 7.2 Minimum Detectable Backscatter 1064 nm: Night: 5.00E-5 $\pm$ 77E-5 km$^{-1}$sr$^{-1}$ / Day: 1.30E-3 $\pm$ 0.24E-3 km$^{-1}$sr$^{-1}$ - for cirrus clouds; Yorks et al., 2016). The detection limitation of CATS may propagate in scientific studies implementing CATS through introduced underestimations and possible biases.

The assessment of accuracy of CATS Level 2 against EARLINET collocated and concurrent observations is performed on the basis of backscatter coefficient profiles, because this product constitutes the CATS Level 2 parameter with the lowest influence of a-priori assumptions (e.g. lidar ratio). In addition CATS Level 2 provides the feature classification of the detected layers and associated confidence level of the classification. The Cloud-Aerosol discrimination though is not performed perfectly. Thus misclassified aerosol layers may be classified as clouds, and vice versa. In the framework of the study, for the assessment process of the CATS Level 2 aerosol quality, strict cloud-filtering is applied. In particular, cloud contaminated profiles (Sky Condition 2, 3) and aerosol layers characterized by medium/low classification confidence (Feature_Type_Score $\geq$ -1) are filtered. The strict cloud screening is applied because of our basic idea to establish the accuracy of CATS aerosol backscatter coefficient profiles based on intercomparison against EARLINET, preventing any contamination of cloud features to propagate into the dataset.

### 3.1.3 Case III: ISS-CATS over Dushanbe - 25/05/2015 18:53 UTC

As discussed in the case of Leipzig overpass, on average, the agreement between CATS Level 2 backscatter coefficient profiles and EARLINET is good, especially under relative homogeneous cloud-free atmospheric conditions. Under complex atmospheric conditions though, of coexistent and adjacent aerosol and cloud features, the impact of the CATS Feature Type Score on the CATS aerosol retrievals becomes significant. Figure 5 shows the

CATS footprint for the nighttime ISS orbit, on May 25, 2015 (blue line), at a minimum distance of 24.3 km from the EARLINET Dushanbe - Tajikistan station (Hofer et al., 2017), at 18:53:19 UTC (Fig. 5a). This EARLINET station is located in a natural basin surrounded by mountain ridges of variable height, between 0.7 and 4 km height (a.s.l.). CATS particulate backscatter coefficient cross section at 1064 nm (Fig. 5b) shows the predominant presence of aerosols, up to 3.6 km height (a.s.l.), adjust to broken thin clouds. These cloud characteristics though are not consistent with the observations performed at Dushanbe station between 13:00 and 23:00 UTC on May 25, 2015 that reported the absence of cloud features below 6 km. CATS lidar profile and the EARLINET-Dushanbe profile yield different behavior in terms of backscatter coefficient (Fig. 5c). The Dushanbe lidar reports a weak presence of aerosols, up to approximately 4 km height (a.s.l.). The backscatter comparison against CATS profile reveals enhanced discrepancies in segments of the CATS profile, denoted by the high backscatter coefficient values ($> 2$ $Mm^{-1}sr^{-1}$). The cloud features that cause the observed discrepancies are classified by CATS CAD algorithm as aerosol layers, contaminating the CATS profile, despite the strict cloud screening. Features with invalid CATS CAD Score, although not frequently observed, may impact the quality of the column aerosol optical depth (AOD) and related climatological studies. In addition, complex topography in terms of geographical characteristics, erroneous mean backscatter coefficient profiles due to the high variability of aerosol load in the Planetary Boundary Layer, the horizontal distance between the CATS lidar footprint and the ground-based lidar stations and surface returns enhance further these discrepancies, especially in the lowermost part of the profiles. Based on this analysis and comparisons with CALIPSO, the CATS cloud-aerosol discrimination algorithm was updated for the V3-00 Level 2 data products (released in the end of 2018) to improve the accuracy of the Feature Type and Feature Type Score, especially during daytime.

**3.2 EARLINET-CATS comparison statistics**

In this section an overall assessment of the CATS backscatter coefficient product at 1064 nm is given, using the entire dataset of CATS-EARLINET collocated profiles. To address quantitatively the accuracy and representativeness of the satellite-based lidar retrievals the estimation of possible biases in the CATS backscatter coefficient is performed. Towards this assessment, in the comparison of CATS against EARLINET we implement the $CATS_i$-$EARLINET_i$ residuals for each pair of observations "i", as statistical indicator of CATS average overestimation or underestimation of the aerosol load, in terms of backscatter coefficient values.

Figure 6 shows the distributions of $CATS_i$-$EARLINET_i$ backscatter coefficient differences. On average, the agreement is good demonstrating the high performance of CATS, with mean and median residual values close to zero and typically within 0.4 $Mm^{-1}sr^{-1}$. The intercomparison between CATS satellite-based and EARLINET ground-based lidar retrievals reveals the presence of negative biases in the CATS 1064 nm backscatter coefficient profiles. The $CATS_i$-$EARLINET_i$ differences, for all the available 21 daytime (Fig. 6a) and 26 nighttime (Fig. 6b) cases of paired correlative observations show an underestimation of the CATS retrievals, more pronounced during daytime than nighttime. In the case of daytime observations, the calculated mean (median) CATS difference from EARLINET is -0.123 $Mm^{-1}sr^{-1}$ (-0.095 $Mm^{-1}sr^{-1}$). In the case of nighttime observations, the corresponding mean (median)

difference from EARLINET is -0.031 $Mm^{-1}sr^{-1}$ (-0.065 $Mm^{-1}sr^{-1}$). The observed standard deviation (SD) is 0.431 $Mm^{-1}sr^{-1}$ over daytime and 0.342 $Mm^{-1}sr^{-1}$ during nighttime. During daytime, minimum and maximum CATS-EARLINET residual values of -1.802 $Mm^{-1}sr^{-1}$ and 1.189 $Mm^{-1}sr^{-1}$ are observed, while the corresponding minimum and maximum values for nighttime are -1.348 $Mm^{-1}sr^{-1}$ and 1.149 $Mm^{-1}sr^{-1}$. The $CATS_i$-$EARLINET_i$ daytime mean absolute bias and root mean square error (RMSE) statistical indicators (Binietoglou et al., 2015) of daytime observations are 0.323 $Mm^{-1}sr^{-1}$ and 0.448 $Mm^{-1}sr^{-1}$, while the respective statistical indicators for the nighttime cases are 0.249 $Mm^{-1}sr^{-1}$ and 0.343 $Mm^{-1}sr^{-1}$. CATS performance is also quantified through the linear correlation coefficient between the CATS and EARLINET backscatter coefficient distributions, with correlation coefficient of 0.54 and 0.69, during daytime and nighttime respectively. The correlations between CATS and EARLINET distributions are not very good, as expected due to the significant influence of the topography, the high inhomogeneities within the local Planetary Boundary Layer (PBL), and the effect of the horizontal distance and temporal measurement differences. The fractional bias values for daytime and nighttime are -0.676 and -0.773 respectively, while the fractional gross error ranges between 1.061 for daytime and 0.999 for nighttime cases. Overall, the agreement between CATS and EARLINET is good. On average though, slight underestimations of CATS compared to EARLINET are observed, 6.3 % during nighttime and 22.3 % during daytime. The intercomparison statistical values between CATS and EARLINET are summarized in Table 4.

Figure 7 reports the mean aerosol backscatter coefficient profiles at 1064 nm as provided by CATS and EARLINET daytime (Fig. 7a) and nighttime (Fig. 7b) lidar observations. On average, the mean aerosol backscatter coefficient profiles reveal similar characteristics between CATS and EARLINET, although the comparisons are subject to the different number of available cases, 21 and 26 for daytime and nighttime respectively. Both CATS and EARLINET daytime and nighttime backscatter coefficient profiles yield higher values close to the surface level, gradually decreasing with altitude. Especially in the range between the full overlap region of the laser beam and the telescope of the EARLINET systems (approximately 1 km) and the middle free-troposphere (~6 km a.s.l.), the mean backscatter coefficient profile of CATS is well within the standard deviation of the EARLINET provided scenes. Nonetheless, discrepancies are also evident. CATS, as a result of the high spatial atmospheric variability, yields usually higher values of standard deviation than EARLINET. In addition, at altitudes higher than 6 km (a.s.l.), CATS mean backscatter coefficient profile yields zero or close-to-zero values, while EARLINET shows the presence of elevated aerosols, with rather low mean backscatter values, lower than 0.2 $Mm^{-1}sr^{-1}$.

The CATS Level 2 backscatter coefficient product evaluation study shows that CATS agrees reasonably well with ground-based EARLINET measurements, although generally biased low. To assess the ability of CATS lidar to detect aerosol features and optical properties and to shed light on the origin of observed CATS-EARLINET discrepancies the conducted CALIOP validation studies offer an unprecedented basis. This is due to the similar viewing geometry between CATS and CALIOP and to the similarities between Level 1B and Level 2 processing algorithms (McGill et al., 2015; Yorks et al., 2016; 2019).

Since CALIPSO joined the A-Train constellation of Earth observation satellites in June 2006 (Winker et al., 2007), several studies have been conducted to validate and evaluate CALIOP Level 1B, Level 2 and Level 3 products, against ground-based, airborne, and spaceborne measurements. Airborne NASA Langley HSRL (Hair et al., 2008)

and CPL (McGill et al., 2002) flights, of close spatial and temporal coincidence with the CALIPSO satellite documented on the high performance of CALIOP, although with the presence of low negative biases (Burton et al., 2010; 2013; McGill et al., 2007; Rogers et al., 2011; 2014). Kacenelenbogen et al. (2014) reports on the detection of aerosols-above-cloud (AAC) in only 151 of 668 CALIOP-HSRL coincident airborne cases (23 %). The use of ground-based Raman lidar observations also reports that CALIOP Level 1B and Level 2 products are biased low (Mamouri et al. 2009; Mona et al., 2009; Pappalardo et al., 2010; Tesche et al., 2013). In terms of columnar measurements, the conducted validation activities based on collocated observations between CALIOP and AErosol RObotic NETwork (AERONET; Dubovik et al., 2000) showed CALIPSO AOD underestimations (Amiridis et al., 2013; Omar et al., 2013; Schuster et al., 2012). In addition, evaluation studies of AOD observations from the passive spaceborne MODerate resolution Imaging Spectroradiometer (MODIS; Remer et al., 2005) show that CALIOP provides reasonably well known climatic features, although with apparent AOD underestimations (Amiridis et al., 2013; Kittaka et al., 2011; Oo and Holz, 2011; Redemann et al., 2012). The magnitude of the documented agreements and biases in the detection of aerosol features vary from study to study, with respect to the different CALIOP versions. Substantially improvement in the detection of aerosol features is expected in the latest CALIPSO Version 4 (AMT CALIPSO special issue).

Overall, CATS, much like CALIOP, observes reasonably well the vertical distribution of atmospheric aerosol backscatter coefficient, although with slight underestimations. The observed discrepancies in the compared CATS-EARLINET profiles are attributed to several sources.

First, the retrieval accuracy of CATS Level 2 data products, such as the aerosol and cloud backscatter and extinction coefficient profiles, the vertical feature mask and the integrated parameters (e.g. AOD), depends crucially on the calibration of the lidar system and the calibration region (Kar et al., 2018). CATS total attenuated backscatter from molecules and particles in the atmosphere is performed in the calibration region between 22 and 26 km starting with V2-08 of the L1B data (Russell et al., 1979; Del Guasta 1998; McGill et al., 2007; Powell et al., 2009). Uncertainties in the CATS Level 1B backscatter calibration are attributed to random and systematic errors (CATS ATBD). Random errors result mainly from normalizing the 1064 nm lidar signal to modeled molecular signal and are dominated by lidar noise. On the contrary, systematic errors result from a number of different sources, including uncertainties in the CALIOP stratospheric scattering ratios and molecular backscatter coefficient values generated from the Goddard Earth Observing System (GEOS) atmospheric general circulation model and assimilation system used to calculate molecular and ozone atmospheric transmission (Rienecker et al., 2008), and from the non-ideal performance of CATS. The total uncertainty due to the CATS calibration constants is estimated between 5% and 10% (CATS ATBD).

Secondly, CATS detection and classification schemes, similar to CALIOP, provide Level 2 aerosol products only in regions where aerosol features are detected and identified. This implies that optically thin aerosol layers can go undetected by CATS, due to weak backscattering intensities below the CATS detection thresholds (Kacenelenbogen et al., 2014; Thorsen et al., 2015). To increase the detection of tenuous aerosol layers CATS incorporates an iterated horizontal averaging scheme (5 and 60 km; Yorks et al., 2019). Failures of spaceborne lidar instruments and algorithms to detect tenuous aerosol layers (Toth et al., 2018) result in range bin backscatter coefficient assignments

to 0.0 $Mm^{-1}sr^{-1}$. The faint undetected aerosol layers do not contribute to the CATS aerosol backscatter profiles, consequently neither to extinction coefficient profiles, nor to estimates of CATS AOD, similar to CALIOP AOD (Kim et al., 2013; Rogers et al., 2014; Thorsen and Fu, 2015). The detection sensitivity is attributed to the solar background and sunlight illumination conditions, due to the significantly lower CATS SNR during daytime than nighttime (Rogers et al., 2014). The undetected aerosol layers, although of low aerosol load, introduce negative biases in the CATS-EARLINET comparison. The total uncertainty, the sum of the systematic and random errors, in the CATS ATB at 1064nm is estimated at 10-20% for nighttime data and 20-30% for daytime data.

Another source of discrepancy between CATS and EARLINET is attributed to the effect of horizontal distance between the ground-based lidar systems and the space-based lidar footprint. Studies performed in the framework of EARLINET attribute an introduced discrepancy of the order of 5 % to the intercompared profiles, when the horizontal distance is below 100 km (Mamouri et al., 2009; Pappalardo et al., 2010; Papagiannopoulos et al., 2016). The different - opposite viewing geometry (upward for EARLINET/downward for CATS/CALIPSO) and the different transmittance terms are further sources of discrepancies (Mona et al., 2009). In addition, enhanced disagreements observed between CATS and EARLINET in the lowermost part of the mean backscatter coefficient profiles are attributed to the high spatial and temporal variability of the aerosol content within the PBL, to the complexity of the local topography and to surface returns.

Finally, regarding the utility of CATS for climatic studies, another common reason of satellite-based lidar overestimations or underestimations is attributed to the absence of detailed aerosol properties in the classification of the detected aerosol layers. The aerosol-subtype classification scheme frequently results in aerosol layer misclassifications, as has been shown in the case of coincident HSRL-CALIPSO under-flights (Burton et al., 2012). Misclassified aerosol layers incorporate erroneous values of lidar ratio. Possible underestimation or overestimation of aerosol backscatter coefficient profiles, considered with erroneous aerosol-subtype classification, introduce biases in corresponding extinction coefficient profiles and eventually in total columnar AOD retrievals. The CATS V3-0 Level 2 data products improve errors in cloud-aerosol typing identified in these CATS-EARLINET comparisons. Furthermore, Wandinger et al. (2010), based on CALIOP extinction coefficient profiles in case of dust aerosol layers and collocated ground-based Raman lidar measurements, showed that multiple scattering effects can result in negative biases if not considered in the algorithm inversions schemes. Data users should be aware of these multiple scattering effects and cloud-aerosol typing errors when using the CATS data for climate studies, utilizing the CATS Feature Type Scores to reduce uncertainties in the analysis.

## 4. Summary and conclusions

This study implements independent retrievals carried out at several EARLINET stations, to qualitatively and quantitatively assess the performance of the NASA's CATS lidar operating onboard the ISS from February 2015 to October 2017. We compared satellite-based CATS and ground-based independent measurements over twelve high-performance EARLINET stations across Europe and one located in Central Asia. Our analysis is based on the first twenty months of CATS operation (02/2015-09/2016). Comparison of CATS Level 2 and EARLINET backscatter

coefficient profiles at 1064 nm is allowed only in cases of maximum distance between the ISS overpass and the EARLINET stations below 50 km. EARLINET contributed with observations as close in time as possible, typically with starting time or stop time of the measurements within 90 minutes of the ISS overpass. The analysis was restricted to cloud-free profiles to avoid possible cloud-contamination of the intercompared aerosol backscatter coefficient profiles.

In the quantitative assessment of the performance of CATS, 47 collocated, concurrent and cloud-free measurements of CATS the EARLINET were identified (21 daytime and 26 nighttime), offering a unique opportunity for the evaluation of the spaceborne lidar system. The results of the generic comparison are encouraging, demonstrating the overall good performance of CATS, although with negative biases. The agreement, as expected due to higher SNR, is better during nighttime operation, with observed underestimation of 22.3 %, during daytime and 6.1 % during nighttime respectively.

In addition to the generic comparison, three CATS-EARLINET comparison cases were examined to demonstrate the system's performance, under different study conditions. The comparison showed that under cloud-free, relative homogeneous atmospheric aerosol conditions, the spatial averaged CATS backscatter coefficient profiles are in good agreement with EARLINET, independently of light conditions. The deficiency of CATS though to detect tenuous aerosol layers, due to the inherent limitations of space-based lidar systems, may lead to systematic deviations and slight underestimations of the total AOD in climatic studies. In addition, the CATS V2-01 Feature Type Score misclassification of aerosol layers as clouds, and vice versa, in cases of coexistent and/or adjacent aerosol and cloud features, may lead to non-representative, unrealistic and cloud contaminated aerosol profiles. While CATS feature identification will improve in V3-01 data products, the most crucial reason for the observed discrepancies between CATS and EARLINET in the lowermost part of the profiles is related to the complexity of the topography and the geographical characteristics. Especially in the case of large elevation/slope differences, the effects of both inadequate sampling lower than the maximum elevation and of the different atmospheric sampling volumes, result in large AOD biases and unrealistic AOD values.

The qualitative and quantitative agreement between CATS and EARLINET reported in this study is encouraging, especially during nighttime, agreement that will hopefully facilitate further studies implementing CATS observations in the future. CATS, for a period of almost three years, provided an unprecedented global dataset of vertical profiles of aerosols and clouds, much like CALIOP, taking though advantage of the unique orbital characteristics of the ISS. ISS enabled CATS to provide for the first time satellite-based lidar measurements of the diurnal evolution of aerosols and clouds over the tropics and midlatitudes, and to be more specific to latitudes below 52°. Since CALIPSO and Aeolus (and in the future also EarthCARE) are polar sun-synchronous satellites of fixed equatorial crossing time (01:30 and 13:30 LT for CALIOP, 06:00 and 18:00 for ALADIN), it is expected that, at least for the near future, CATS dataset will remain the only available satellite-based lidar source of nearly global diurnal measurements of atmospheric aerosols and clouds. In addition, while CALIOP is a two-wavelength lidar system operating at 532 nm and 1064 nm with depolarization capabilities at 532 nm, CATS provided satellite-based aerosol and cloud depolarization profiles at 1064 nm, thus in a different wavelength. This dataset, much like CALIOP dataset, is especially useful for studies of the three-dimensional distribution of non-spherical aerosol particles in the atmosphere

(e.g. mineral dust and volcanic ash), and especially since it is an active sensor, over regions of high reflectivity (e.g. deserts, ice). Future studies including the exploitation of CATS unique observations may help the scientific community to shed new light on physical processes of aerosols and clouds in the Earth's atmosphere.

**Author contribution.** E. Proestakis coordinated the project, communicated with all EARLINET groups and CATS Team, collected all EARLINET and CATS data, directed the study and prepared the manuscript with contributions from all co-authors. V. Amiridis directed the preparation of the manuscript and supervised the study. EARLINET co-authors have contributed with performed and processed ground-based lidar measurements of the vertical distribution of aerosols at fourteen stations: V. Amiridis, E. Marinou, A. Gialitaki and E. Tetoni at the EARLINET-

no station (Athens-Greece during the study period), U. Wandinger, A. Ansmann, R. Engelmann and H. Baars at the EARLINET-le station (Leipzig, Germany), D. Althausen, J. Hofer, A. Makhmudov and S. Abdullaev at the EARLINET-du station (Dushanbe, Tajikistan), A. Papayannis and M. Mylonaki at the EARLINET-at station (Athens, Greece), D. Balis, N. Siomos and K. A. Voudouri at the EARLINET-th station (Thessaloniki, Greece), D. Nicolae and I. Binietoglou at the EARLINET-bu station (Bucharest, Romania), I. Mattis at the EARLINET-oh station

(Observatory Hohenpeissenberg, Germany), I. S. Stachlewska at the EARLINET-wa station (Warsaw, Poland), M. Sicard and C. Muñoz-Porcar at the EARLINET-ba station (Barcelona, Spain), A. Apituley and D. Mamali at the EARLINET-ca station (Cabauw, Netherlands), D. Bortoli and M. J. Costa at the EARLINET-ev station (Evora, Portugal), M. R. Perrone and P. Burlizzi at the EARLINET-lc station (Lecce, Italy), G. L. Liberti and D. Dionisi at the EARLINET-lm station (Roma-Tor Vergata, Italy), A. Pietruczuk and A. Szkop at the EARLINET-be station

(Belsk, Poland), and G. Pappalardo, L. Mona and N. Papagiannopoulos at the EARLINET-po station (Potenza, Italy). EARLINET co-authors participated in the maintenance and calibration of the lidar systems throughout the study period, data curation and preprocessing. J. Yorks, E. Nowottnick and R. Pauly provided advice and support throughout the process regarding the NASA's CATS lidar system.

**Competing interests.** The authors declare that they have no conflict of interest.

**Acknowledgements**

The authors acknowledge EARLINET for providing aerosol lidar profiles (EARLINET;

https://www.earlinet.org/index.php?id=earlinet_homepage, last access: 20 December 2018). This project receives funding from the European Union's Horizon 2020 research and innovation programme under grant agreements No 654109 and 739530. The authors acknowledge the ISS NASA Research Office (NRO) for the CATS instrument and the NASA Science Mission Directorate (SMD) for the CATS data products and processing algorithms. CATS browse images and data products are freely distributed via the CATS web site at http://cats.gsfc.nasa.gov/data/ (last access:

20 December 2018). Emmanouil Proestakis, Anna Gialitaki and Eleni Tetoni acknowledge support from the Stavros Niarchos Foundation. The research leading to these results acknowledge support through the European Research Council under the European Community's Horizon 2020 research and innovation framework program / ERC Grant

Agreement 725698 (D-TECT). We acknowledge support of this work by the project "PANhellenic infrastructure for Atmospheric Composition and climatE change" (MIS 5021516) which is implemented under the Action "Reinforcement of the Research and Innovation Infrastructure", funded by the Operational Programme "Competitiveness, Entrepreneurship and Innovation" (NSRF 2014-2020) and co-financed by Greece and the

European Union (European Regional Development Fund). The Portuguese team acknowledges the support from the Portuguese Science Foundation (FCT), in the frame of the European Regional Development Fund - COMPETE 2020, under the projects UID/GEO/04683/2013 (POCI-01-0145-FEDER-007690). Lucia Mona and Nikolaos Papagiannopoulos acknowledge the European Union through the EU's Horizon 2020 research and innovation program for societal challenges – smart, green and integrated transport under grant agreement no. 723986 (project

EUNADICS-AV – European Natural Disaster Coordination and Information System for Aviation). The measurements in Tajikistan were funded by the German Federal Ministry of Education and Research (BMBF) in the context of "Partnerships for sustainable problem solving in emerging and developing countries" under the grant number 01DK14014. Lidar measurements in Barcelona were also supported by the Spanish Ministerio de Economía y Competitividad (project TEC2015-63832-P) and EFRD (European Fund for Regional Development); the Spanish

Ministry of Science, Innovation and Universities (project CGL2017-90884-REDT), and the Unidad de Excelencia Maria de Maeztu (project MDM-2016-0600) financed by the Spanish Agencia Estatal de Investigación.

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

**Tables and Figures**

**Table 1:** Contributing EARLINET Lidar Stations, including Identification Codes, Geographical Coordinates and elevation.

| EARLINET Station | Identification Code | Latitude ($^{\circ}$N) | Longitude ($^{\circ}$E) | Altitude a.s.l. (m) |
|---|---|---|---|---|
| Athens-NOA | no | 37.97 | 23.72 | 86 |
| Athens-NTUA | at | 37.96 | 23.78 | 212 |
| Barcelona | ba | 41.39 | 2.12 | 115 |
| Belsk | be | 51.83 | 20.78 | 180 |
| Bucharest | bu | 44.35 | 26.03 | 93 |
| Cabauw | ca | 51.97 | 4.93 | 0 |
| Dushanbe | du | 38.56 | 68.86 | 864 |
| Évora | ev | 38.57 | -7.91 | 293 |
| Observatory Hohenpeissenberg | oh | 47.8 | 11.01 | 974 |
| Lecce | lc | 40.33 | 18.10 | 30 |
| Leipzig | le | 51.35 | 12.43 | 90 |
| Potenza | po | 40.60 | 15.72 | 760 |
| Thessaloniki | th | 40.63 | 22.95 | 50 |
| Warsaw | wa | 52.21 | 20.98 | 112 |

Table 2: ISS-CATS and EARLINET cases considered in the evaluation process of CATS backscatter coefficient profiles at 1064 nm.

| Day-Night Flag | Date yyyy/mm/dd | Time hh:mm:ss (UTC) | EARLINET station | min Distance (km) | EARLINET Date (yyyy/mm/dd) \| measuring time cloud-free window (UTC) |
|---|---|---|---|---|---|
| N | 2015/11/25 | 03:44:09 | Athens | 40.42 | 2015/11/25 \| 03:30:00 – 04:30:00 |
| N | 2016/01/29 | 01:46:08 | Athens | 46.84 | 2016/01/29 \| 01:00:00 – 02:30:00 |
| N | 2016/02/01 | 17:23:36 | Athens | 23.29 | 2016/02/01 \| 17:45:00 – 19:30:00 |
| N | 2016/02/01 | 17:23:37 | Athens NTUA | 18.58 | 2016/02/01 \| 18:20:51 – 19:57:41 |
| D | 2016/05/03 | 06:45:15 | Barcelona | 45.93 | 2016/05/03 \| 08:59:00 – 09:59:00 |
| D | 2015/08/13 | 17:29:18 | Belsk | 2.39 | 2015/08/13 \| 18:02:10 – 18:45:40 |
| N | 2016/08/08 | 17:34:50 | Belsk | 6.56 | 2016/08/08 \| 17:31:08 – 18:12:05 |
| N | 2016/07/28 | 19:15:24 | Bucharest | 45.35 | 2016/07/28 \| 17:41:22 – 18:41:22 |
| N | 2016/09/14 | 04:21:09 | Cabauw | 21.01 | 2016/09/14 \| 05:27:25 – 06:00:03 |
| N | 2015/08/03 | 21:40:39 | Dushanbe | 42.64 | 2015/08/03 \| 20:00:00 – 22:00:00 |
| N | 2016/08/14 | 15:39:07 | Dushanbe | 22.08 | 2016/08/14 \| 15:57:00 – 17:19:00 |
| D | 2015/06/20 | 08:38:33 | Dushanbe | 13.33 | 2015/06/20 \| 08:54:00 – 09:07:00 |
| D | 2015/07/12 | 06:47:07 | Dushanbe | 33.46 | 2015/07/12 \| 06:25:00 – 07:10:00 |
| D | 2016/05/02 | 07:35:38 | Evora | 47.27 | 2016/05/02 \| 07:58:50 – 08:00:21 |
| D | 2016/05/31 | 19:43:41 | Evora | 39.42 | 2016/05/31 \| 19:29:56 – 19:59:35 |
| N | 2016/01/30 | 00:50:16 | Hohenpeissenberg | 13.36 | 2016/01/30 \| 00:20:00 – 01:20:00 |
| N | 2016/03/17 | 02:12:09 | Hohenpeissenberg | 43.40 | 2016/03/17 \| 01:42:00 – 02:42:00 |
| D | 2015/10/31 | 12:56:05 | Hohenpeissenberg | 34.41 | 2015/10/31 \| 12:26:00 – 13:26:00 |
| D | 2016/04/12 | 15:29:18 | Hohenpeissenberg | 12.77 | 2016/04/12 \| 14:55:00 – 16:05:00 |
| D | 2016/08/07 | 16:49:29 | Hohenpeissenberg | 31.81 | 2016/08/07 \| 16:19:30 – 17:19:30 |
| D | 2016/08/23 | 10:42:43 | Hohenpeissenberg | 36.11 | 2016/08/23 \| 10:12:30 – 11:12:30 |
| D | 2016/09/14 | 05:58:59 | Hohenpeissenberg | 28.37 | 2016/09/14 \| 04:59:00 – 05:59:00 |
| N | 2015/07/27 | 21:14:35 | Lecce | 34.69 | 2015/07/27 \| 20:42:00 – 21:09:00 |
| N | 2016/08/04 | 22:44:06 | Lecce | 4.72 | 2016/08/04 \| 20:50:00 – 21:20:00 |
| N | 2015/07/30 | 00:18:19 | Leipzig | 41.16 | 2015/07/30 \| 00:34:00 – 01:04:00 |
| N | 2015/08/03 | 21:29:44 | Leipzig | 15.81 | 2015/08/03 \| 21:31:00 – 22:00:00 |
| N | 2015/09/24 | 01:13:34 | Leipzig | 25.05 | 2015/09/24 \| 01:01:00 – 01:30:00 |
| N | 2015/09/29 | 00:05:33 | Leipzig | 36.49 | 2015/09/28 \| 22:42:00 – 23:12:00 |
| N | 2015/09/29 | 23:13:24 | Leipzig | 48.46 | 2015/09/28 \| 22:55:00 – 23:24:00 |
| N | 2015/09/30 | 22:21:13 | Leipzig | 12.89 | 2015/09/30 \| 21:25:00 – 21:34:00 |
| N | 2016/06/05 | 20:14:01 | Leipzig | 36.93 | 2016/06/05 \| 20:02:00 – 20:31:00 |
| N | 2016/09/13 | 03:37:49 | Leipzig | 3.79 | 2016/06/05 \| 00:00:00 – 02:30:00 |
| N | 2016/09/12 | 04:29:46 | Leipzig | 45.08 | 2016/09/12 \| 00:00:00 – 02:30:00 |
| N | 2016/09/15 | 03:30:25 | Leipzig | 48.36 | 2016/09/15 \| 00:00:00 – 02:30:00 |
| D | 2015/04/21 | 14:54:35 | Leipzig | 6.73 | 2015/04/21 \| 16:04:00 – 16:33:00 |
| D | 2015/04/21 | 16:31:00 | Leipzig | 31.28 | 2015/04/21 \| 16:34:00 – 17:04:00 |
| D | 2015/04/24 | 15:25:13 | Leipzig | 47.83 | 2015/04/24 \| 14:03:00 – 14:32:00 |
| D | 2015/08/13 | 17:27:54 | Leipzig | 1.36 | 2015/08/13 \| 19:01:00 – 19:30:00 |
| D | 2016/08/24 | 11:26:39 | Leipzig | 3.46 | 2016/08/24 \| 10:00:00 – 12:00:00 |
| D | 2016/08/24 | 13:03:12 | Leipzig | 48.97 | 2016/08/24 \| 10:00:00 – 12:00:00 |
| N | 2015/07/21 | 00:13:26 | Potenza | 2.01 | 2015/07/21 \| 00:00:00 – 02:52:19 |
| D | 2015/11/06 | 10:54:52 | Thessaloniki | 19.46 | 2015/11/06 \| 11:57:03 – 12:27:20 |
| N | 2016/01/28 | 19:17:11 | Thessaloniki | 39.54 | 2016/01/28 \| 20:08:40 – 20:38:57 |
| D | 2015/08/13 | 17:29:20 | Warsaw | 42.95 | 2015/08/13 \| 17:00:00 – 17:22:00 |
| D | 2015/08/19 | 15:22:30 | Warsaw | 44.47 | 2015/08/19 \| 15:25:00 – 15:47:00 |
| D | 2016/06/07 | 18:29:46 | Warsaw | 41.22 | 2016/06/07 \| 18:15:00 – 18:43:00 |
| N | 2016/08/08 | 17:34:53 | Warsaw | 46.99 | 2016/08/08 \| 17:00:00 – 17:23:00 |

**Table 3:** List of CATS quality assurance thresholds applied in the EARLINET comparison.

| Mode | 7.2 |
|---|---|
| Level | 2 |
| Parameter | Backscatter Coefficient |
| Wavelength | 1064nm |
| Distance | ≤ 50km radius from the EARLINET stations |
| Feature Type Score | ≤ -2 |
| Sky Condition | 0 – clean skies *and* 1 – clear skies (no clouds) |
| Backscatter Coefficient | $0 \leq b_{1064nm} \leq 2$ [Mm$^{-1}$sr$^{-1}$] |
| Vertical range window | ≤ 10 km (a.s.l.) |

**Table 4:** CATS-EARLINET comparison statistics on mean bias, median, mean absolute bias, standard deviation, root mean square error (RMSE), minimum/maximum values on the observed backscatter coefficient profiles at 1064 nm (Mm$^{-1}$sr$^{-1}$) for daytime and nighttime correlative cases.

| Metric | Daytime | Nighttime |
|---|---|---|
| Mean Bias [Mm$^{-1}$sr$^{-1}$] | -0.123 | -0.031 |
| Median Differences [Mm$^{-1}$sr$^{-1}$] | -0.094 | -0.065 |
| Mean Absolute Bias [Mm$^{-1}$sr$^{-1}$] | 0.323 | 0.249 |
| Mean Relative Bias [%] | -24.062 | -19.843 |
| SD [Mm$^{-1}$sr$^{-1}$] | 0.431 | 0.342 |
| (min / max Differences) [Mm$^{-1}$sr$^{-1}$] | (-1.802 / 1.189) | (-1.348 / 1.149) |
| RMSE [Mm$^{-1}$sr$^{-1}$] | 0.448 | 0.343 |
| Correlation Coefficient | 0.547 | 0.694 |
| Fractional Bias | -0.773 | -0.676 |
| Fractional Gross Error | 0.999 | 1.061 |
| Number of Cases (#) | 21 | 26 |

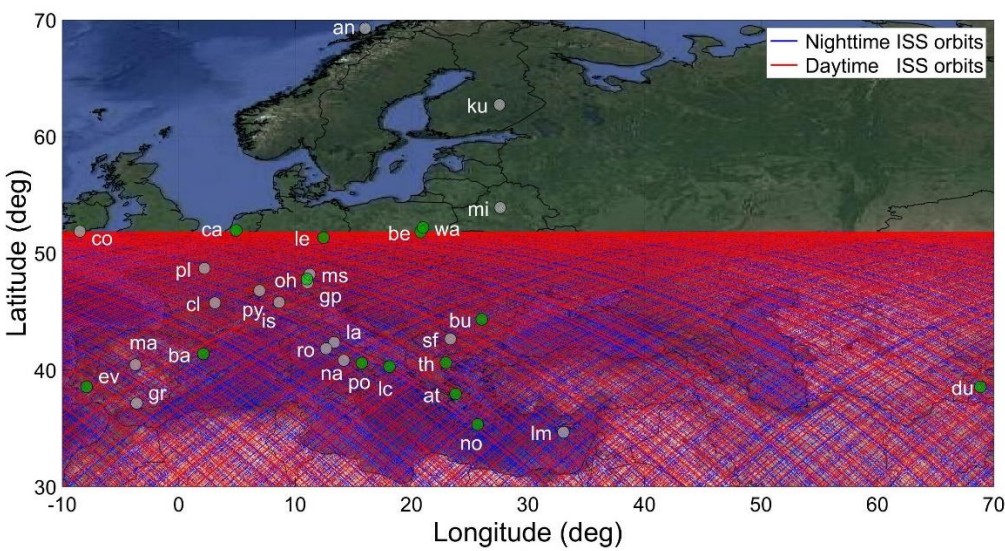

**Figure 1: Distribution of EARLINET lidar stations over Europe and West Asia. Green dots: stations used in the inter-comparison. ISS orbits between 02/2015 and 09/2016 are overlaid in red for daytime and in blue for nighttime overpasses.**

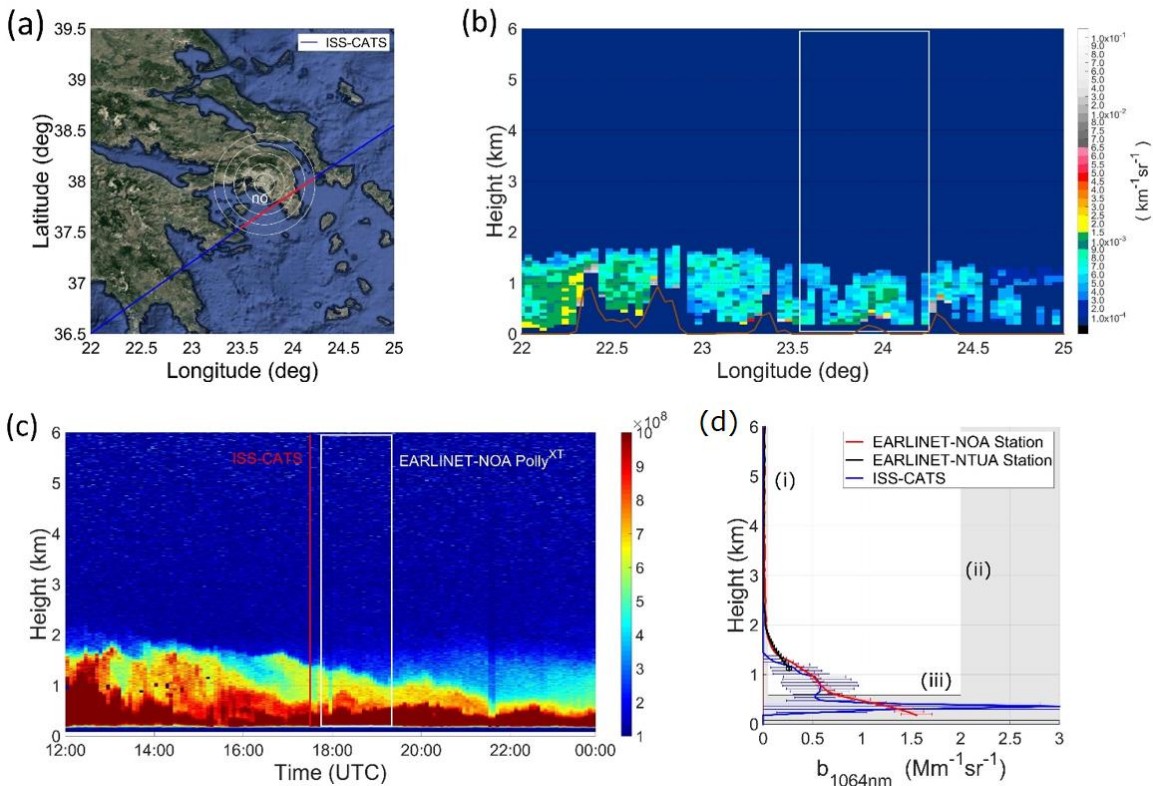

**Figure 2: (a) Nighttime ISS orbit over Athens-Greece on the 1st of February 2016 (blue line). The concentric white circles denote regions of 10, 20, 30, 40 and 50 km from the location of PollyXT-NOA lidar system (white dot). Red colour in the ISS footprint indicates CATS observations within 50 km distance from the NOA PollyXT lidar system. (b) CATS Backscatter Coefficient at 1064 nm on 2016-02-01, 17:24 UTC. The white box depicts CATS observations used for the profile intercomparison. (c) PollyXT-NOA range-corrected signal time-series at 1064 nm. The white box delineates the temporal averaging of the lidar signals (17:45-19:30 UTC) while the red line denotes the ISS overpass at 2016-02-01, 17:24 UTC - closest distance time. (d) CATS (blue line) and PollyXT-NOA (red line) mean profiles and standard deviations of backscatter coefficient at 1064 nm (0-6 km).**

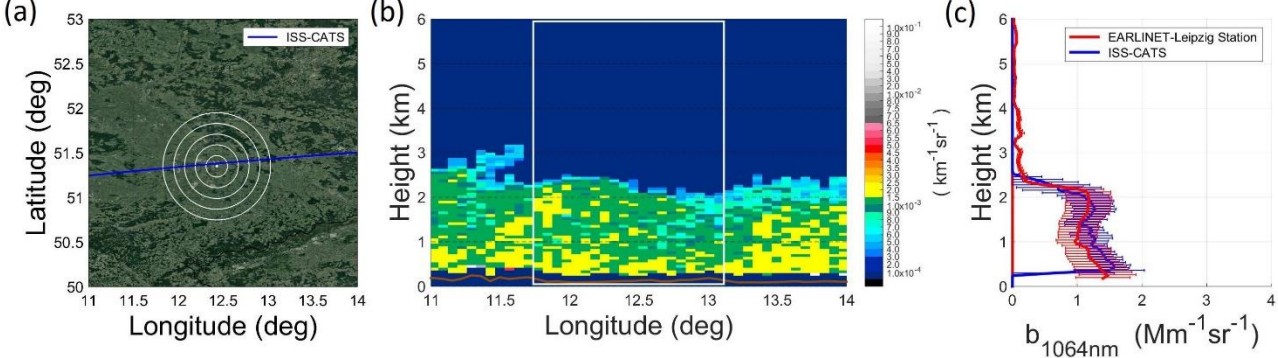

**Figure 3: (a) Nighttime ISS orbit over the EARLINET Leipzig station on the 13th of September 2016 at 03:37:49 UTC and of closest distance between the footprint of CATS and the EARLINET- Leipzig station of 3.79km. The white dot denotes the location of Leipzig lidar system while the blue line shows the lidar footprint of CATS. (b) CATS Backscatter Coefficient at 1064 nm. (c) CATS (blue line) spatially and EARLINET-Leipzig (red line) temporarily averaged backscatter coefficient profiles (1064 nm). The implemented EARLINET-Leipzig time window of cloud-free measurements was between 00:00:00 and 02:30:00 UTC. The horizontal blue and red lines denote the variability (one standard deviation) of the CATS and EARLINET measured atmospheric scenes, respectively.**

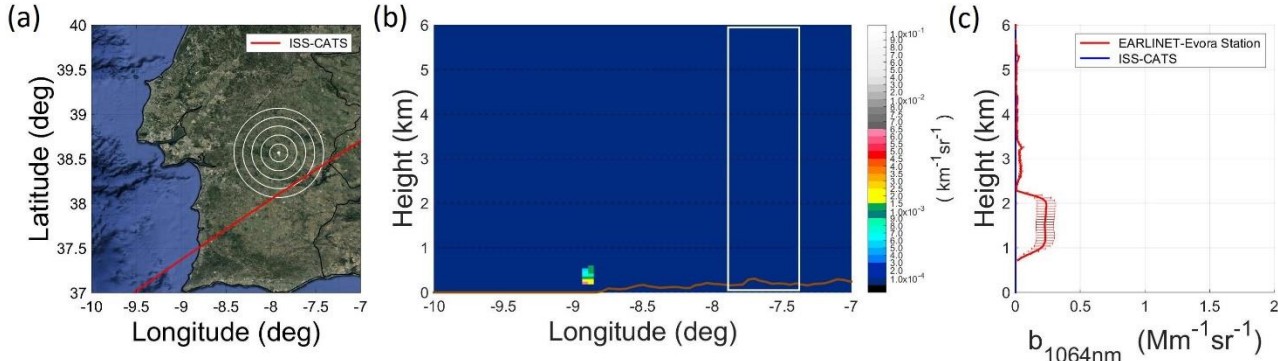

**Figure 4: (a) Daytime ISS orbit over the EARLINET Évora station on the 31th of May 2016 at 19:43:31 UTC and of closest distance between the footprint of CATS and the EARLINET- Évora station of 39.42km. The white dot denotes the location of Évora lidar system while the red line shows the lidar footprint of CATS. (b) CATS Backscatter Coefficient at 1064 nm. (c) CATS (blue line) spatially and EARLINET-Évora (red line) temporarily averaged backscatter coefficient profiles (1064 nm). The implemented EARLINET-Évora time window of cloud-free measurements was between 19:29:56 and 19:59:35 UTC. The horizontal blue and red lines denote the variability (one standard deviation) of the CATS and EARLINET measured atmospheric scenes, respectively.**

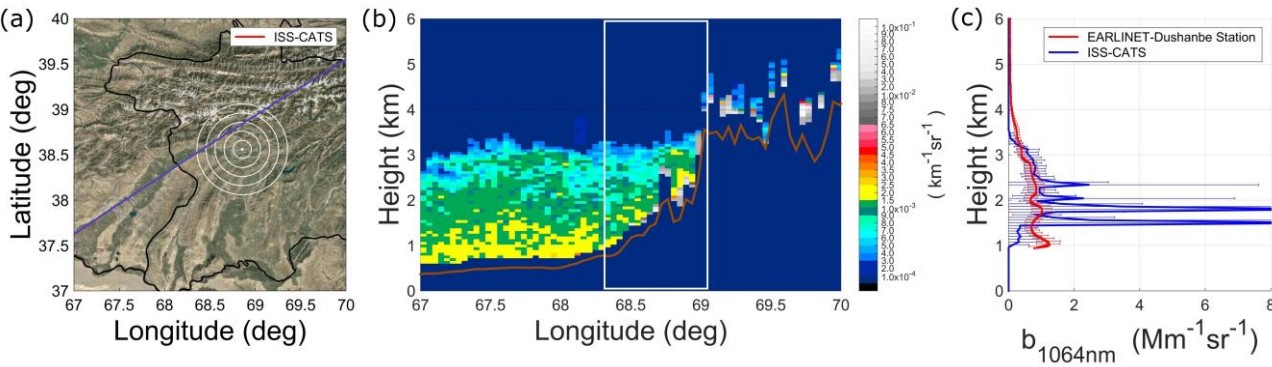

**Figure 5: (a) Nighttime ISS orbit over the EARLINET Dushanbe station on the 25th of May 2015 at 18:53:19 UTC and of closest distance between the footprint of CATS and the EARLINET- Dushanbe station of 24.3km. The white dot denotes the location of Dushanbe lidar system while the blue line shows the lidar footprint of CATS. (b) CATS Backscatter Coefficient at 1064 nm. (c) CATS (blue line) spatially and EARLINET-Dushanbe (red line) temporarily averaged backscatter coefficient profiles (1064 nm). The implemented EARLINET-Dushanbe time window of cloud-free measurements was between 18:00:00 and 20:00:00 UTC. The horizontal blue and red lines denote the variability (one standard deviation) of the CATS and EARLINET measured atmospheric scenes, respectively.**

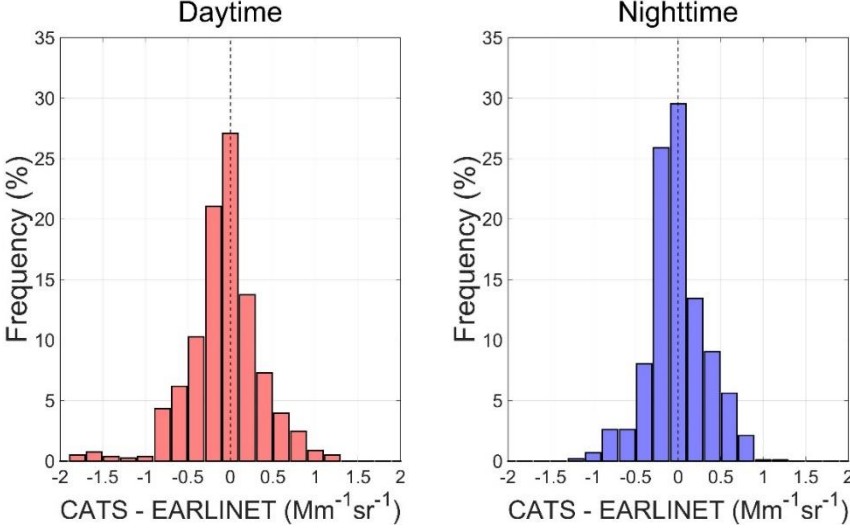

**Figure 6. Distributions of the differences between CATS Level 2 and the corresponding EARLINET backscatter coefficient measurements, calculated over (a) daytime (21 collocated cases) and (b) nighttime (26 collocated cases).**

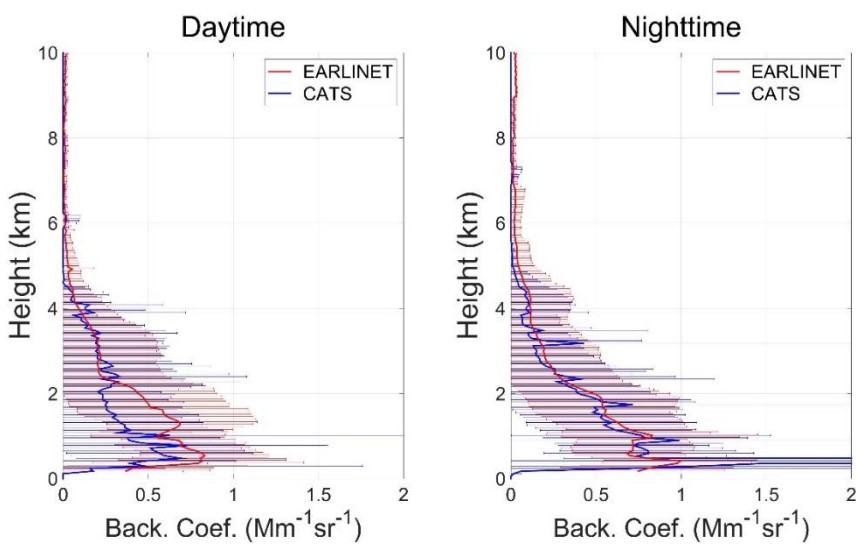

**Figure 7. CATS (blue line) and EARLINET (red line) mean profiles of backscatter coefficient at 1064 nm for (left)**
10    **daytime and (right) nighttime. The horizontal lines represent the SD of CATS (blue colour) and EARLINET (red colour) profiles.**