# Peer review of "EARLINET evaluation of the CATS L2 aerosol backscatter coefficient product"

_Atmospheric Chemistry and Physics, 2019_

## Referee Comment (RC1) · Anonymous Referee #2 · 17 Mar 2019

General comments:

This manuscript compares EARLINET (ground-based) and CATS (onboard the international spatial station) retrievals of the aerosol backscatter coefficient over 12 European sites and 1 Asian site. The paper is well written, however, I did miss some explanation in the introduction about the importance of CATS product. I believe this could be easily achieved by modifying the order of some paragraphs and including extra information. In particular, I suggest moving the second paragraph of Section 2.2 (page 4, line 30 to page 5, line 12) to the introduction, with the due adjustments. I also suggest comparing some scenes of coincident vertical profiles of CATS and CALIOP. Would that be possible? I believe this would dramatically improve the visibility of the paper. Also, it wasn't clear to me whether CATS should only be used to fill a gap in space-based lidar
observations or if it is as reliable as CALIOP. I believe this should be further clarified in the text. I also believe a final paragraph stating the main conclusion is needed (that is, what are your suggestions for future studies: should we use CATS or not, under which conditions these retrievals are reliable, what are their advantages and disadvantages and how could future studies benefit - or not - from CATS).

Specific comments:

page 2, line 3 - Please modify "Physic" to "Physics". page 2, line 20 - Please reformulate the sentence (suggestion: "Quality assessment of CATS..."). page 2, line 24 - Please modify "consists" to "consists of". page 3, line 15 - What is the difference between capacity and capability? page 3, line 16 - Please reformulate or remove the sentence "EARLINET stations are classified as active on condition of...". page 4, line 32 - Please modify "space-borne" to "spaceborne". page 6, line 16 - It's not clear to me if observations more than 90 minutes apart were compared or not. Could you clarify this? page 6, line 24 - What does "including cirrus clouds" mean? Cirrus clouds scenes were used or not? page 7, line 19 - Please modify "participated" to "participating". page 7, line 20 - "exited". Did you mean "excited"? page 9, line 14 - Please modify "in details" to "in detail". page 9, line 24 - Please modify "below" to "of". page 9, line 37 - Please modify "over-lying" to "overlaying". page 11, line 22 - Has the new product already been released? How does the new algorithm differ from the previous one? What kind of improvements does it present? pages 11 and 12, Section 3.2 and Table 2: It would be interesting to show the mean relative bias (that is bias over mean value). page 14, line 10, Please modify "discrepancies" to "discrepancy". page 15, line 1, Please modify "based to" to "based on". Figs. 3 to 5: Please use "b) CATS backscatter coefficient at 1064 nm", or "(1064 nm)". Fig. 5: I would guess topography influence CATS coefficient quite significantly. Could it be causing the spykes shown in this figure? Could you provide a quantitative estimate of the contributing effect of topography on the discrepancy observed in this figure? What about an estimate of the other contributing effects?

---

## Referee Comment (RC2) · Anonymous Referee #1 · 18 Mar 2019

General Comments: In this study the authors are presenting the coordinated effort of the European Aerosol Research Lidar Network (EARLINET), to evaluate the Level 2 aerosol backscatter coefficient product derived by the space borne backscatter lidar namely Cloud-Aerosol Transport System (CATS). The manuscript is well written and has a scientific merit. Therefore, in my opinion it worth being published under the special issue "EARLINET aerosol profiling: contributions to atmospheric and climate research" of the Atmospheric Chemistry and Physics journal. However, in order to help improving the manuscript, I would kindly suggest the authors to take into account the following specific comments. Specific Comments: 1. Abstract: Page 2, line 1: "Independently of daytime/nighttime conditions.". Please consider revising this statement. At the end of this paragraph the authors are mentioning an underestimation of 22.3%

during day and 6.1% during night time. So there is a significant difference in the comparison based on the sky light conditions something that has to be mentioned clearly in the abstract. Where you can attribute this difference? e.g. SNR issue, significance of your day-night statistical sample? 2. Introduction: The Introduction is well written however I am missing the scientific question that this manuscript envisages to answer. Please try to make this clear in this section and consider mentioning the achievements and progress of the scientific community so far towards this topic. Are there any similar activities for CATS? The results presented here are having great difference with similar studies for other space borne lidars? The reader has to reach section 2.1 in order to find some answers on the aforementioned concerns. 3. Section 2.3.1: I think it would be beneficial for the manuscript to include a flowchart showing the methodology of the comparison followed by the authors. The entire process can be summarized there along with the methodology requirements followed by the authors. e.g. the spatial - temporal constraints, cloud screening requirements, etc. The information exists in the manuscript but I feel like it is scattered among the sections. 4. Page 7, lines 18-19: ". . . is less than 30%, . . . requirements of EARLINET". The authors are kindly requested to provide a reference for this statement. 5. Page 8, line 8: "scattering respectively"-> "backscattering respectively" 6. Section 2.3.3: This section is important for following up the manuscript and has to be highlighted. Therefore, I would kindly suggest to the authors to list it as 2.4. 7. Page 9, line 6: "The discussed constraints. . .": How much these constrains affect the final dataset (in terms of number of measurements and overall evaluation)? 8. Page 9, line 18: "here considered"-> "considered here" 9. Page 9, lines 32-33: I cannot understand this conclusive statement. How "the absence of significant biases, both daytime and nighttime" is obvious from figure 3c. 10. Page 10, lines 9-10: "due to the different SNR. . .": I think that indeed this is the case. But this contradicts to the author statement of no significant bias between day and night conditions stated earlier (page 9, lines 32-33). 11. Page 10, lines 24-29: I have the feeling that this information should be moved to section 2.2 where the description of CATS data level product is already given. At that section, the authors can present a

detailed description of their methodology followed for could screening. 12. Page 11, line 23: "end of 2018. . ." -> Maybe "end of 2019" ? 13. Section 3.2: I wonder why the authors constrained their study only to the comparison of aerosol backscatter and they did not proceed with comparison of other aerosol related properties as well (e.g. physical and not properties such as integrated backscatter, AOD, lidar ratio, layer center of mass-thickness). I have the feeling that by taking into account more properties in their comparison will improve the manuscript and will enhance the arguments (i.e. argument of tenuous layer, argument of lidar ratio assumption) for the discrepancies shown here. In addition to that the information provided by each station individually is lost in the analysis demonstrated here. For example, a figure showing the differences between CATS-EARLINET for day and night time conditions per station along with the mean value may explain some of the discrepancies shown in this section (e.g. the argument of topography) or it may reveal other discrepancy patterns if any (i.e. latitudinal). 14. The pair of observation "i" refer to the vertical height of each case study or to each case study individually? This a general comment related to the comparison methodology followed by the authors: I speculate that the initial vertical resolution of the two profiles is not the same. For example, the L1 data products obtained by CATS are within 60 m vertical resolution (Yorks et al., 2011). On the other hand, the data products obtained by EARLINET (especially the Raman retrievals) are processed (application of low-pass filter on the signal) leading to range-resolution loss. A concept of effective resolution is already discussed in the literature (e.g. Iarlori et al., 2015). Therefore, it is not so clear to the reader how the authors managed to compare values obtained from different atmospheric heights? Did they interpolate their values or they used mean values in specific vertical height windows? In any case the authors are kindly suggested to comment their approach on this. (Iarlori, M., Madonna, F., Rizi, V., Trickl, T., Amodeo, A., Effective resolution concepts for lidar observations, Atmos. Meas. Tech., 8, 5157–5176, 2015 www.atmos-meas-tech.net/8/5157/2015/ doi:10.5194/amt-8-5157-2015) 15. Page 13, line 30: "CALIOP" -> Maybe "CATS" instead of CALIOP? 16. Page 15, line 18, lines 24-25: "slight underestimations of the

total AOD in climatic studies." "results in large AOD biases and unrealistic AOD values." I agree with these statements. However, in the current state of the manuscript there is no straight forward comparison of AOD but only backscatter coefficient. See also my previous specific comment No. 14. 17. Page 29, line 13: "The white circle" -> "The white dot denotes the location". The white circle refers to points at various distances from the lidar station as stated by the authors in Figure 2. Please consider correcting this minor typo in figures 3, 4, and 5. 18. Figure 7: For the night time mean profiles the discrepancies are negligible but for the day time and specifically for the height region from 1-2 km large differences are observed. What is the main reason behind this? The significant influence of the topography? In that case why this difference is not shown also in the nigh-time profiles, considering this as a bias from one or more stations. The low daytime CATS SNR? In that case I would expect to see higher discrepancies than sown inside the PBL (longer atmospheric path), compared to 1-2 km. The calibration region of CATS? In any case, I think that a solid and quantitative explanation on this is missing.

―――――――――――――――――――

---

## Referee Report (RR1)

General Comments:

This manuscript evaluates the Level 2 aerosol backscatter coefficient retrieved by the spaceborne backscatter lidar CATS (Cloud-Aerosol Transport System) using collocated ground-based measurements from 14 EARLINET (European Aerosol Research Lidar Network) stations. The manuscript is well written and its contribution to the scientific aerosol community is valuable. I believe that the paper is adequate for publication under the special issue "EARLINET aerosol profiling: contributions to atmospheric and climate research" of the Atmospheric Chemistry and Physics journal after minor revision.

Specific Comments:

Page 2, Line 5: "…underestimations of the total Aerosol Optical Depth (AOD)". Please reframe this sentence. The way it is currently written it gives the impression that the AOD exploration is part of this study.

Page 2, Line 29: "CATS retrievals…..complementarily used". This sentence is incomplete as written. Please reframe.

Page 4, Line 3: Could the authors specify the reason behind choosing the aerosol backscatter at 1064 nm as the only parameter in the comparison? As explained in Section 2.1, Mode 2 gives the opportunity to include the aerosol backscatter coefficient at 532 nm in the comparison which would have enabled a better evaluation of CATS products spectrally and would have also enabled an error estimation for secondary derived lidar parameters such as the backscatter-related Ångström exponent.

Page 4, Line 30: "CATS was a….to three years". Difficult to read sentence. Please rephrase.

Page 35, Line 35: "CATS products and….of processing". Please rephrase the sentence. The part "…and provided in …" is not in the correct tense or it is not a continuation of the previous text.

Page 5, Line 1: What is the error in CATS aerosol backscatter retrievals?

Page 7, Line 4: Hohenpeissenberg site is not listed here. Please add it.

Page 7, Line 22: "widow" → window

Page 8, Line 11-13: The authors used two different processing algorithms for the retrieval of the ground-based aerosol backscatters namely the SCC and PollyXT specified retrieval algorithms. Under the SCC, all measurements could have been processed/treated in the same way. Could you comment on this decision not to process all measurements in the same way and whether these two algorithms can introduce discrepancies in the reported CATS comparison?

Page 9, Line 28: CATS has an overpass over Athens-NTUA at the same day even closer to the measurement site than Athens-NOA but at different time frame (a bit later). As the authors explain, the atmospheric conditions were rather stable at that day. To authors' discretion, I find it valuable/informative if the profiles from that station would be added to Figure 2d and discuss further on the possible differences or similarities.

Page 10, Line 30. I suggest to put each of the cases into a different section giving a short title indicating the complexity of the example.

Page 12, Line 1: I suggest to report the numbers with the same accuracy, for example the numbers at the end of this sentence are inconsistent. Similarly to Table 4.

Page 14, Line 10. To my understanding, the authors mention at Page 10, lines 19-21 that cases where CATS backscatter coefficient is zero or it is at its minimum detection limit have been eliminated from the study yet they are present in this figure for altitudes higher than 6km. Could you clarify?

Page 16, Line 4: The authors have used the Level 2 v2.01 for the evaluation. Nonetheless the latest available version is the v3.01. How this versioning is going to change the associations reported here? Could you correct the versioning to the latest available as in here and line 1 at page 17, for consistency?

Page 17, Line, 11: "52o" → "52°"

Page 17, Line 20: "explotations" → "explotation"

Page 34, Line 2: Please include a complete explanation of the figure, e.g the time frame of the ground-based lidar retrieval and the overpass of CATS along with explanation on the error bar in Figure c. The same applies to the rest of the cases (Figures 4 and 5).

Page 34, Line 12: In Figure 5a the overpass, although nighttime, it is colored as red. To my understanding nighttime overpasses are blue colored. Please ignore this comment if not relevant.

---

## Author Response (AR2)

**General Comments: This manuscript evaluates the Level 2 aerosol backscatter coefficient retrieved by the spaceborne backscatter lidar CATS (Cloud-Aerosol Transport System) using collocated ground-based measurements from 14 EARLINET (European Aerosol Research Lidar Network) stations. The manuscript is well written and its contribution to the scientific aerosol community is valuable. I believe that the paper is adequate for publication under the special issue "EARLINET aerosol profiling: contributions to atmospheric and climate research" of the Atmospheric Chemistry and Physics journal after minor revision.**

The authors would like to thank the reviewer for the interesting and at the same time substantial comments and suggestions. We tried, and did our best, to incorporate the proposed changes and corrections in the revised manuscript, aiming at improving the presented paper. Following, you will find our responses, one by one to the comments addressed.
Kind regards,
Emmanouil Proestakis

**Specific Comments:**
**Page 2, Line 5: "…underestimations of the total Aerosol Optical Depth (AOD)". Please reframe this sentence. The way it is currently written it gives the impression that the AOD exploration is part of this study.**

The authors agree with the reviewer regarding CATS AOD at 1064nm. CATS AOD at 1064 nm has been investigated by a significant number of research groups, towards the assessment of CATS performance (e.g. Rajapakshe et al., 2017; Lee et al., 2018; Noel et al., 2018), with the findings of the aforementioned studies to the manuscript. To be more specific, CATS performance has been validated against ground-based AERONET (Holben et al., 1998) measurements and evaluated against satellite-based AOD retrievals of MODIS (Levy et al., 2013) Aqua and Terra and active CPL (McGill et al., 2002) and CALIPSO CALIOP (Winker et al., 2009) profiles of extinction coefficient and AOD at 1064 nm. Lee et al. (2018) based on AERONET AOD (1020 nm) found a reasonable agreement with CATS AOD with a correlation of 0.64. A comparative analysis of CATS and MODIS C6.1 DT AOD retrievals reported correlation of 0.75 and slope of 0.79, over ocean. In addition, Lee et al., (2019) evaluated AOD and extinction coefficient profiles from CATS through intercomparison with CALIOP and showed correlation of 0.62 and 0.52 over land and ocean respectively during daytime, and 0.84 and 0.81 over land and ocean respectively during nighttime. Comparison of CATS and CALIOP collocated extinction coefficient profiles shows also good shape agreement. Rajapakshe et al. (2017) reported on similar geographical patterns regarding Above Cloud Aerosols and Cloud Fraction between CATS and CALIOP retrievals. Furthermore, CATS retrievals were used to document the diurnal cycle and variations of clouds. Noel et al. (2018) showed that both CATS and CALIOP profiles of Clouds agree well, with minor differences of the order of 2-7% throughout the entire profiles. In addition, CATS depolarization measurements were investigated in the case of desert dust, smoke from biomass burning and cirrus clouds (Yorks et al., 2016), and were found consistent and in good agreement with depolarization measurements from previous studies and historical datasets implementing CPL (Yorks et al., 2011) and CALIOP (Liu et al., 2015). The studies report in general on the good performance of

CATS, despite apparent underestimations. Similarly, to the aforementioned studies, the present study also reports on the good performance of CATS, especially during nighttime, despite underestimations in the backscatter coefficients. The performed studies, not only the present study, report on CATS underestimations, starting from CATS L1 to CATS L3 data, for this reason the sentence includes the phrase: "CATS low negative biases, partially attributed to the deficiency of lidar systems to detect tenuous aerosol layers of backscatter signal below the minimum detection thresholds, *may* lead to systematic deviations and slight underestimations of the total Aerosol Optical Depth (AOD) in climate studies.".

**Page 3, Line 29: "CATS retrievals…..complementarily used". This sentence is incomplete as written. Please reframe.**

According to the reviewer's recommendation, the commented sentences of the manuscript were modified from:
"*CATS retrievals were used to document the diurnal cycle and variations of clouds, with CALIOP complementarily used. Noel et al. (2018) showed that both CATS and CALIOP profiles of CF agree well on both the vertical patterns and values at 01:30 and 13:30 LT, over both land and ocean, with minor differences of the order of 2-7% throughout the entire profiles of cloud fraction*"
to:
"*Noel et al. (2018), implemented measurements from CATS to investigate the diurnal cycle and variations of clouds over land and ocean. The authors showed that both CATS and CALIOP profiles and CF agree well on both the vertical patterns and values at 01:30 and 13:30 LT, over both land and ocean, with minor differences of the order of 2-7% throughout the entire cloud profiles*".

**Page 4, Line 3: Could the authors specify the reason behind choosing the aerosol backscatter at 1064 nm as the only parameter in the comparison? As explained in Section 2.1, Mode 2 gives the opportunity to include the aerosol backscatter coefficient at 532 nm in the comparison which would have enabled a better evaluation of CATS products spectrally and would have also enabled an error estimation for secondary derived lidar parameters such as the backscatter-related Ångström exponent.**

The reviewer is right regarding the potential value of the Mode 7.2 532nm product. However as mentioned in the CATS "Data Release Notes" and the CATS "Algorithm Theoretical Basis Document" (ATBD), unlike the M7.1 data, where the 532 and 1064 nm signals are comparable, the M7.2 532 and 1064 nm signals are very different. Mode 7.2 data at 532 nm is noisy due to issues with stabilizing the seeded laser (laser 2). Since the frequency stability is poor on laser 2, the laser is not aligned properly with the CATS etalon, causing very weak signal transmission. Therefore, it is highly recommended by the NASA CATS team not to use the M7.2 532 nm data for any application, especially for daytime. On the contrary, the use of the 1064 nm data is recommended, though only for studies that are wavelength-independent (i.e. layer detection, relative backscatter intensity).
(https://cats.gsfc.nasa.gov/media/docs/CATS_Release_Notes7.pdf)

**Page 4, Line 30: "CATS was a….to three years". Difficult to read sentence. Please rephrase.**

According to the reviewer's recommendation, the commented sentences of the manuscript were modified from:
"*CATS was a technology demonstration designed to operate on-orbit for a minimum of six months and up to three years*"

to:
"*CATS was a technology demonstration designed to operate on-orbit between six months and three years*".

**Page 4, Line 35: "CATS products and….of processing". Please rephrase the sentence. The part "…and provided in …" is not in the correct tense or it is not a continuation of the previous text.**

According to the reviewer's recommendation, the commented sentences of the manuscript were modified from:
"*CATS products and processing algorithms (Pauly et al., 2019) rely heavily on the processing algorithms developed in the framework of the CPL, ACATS and CALIPSO lidar systems (Palm et al., 2002; Yorks et al., 2011; Hlavka et al., 2012) and provided in different levels of processing.*"
to:
"*CATS processing algorithms (Pauly et al., 2019) rely heavily on the processing algorithms developed in the framework of the CPL, ACATS and CALIPSO lidar systems (Palm et al., 2002; Yorks et al., 2011; Hlavka et al., 2012), while CATS products are provided in different levels of processing*".

**Page 5, Line 1: What is the error in CATS aerosol backscatter retrievals?**

The primary sources of uncertainties in the CATS attenuated backscatter signal are the calibration constant and signal noise. Thus if the calibration constant is accurate, the CATS The source of the systematic error in the CATS ATB is the uncertainty in the calibration constant and is estimated at 5-10% for 1064 nm at night (10-20% for daytime data). The random error in the ATB is dominated by noise in the lidar signal. The total uncertainty, sum of the systematic and random errors, in the CATS ATB 1064 nm is estimated at 10-20% for nighttime data and 20-30% for daytime data.

**Page 7, Line 4: Hohenpeissenberg site is not listed here. Please add it.**

The authors would like the reviewer for observing that Hohenpeissenberg EARLINET station was not included in the list. The list is updated to include Hohenpeissenberg station.

**Page 7, Line 22: "widow" -> window**

The text is corrected according to the reviewer's comment.

**Page 8, Line 11-13: The authors used two different processing algorithms for the retrieval of the ground-based aerosol backscatters namely the SCC and PollyXT specified retrieval algorithms. Under the SCC, all measurements could have been processed/treated in the same way. Could you comment on this decision not to process all measurements in the same way and whether these two algorithms can introduce discrepancies in the reported CATS comparison?**

Indeed, in case of Polly$^{XT}$ systems, an algorithm designed specifically for these systems is used for the retrieval of aerosol backscatter coefficient profiles. The reason why we decided not to process these data with the Single Calculus Chain (SCC) is that up to now, the SCC is able to support very efficiently the processing of lidar signals for systems employing two different acquisition receivers (i.e. one in analogue and one in photon counting mode or both in photon counting mode) for the acquisition of far range and near range signals, but only one receiving telescope. However, it does not yet support the simultaneous processing of lidar signals in

case of systems employing two different telescopes for the acquisition of far range and near range signals (i.e. Polly[XT] systems). Therefore, for Polly[XT] systems in the SCC, the only way in order to correct for the incomplete overlap effect at lower altitudes would be to use an overlap function profile. On the contrary, the Polly[XT] specified retrieval algorithm enables the processing of the two signals (collected from far and near range telescopes) simultaneously, for this reason we decided not ignore the useful information collected by these two different receiving units and use this algorithm instead. The aforementioned algorithm has been excessively studied in the past (Baars et al., 2016) and proven to provide accurate results in an efficient manner. We thus believe that in this way the comparison of Polly[XT] profiles with profiles from other lidar systems, processed with SCC is more accurate since for the former an assumption of the overlap profile is not needed.

**Page 9, Line 28: CATS has an overpass over Athens-NTUA at the same day even closer to the measurement site than Athens-NOA but at different time frame (a bit later). As the authors explain, the atmospheric conditions were rather stable at that day. To authors' discretion, I find it valuable/informative if the profiles from that station would be added to Figure 2d and discuss further on the possible differences or similarities.**

According to the reviewer's observation and comment, Figure 2d was adapted in order to include not only the CATS aerosol profile (denoted by blue line) and the observations of the EARLINET-NOA (denoted by red line) station but in addition the aerosol profile of the EARLINET-NTUA station (denoted by black line). The different of the EARLINET-NOA and EARLINET-NTUA profiles are related to the significant different elevation of the two stations (212m and 95m respectively), to the differences of the two systems, including the implementation of a near field at the Polly[XT] lidar system of the National Observatory of Athens, in contrast to the NTUA system.

From:                                    To:

[revised manuscript text omitted]

**Page 10, Line 30. I suggest to put each of the cases into a different section giving a short title indicating the complexity of the example.**

According to reviewer's recommendation, the suggested part of the manuscript is modified to include the following headers, indicating the different EARLINET-CATS correlative cases:
3.1.1 Case I: ISS-CATS over Leipzig - 13/09/2016 03:37 UTC
3.1.2 Case II: ISS-CATS over Évora - 31/05/2016 19:43 UTC
3.1.3 Case III: ISS-CATS over Dushanbe - 25/05/2015 18:53 UTC

**Page 12, Line 1: I suggest to report the numbers with the same accuracy, for example the numbers at the end of this sentence are inconsistent. Similarly to Table 4.**

The authors agree with the reviewer, therefore the sentence was modified to include the Minimum Detectable Backscatter (MDB) with the same accuracy, as shown in the paper of Yorks et al. (2016) regarding CATS level 1 processing algorithms and data products, as follows: "*CATS M7.2 Minimum Detectable Backscatter 1064 nm: Night: 5.00E-5 ± 77E-5 $km^{-1}sr^{-1}$ / Day: 1.30E-3 ± 0.24E-3 $km^{-1}sr^{-1}$ - for cirrus clouds; Yorks et al., 2016*".

In addition, and according to the reviewer's comment, Table 4 is corrected, to show numbers with the same accuracy, as follows:

| Metric | Daytime | Nighttime |
|---|---|---|
| Mean Bias [$Mm^{-1}sr^{-1}$] | -0.123 | -0.031 |
| Median Differences [$Mm^{-1}sr^{-1}$] | -0.094 | -0.065 |
| Mean Absolute Bias [$Mm^{-1}sr^{-1}$] | 0.323 | 0.249 |
| Mean Relative Bias [%] | -24.062 | -19.843 |
| SD [$Mm^{-1}sr^{-1}$] | 0.431 | 0.342 |
| (min / max Differences) [$Mm^{-1}sr^{-1}$] | (-1.802 / 1.189) | (-1.348 / 1.149) |
| RMSE [$Mm^{-1}sr^{-1}$] | 0.448 | 0.343 |
| Correlation Coefficient | 0.547 | 0.694 |
| Fractional Bias | -0.773 | -0.676 |
| Fractional Gross Error | 0.999 | 1.061 |
| Number of Cases (#) | 21 | 26 |

**Page 14, Line 10. To my understanding, the authors mention at Page 10, lines 19-21 that cases where CATS backscatter coefficient is zero or it is at its minimum detection limit have been eliminated from the study yet they are present in this figure for altitudes higher than 6km. Could you clarify?**

In the framework of the comparison methodology, cases of EARLINET backscatter coefficient values below the CATS minimum detectable backscatter limit at 1064 nm are not included in the comparison, when the corresponding CATS backscatter coefficient is reported to be zero (Fig. 2d - shaded area i). This constrain is applied to account for very thin detected layers from ground-based Lidar systems with backscatter values below the CATS minimum detection limit due to the low Signal-to-Noise Ratio values (SNR). The discussed constrains are employed because of our basic idea to quantitatively assess the representativeness and accuracy of the detected by CATS aerosol features, while preventing possible contaminations (e.g. presence of clouds) to propagate into the CATS-EARLINET dataset. It is applied in the comparison shown for instance in Figure 6, to address quantitatively the accuracy and representativeness of the satellite-based lidar retrievals and to estimate possible biases in the CATS backscatter coefficient. It is applied in the comparison of CATS against EARLINET, to the implementation of the CATSi-EARLINETi residuals for each pair of observations "i", to be used as statistical indicators of CATS average overestimation or underestimation of the aerosol load, in terms of backscatter coefficient values. In this term, since the analysis if focusing to the possible CATS overestimation/underestimation of the aerosol load, the authors compare cases where aerosols are detected by both EARLINET and CATS, or by at least one of the two systems. The comparison statistics on the efficiency of CATS to detect atmospheric features detected by EARLINET systems refers to the aforementioned discussion. However, the study of the evaluation discussion of the mean aerosol backscatter coefficient profiles at 1064 nm as provided by CATS and EARLINET, without further processing, for both daytime (Fig. 7a) and nighttime (Fig. 7b) lidar observations, to investigate the characteristics, similarities and discrepancies between CATS and EARLINET.

**Page 16, Line 4: The authors have used the Level 2 v2.01 for the evaluation. Nonetheless the latest available version is the v3.01. How this versioning is going to change the associations reported here? Could you correct the versioning to the latest available as in here and line 1 at page 17, for consistency?**

The authors would like to thank the reviewer for this comment, the versioning at page 17 was corrected as should be.

Regarding CATS algorithms and the different versions, CATS V3-00 replaced CATS V2-05 on October 1st, 2018. Accordingly, due to algorithm issues present in CATS V3-00 was shortly after replaced by CATS V3-01. Initially, the changes in CATS Level 1 and Level 2 algorithms corresponding to CATS Version 3-00 data was planned to be the final algorithm release for the CATS project, though observed issues in the CATS products led to the modifications of V3-00 and the release of the V3-01 later on in the beginning of 2019. Since CATS products are provided in different levels of processing, the made changes in the algorithms correspond to both L1B and L2O products.

To be more specific, the changes in the L1B algorithms include:

(1) improvement of the nighttime attenuated total backscatter (ATB) profiles due to improvements in the calibration of CATS, thus improvement also in the daytime ATB profiles, since nighttime ATB is implemented in the calculations of the daytime calibration.

(2) changes to the "Depolarization_Quality_Flag", and

(3) implementation of MERRA-2 Reanalysis data instead of GMAO forecasts, for the meteorology in V3-00 and V3-01.

The changes made in the algorithms of CATS L1B reflect on improvements on CATS L2O products. Though additional changes in CATS L2O algorithms include also:

(1) updates in number of profiles in the L2O datasets

(2) improvements in the calculations of uncertainties in the L2O layer-integrated parameters

(3) changes to the "Depolarization_Quality_Flag"

(4) improvements of the Cloud Aerosol Discrimination (CAD) through the implementation of an additional parameter, namely the "Cloud_350m_Fraction_XXX_FOV", to report of the number of 350 L1B profiles within each 5 km L2O bin of the L2O layer product with attenuated total backscatter values greater than 0.03 $km^{-1}sr^{-1}$, thus atmospheric features of high probability of being a cloud. In addition, the parameter "Num_Profs_Avg_LRatio_XXX_FOV" was added to the L2O Layer data product.

(5) improvements in CATS Feature Type and Feature Type Score variables, but also in the Aerosol Subtype classification (replace of "volcanic" with "UTLS Aerosol") and addition of the parameters "Opaque_Feature_Optical_Depth_1064_XXX_FOV" and "Opaque_Feature_Optical_Depth_Uncertainty_1064_XXX_FOV" in Mode 7.2 L2O datasets.

(6) Updates in the Lidar Ratio (LR) values for cirrus clouds

(7) update of the effective multiple scattering factor for ice clouds values to 0.52.

The above changes in the CATS V3-00 and V3-01 algorithms and the respective products are extensively presented and in-depth discussed in the CATS official website (https://cats.gsfc.nasa.gov/; last visit on: 22/05/2019), in the "Publications" section.

CATS products and processing algorithms are provided in different levels of processing. CATS Level 1B (L1B) data include vertical profiles of total and perpendicular attenuated backscatter signals, range-corrected, calibrated and annotated with ancillary meteorological parameters (McGill et al., 2007; Powell et al., 2009; Vaughan et al., 2010). CATS Level 2 (L2) products provide the vertical distribution of aerosol and cloud properties (depolarization ratio, backscatter and extinction coefficient profiles at 1064 nm – FFOV), with a horizontal and vertical resolution of 5 km and 60 m respectively. In addition, L2 data include geophysical parameters of the identified atmospheric layers (vertical feature mask - feature type, aerosol subtype), the required horizontal averaging and information on the feature type classification confidence (Yorks et al., 2019).

Regarding the way this versioning is change the associations reported in the present manuscript, more information are included in the CATS validation led by NASA GSFC Team, and more specific by Dr. Rebecca Pauly (Science Systems and Applications Inc., Lanham, 20706, United States Science Systems and Applications Inc., Lanham, 20706, United States), member of the CATS Team. The study is already submitted on AMT journal:

"*Pauly, R. M., Yorks, J. E., Hlavka, D. L., McGill, M. J., Amiridis, V., Palm, S. P., Rodier, S. D., Vaughan, M. A., Selmer, P. A., Kupchock, A. W., Baars, H., and Gialitaki, A.: Cloud Aerosol Transport System (CATS) 1064 nm Calibration and Validation, Atmos. Meas. Tech. Discuss., https://doi.org/10.5194/amt-2019-172, in review, 2019*".

This study, similarly to the present study, implements specific lidar systems of EARLINET. However, the study of CATS performance will continue and not end with the present manuscript. In this study, the EARLINET authors in collaboration with the CATS Team evaluate CATS Level 2 Mode 7.2 v2-05 backscatter profiles at 1064nm (Palm et al., 2016). The authors have focused on particulate backscatter coefficients ($km^{-1}sr^{-1}$), since this is the product directly derived from measurements, the sum of the parallel and perpendicular backscatter measurements (i.e., $\beta_{1064nm\_total} = \beta_{1064nm\_parallel} + \beta_{1064\_perpendicular}$).

An already ongoing study includes high collocated analysis on the CATS performance (V3-01) and representativeness, including the issues mentioned by the reviewer. The study is based on high temporally and spatially collocated measurements between airborne FAAM Bae-146 research aircraft and ISS measurements, performed in the framework of the AER-D/ICE-D campaign, over Cape-Verde (Santiago island), on August 6-25, 2015, as introduced by Marenco et al. (2018) – Figure 5.

[Figure]

*Figure 5: B920 flight on August 7ᵗʰ, 2015 over Cape Verde, high collocated with ISS-CATS overpass. Left: B920 flight and ISS footprint (left), and CATS backscatter coefficient 1064nm scene (right).*

**Page 17, Line, 11: "52o" -> "52°"**

The text is corrected according to the reviewer's comment.

**Page 17, Line 20: "explotations" -> "explotation"**

The text is modified according to the reviewer's comment.

**Page 34, Line 2: Please include a complete explanation of the figure, e.g the time frame of the ground-based lidar retrieval and the overpass of CATS along with explanation on the error bar in Figure c. The same applies to the rest of the cases (Figures 4 and 5).**

Figure 3
From:
"*Figure 3: (a) Nighttime ISS orbit over EARLINET Leipzig station on the 13ᵗʰ of September 2016 (blue line). The white dot denotes the location of Leipzig lidar system, (b) CATS Backscatter Coefficient at 1064 nm and (c) CATS (blue line) and EARLINET-Leipzig (red line) backscatter coefficient profiles (1064 nm)*".
to:
*Figure 3: (a) Nighttime ISS orbit over the EARLINET Leipzig station on the 13ᵗʰ of September 2016 at 03:37:49 UTC and of closest distance between the footprint of CATS and the EARLINET-Leipzig station of 3.79km. The white dot denotes the location of Leipzig lidar system while the blue line shows the lidar footprint of CATS. (b) CATS Backscatter Coefficient at 1064 nm. (c) CATS (blue line) spatially and EARLINET-Leipzig (red line) temporarily averaged backscatter coefficient profiles (1064 nm). The implemented EARLINET-Leipzig time window of cloud-free measurements was between 00:00:00 and 02:30:00 UTC. The horizontal blue and red lines denote the variability (one standard deviation) of the CATS and EARLINET measured atmospheric scenes, respectively.*".

Figure 4
From:

*"Figure 4: (a) Daytime ISS orbit over Évora EARLINET station on the 31th of May 2016 (red line). The white dot denotes the location of Évora lidar system, (b) CATS Backscatter Coefficient at 1064 nm and (c) CATS (blue line) and EARLINET-Évora (red line) backscatter coefficient profiles (1064 nm)"*.
to:
*"Figure 4: (a) Daytime ISS orbit over the EARLINET Évora station on the 31th of May 2016 at 19:43:31 UTC and of closest distance between the footprint of CATS and the EARLINET- Évora station of 39.42km. The white dot denotes the location of Évora lidar system while the red line shows the lidar footprint of CATS. (b) CATS Backscatter Coefficient at 1064 nm. (c) CATS (blue line) spatially and EARLINET-Évora (red line) temporarily averaged backscatter coefficient profiles (1064 nm). The implemented EARLINET-Évora time window of cloud-free measurements was between 19:29:56 and 19:59:35 UTC. The horizontal blue and red lines denote the variability (one standard deviation) of the CATS and EARLINET measured atmospheric scenes, respectively."*.

Figure 5
From:
*"Figure 5: (a) Nighttime ISS orbit over Dushanbe EARLINET station on the 25th of May 2015 (blue line). The white dot denotes the location of Dushanbe lidar system, (b) CATS Backscatter Coefficient at 1064 nm and (c) CATS (blue line) and EARLINET-Dushanbe (red line) backscatter coefficient profiles (1064 nm)."*
to:
*"Figure 5: (a) Nighttime ISS orbit over the EARLINET Dushanbe station on the 25th of May 2015 at 18:53:19 UTC and of closest distance between the footprint of CATS and the EARLINET-Dushanbe station of 24.3km. The white dot denotes the location of Dushanbe lidar system while the blue line shows the lidar footprint of CATS. (b) CATS Backscatter Coefficient at 1064 nm. (c) CATS (blue line) spatially and EARLINET-Dushanbe (red line) temporarily averaged backscatter coefficient profiles (1064 nm). The implemented EARLINET-Dushanbe time window of cloud-free measurements was between 18:00:00 and 20:00:00 UTC. The horizontal blue and red lines denote the variability (one standard deviation) of the CATS and EARLINET measured atmospheric scenes, respectively."*.

**Page 34, Line 12: In Figure 5a the overpass, although nighttime, it is colored as red. To my understanding nighttime overpasses are blue colored. Please ignore this comment if not relevant.**

The reviewer is right regarding the Dushanbe station, regarding the time of ISS overpass and the frequently type of color used, at least in CALIPSO CALIOP quicklooks. Although in the case of ISS-CATS orbits are not presented in the same colors, blue and red for nighttime and daytime orbits respectively, we have used the same color here, thus Figure 5a was adapted accordingly, as follows:

[revised manuscript text omitted]

lie below the detection thresholds (CATS M7.2 Minimum Detectable Backscatter 1064 nm: Night: 5.00E-5 ± 77E-5 km⁻¹sr⁻¹ / Day: 1.30E-3 ± 0.24E-3 km⁻¹sr⁻¹ - for cirrus clouds; Yorks et al., 2016). The detection limitation of CATS may propagate in scientific studies implementing CATS through introduced underestimations and possible biases.

The assessment of accuracy of CATS Level 2 against EARLINET collocated and concurrent observations is performed on the basis of backscatter coefficient profiles, because this product constitutes the CATS Level 2 parameter with the lowest influence of a-priori assumptions (e.g. lidar ratio). In addition CATS Level 2 provides the feature classification of the detected layers and associated confidence level of the classification. ~~In addition to the backscatter coefficient, CATS Level 2 data provide the feature classification of the detected layers (
[revised manuscript text omitted]
 space borne backscatter lidar namely Cloud-Aerosol Transport System (CATS). The manuscript is well written and has a scientific merit. Therefore, in my opinion it worth being published under the special issue "EARLINET aerosol profiling: contributions to atmospheric and climate research" of the Atmospheric Chemistry and Physics journal. However, in order to help improving the manuscript, I would kindly suggest the authors to take into account the following specific comments.**

The authors would like to thank the reviewer for the interesting and at the same time substantial comments and suggestions. We tried, and did our best, to incorporate the proposed changes and corrections in the revised manuscript, aiming at improving the presented paper. Following, you will find our responses, one by one to the comments addressed.
Kind regards,
Emmanouil Proestakis

**Specific Comments:**
**1. Abstract: Page 2, line 1: "Independently of daytime/nighttime conditions.". Please consider revising this statement. At the end of this paragraph the authors are mentioning an underestimation of 22.3% during day and 6.1% during night time. So there is a significant difference in the comparison based on the sky light conditions**

**something that has to be mentioned clearly in the abstract. Where you can attribute this difference? e.g. SNR issue, significance of your day-night statistical sample?**

The authors agree with the statement of the reviewer. Therefore, the sentence was modified from:

*"In addition, CATS misclassification of aerosol layers as clouds, and vice versa, in cases of coexistent and/or adjacent aerosol and cloud features, may lead to non-representative, unrealistic and cloud contaminated aerosol profiles. The distributions of backscatter coefficient biases show the relatively good agreement between the CATS and EARLINET measurements, although on average underestimations are observed, 22.3 % during daytime and 6.1 % during nighttime."*

To:

*"In addition, CATS misclassification of aerosol layers as clouds, and vice versa, in cases of coexistent and/or adjacent aerosol and cloud features, may lead to non-representative, unrealistic and cloud contaminated aerosol profiles. Regarding solar illumination conditions, low negative biases in CATS backscatter coefficient profiles, of the order of 6.1%, indicate the good nighttime performance of CATS. During daytime, reduced signal-to-noise ratio by solar background illumination prevents retrievals of weakly scattering atmospheric layers that would otherwise be detectable during nighttime, leading to higher negative biases, of the order of 22.3%, in CATS daytime performance."*

Regarding the comment of the reviewer, where the authors attribute this difference, the effect of SNR is considered the most critical factor, because measurement noise by solar illumination background and layer detection are different during daytime and nighttime, with the effect propagating through the retrieval algorithms to atmospheric layer detection and classifications and eventually to Level 2 and Level 3 products. Example of the critical level of SNR effect is the Minimum Detectable Backscatter (MDB), as reported by McGill et al. (2007), for both CALIOP and CATS and for both daytime and nighttime conditions (Table 1). According to *Table 1* the detection sensitiveness of thin, weakly scattering atmospheric layers at CATS M7.2 1064 nm is two orders of magnitude higher during nighttime that during daytime (MDB two orders of magnitude lower during nighttime than during daytime). In the case of CALIOP, both for 532 and 1064 nm, MDB during nighttime is an order of magnitude lower during nighttime than during daytime.

| Table 1: CATS and CALIPSO 532 and 1064 nm Minimum Detectable Backscatter (MDB) with Units in $Km^{-1}sr^{-1}$ (McGill et al., 2007). | | |
|---|---|---|
| | CATS 7.2 | CALIPSO |
| 532 nm night | $1.6 \times 10^{-2} \pm 0.84 \times 10^{-3}$ | $1.6 \times 10^{-4} \pm 0.84 \times 10^{-4}$ |
| 1064 nm night | $5.0 \times 10^{-5} \pm 0.77 \times 10^{-5}$ | $1.6 \times 10^{-4} \pm 0.84 \times 10^{-4}$ |
| 532 nm day | $3.8 \times 10^{-2} \pm 1.05 \times 10^{-3}$ | $1.7 \times 10^{-3} \pm 0.84 \times 10^{-3}$ |
| 1064 nm day | $1.3 \times 10^{-3} \pm 0.24 \times 10^{-3}$ | $1.0 \times 10^{-3} \pm 0.30 \times 10^{-3}$ |

**2. Introduction: The Introduction is well written however I am missing the scientific question that this manuscript envisages to answer. Please try to make this clear in this section and consider mentioning the achievements and progress of the scientific community so far towards this topic. Are there any similar activities for CATS? The results presented here are having great difference with similar studies for other space borne lidars? The reader has to reach section 2.1 in order to find some answers on the aforementioned concerns.**

The authors agree with the reviewer that the manuscript was characterized by a significant lack of mentioning similar achievements and activities, towards the assessment of CATS performance. The authors agree with the reviewer regarding the necessity of including the findings of the aforementioned studies and have adjusted the manuscript accordingly. To be more specific, the following paragraphs were added to the manuscript (Section 1 - Introduction):

*"CATS performance has been validated against ground-based AErosol RObotic NETwork (AERONET; Holben et al., 1998) measurements and evaluated against satellite-based Atmospheric Optical Depth (AOD) retrievals of Aqua and Terra Moderate Imaging Spectroradiometer (MODIS; Levy et al., 2013) and active CPL (McGill et al., 2002) and CALIPSO CALIOP (Winker et al., 2009) profiles of extinction coefficient and AOD at 1064 nm. Lee et al. (2018) compared daytime quality-assured CATS V2-01 vertically integrated extinction coefficient profiles (1064 nm) and AERONET AOD (1020 nm) values, spatially (within 0.4° Longitude and Latitude) and temporally (±30 minutes)*

*collocated, and found a reasonable agreement with a correlation of 0.64. A comparative analysis of CATS and MODIS C6.1 Dark Target (DT) AOD retrievals, through spectral interpolation between 0.87 and 1.24 μm channels, reported correlation of 0.75 and slope of 0.79, over ocean. In addition, Lee et al., (2019) evaluated AOD and extinction coefficient profiles from CATS through intercomparison with CALIOP. With regard to AOD, analysis a total of 2681 CATS and CALIOP collocated observation cases (within 0.4º Longitude/Latitude and ±30 minutes ISS and CALIPSO overpass difference), showed correlation of 0.62 and 0.52 over land and ocean respectively during daytime (1342 cases), and 0.84 and 0.81 over land and ocean respectively during nighttime (1339 cases). Comparison of CATS and CALIOP collocated extinction coefficient profiles based on the closest Euclidian distance on the earth's surface, shows also good shape agreement, despite an apparent CALIOP underestimation in the lowest 2 km height. CATS and CALIOP observations were used by Rajapakshe et al. (2017) to study the seasonally transported aerosol layers over the SE Atlantic Ocean. The performed comparative analysis reported on similar geographical patterns regarding Above Cloud Aerosols (ACA), Cloud Fraction (CF) and ACA occurrence frequency (ACA_F) between CATS and CALIOP retrievals. However, the authors reported also on differences between CATS and CALIOP vertical aerosol distributions, with ACA bottom height identified by CATS lower than the respective of CALIOP. CATS retrievals were used to document the diurnal cycle and variations of clouds, with CALIOP complementarily used. Noel et al. (2018) showed that both CATS and CALIOP profiles of CF agree well on both the vertical patterns and values at 01:30 and 13:30 LT, over both land and ocean, with minor differences of the order of 2-7% throughout the entire profiles of cloud fraction. CATS depolarization measurements, which are critical in the processing algorithms of aerosol subtype classification, were investigated in the case of desert dust, smoke from biomass burning and cirrus clouds (Yorks et al., 2016), and were found consistent and in good agreement with depolarization measurements from previous studies and historical datasets implementing CPL (Yorks et al., 2011) and CALIOP (Liu et al., 2015).*"*

Regarding the question the manuscript envisages to answer, the author have modified/included the following paragraphs to the manuscript (Section 1 - Introduction):

"*Overall, CATS retrievals have been evaluated and found in reasonable agreement with ground-based AERONET, airborne CPL and satellite-based MODIS and CALIOP measurements. However, for the quality assessment of CATS backscatter coefficient profiles, a large-scale and dense network of ground-based lidar systems is needed, in order to facilitate high-quality collocated and concurrent measurements. This necessity is largely related to the ISS orbital characteristics, the CATS near-nadir viewing (0.5º off nadir), the lidar narrow footprint (14.38 m diameter), and the limited number of ISS overpasses. The European Aerosol Research Lidar Network (EARLINET) consists of a unique infrastructure for assessing the validation needs for spaceborne lidar missions.*".

**3. Section 2.3.1: I think it would be beneficial for the manuscript to include a flowchart showing the methodology of the comparison followed by the authors. The entire process can be summarized there along with the methodology requirements followed by the authors. e.g. the spatial - temporal constraints, cloud screening requirements, etc. The information exists in the manuscript but I feel like it is scattered among the sections.**

The authors agree with the reviewer that is would be beneficial to summarize the key parameters and the associated thresholds implemented in the framework of the study. For this reason the following table was included in the manuscript:

*Table: List of CATS quality assurance thresholds applied in the EARLINET comparison.*

| Mode | 7.2 |
|---|---|
| Level | 2 |
| Parameter | Backscatter Coefficient |
| Wavelength | 1064nm |
| Distance | ≤ 50km radius from the EARLINET stations |
| Feature Type Score | ≤ -2 |
| Sky Condition | 0 – clean skies *and* 1 – clear skies (no clouds) |
| Backscatter Coefficient | $0 \leq b_{1064nm} \leq 2$ [Mm$^{-1}$sr$^{-1}$] |
| Vertical range window | ≤ 10 km (a.s.l.) |

Regarding EARLINET, the authors implement the Single Calculus Chain (SCC) is used D'Amico et al, 2015; D'Amico et al, 2016; Mattis et al., 2016), for the homogenization of the lidar data in a standardized output format. SCC facilitates an automatic algorithm developed to further address the quality assurance of the lidar measurements. The EARLINET implementation is described in "*Section 3.2.3*".

**4. Page 7, lines 18-19: ":::: is less than 30%, ::: requirements of EARLINET". The authors are kindly requested to provide a reference for this statement.**

The text is modified according to the reviewer's recommendation, and the following references were included:
"*The comparison showed that by using only the signal from the elastic channels, the mean relative deviation in the calculation of the aerosol backscatter coefficient at 1064 nm is less than 30 % (Althausen et al., 2009; Baars et al., 2012; Engelmann et al., 2016; Hänel et al., 2012), thus meeting the quality assurance requirements of EARLINET.*"
with the following references:
o *Althausen, D., Engelmann, R., Baars, H., Heese, B., Ansmann, A., Müller, D., and Komppula, M.: Portable Raman Lidar PollyXT for Automated Profiling of Aerosol Backscatter, Extinction, and Depolarization, J. Atmos. Ocean. Technol., 26, 2366–2378, 2009.*
o *Baars, H., Ansmann, A., Althausen, D., Engelmann, R., Heese, B., Müller, D., Artaxo, P., Paixao, M., Pauliquevis, T., and Souza, R.: Aerosol profiling with lidar in Amazon Basin during the wet and dry season, J. Geophys. Res., 117, D21201, 2012.*
o *Engelmann, R., Kanitz, T., Baars, H., Heese, B., Althausen, D., Skupin, A., Wandinger, U., Komppula, M., Stachlewska, I. S., Amiridis, V., Marinou, E., Mattis, I., Linne, H. and Ansmann, A.: The automated multiwavelength Raman polarization and water-vapor lidar Polly(XT): the neXT generation, Atmos. Meas. Tech., 9(4), 1767–1784, doi:10.5194/amt-9-1767-2016, 2016.*
o *Hänel, A., Baars, H., Althausen, D., Ansmann, A., Engelmann, R., and Sun, Y. J.: One-year aerosol profiling with EUCAARI Raman lidar at Shangdianzi GAW station: Beijing plume and seasonal variation, J. Geophys. Res., 117, D13201, 2012.*

**5. Page 8, line 8: "scattering respectively"-> "backscattering respectively".**

The text is modified according to the reviewer's recommendation.

**6. Section 2.3.3: This section is important for following up the manuscript and has to be highlighted. Therefore, I would kindly suggest to the authors to list it as 2.4.**

The text is modified according to the reviewer's recommendation.

**7. Page 9, line 6: "The discussed constraints:::": How much these constrains affect the final dataset (in terms of number of measurements and overall evaluation)?**

Regarding the question of the reviewer on the discussed constrains on the dataset, Figures 1-4 show quantitatively the effects of (i) distance between the EARLINET station and the closest profile of the CATS-ISS overpass for each correlative case, (ii) CATS Feature Type, (iii) number of CATS Level 2 (L2) Aerosol Profiles (APro) used in the CATS horizontal average, and the effect of (iv) topography of EARLINET stations. The comparison exercise examines the effect of one discussed constrain at a time, while keeping all the other parameters in the methodology constant, and considers various evaluation metrics, as discussed in the following sections.

*(i) Effect of distance between the EARLINET station and the closest profile of the CATS-ISS overpass*

Figure 1 shows the effect of distance between the closest CATS L2 APro and the respective EARLINET station matchup, for different upper Euclidean distance thresholds (i.e.: 5n km, n∈N={1,10}). To be more specific, the Mean Bias (MB; [Mm$^{-1}$sr$^{-1}$]) - (Fig.1a), Root Mean Square Error (RMSE; [Mm$^{-1}$sr$^{-1}$]) - (Fig.1b), Correlation Coefficient (Fig.1c), and the number of CATS-EARLINET correlative cases per each upper distance threshold are considered. For each upper distance threshold, all the available CATS-EARLINET cases of Euclidean distance lower or equal to the

respective upper limit are considered in the computation of the aforementioned evaluation metrics. This cumulative approach is selected due to the limited number of CATS-EARLINET correlative cases, and is applied separately for daytime and nighttime ISS overpasses, due to the different CATS measurement conditions.

Based on the analysis, during nighttime (daytime), the CATS-EARLINET MB is increasing (decreasing) starting from the 5 km upper distance threshold, to reach -0.0300 (-0.123) $Mm^{-1}sr^{-1}$, for the radius threshold of 50km shown in the study. The computed RMSE values are in the range between 0.447 and 0.343 $Mm^{-1}sr^{-1}$ for nighttime and between 0.357 and 0.448 $Mm^{-1}sr^{-1}$ for daytime, for the distance thresholds of 5km and 50km respectively. The minimum RMSE values are observed when considering ISS overpass cases of closer than 40 km distance to the EARLINET stations during nighttime, corresponding to MB of 0.018 $Mm^{-1}sr^{-1}$. The Correlation Coefficient is decreasing with increasing distance between the ISS overpass and the EARLINET stations. Notably, the Correlation Coefficient is not changing considerably for thresholds between 15 and 40 km for nighttime (~0.8) and between 15 and 30 km for daytime (~ 0.7). Sharp decreases in the Correlation Coefficient are observed during daytime (0.547), for distances closer to the EARLINET stations than during nighttime (0.693), for 35 and 40 km distance respectively. The observed tendencies can be explained in terms of the distance thresholds and number of available cases, since the distance thresholds define the number of cases that are used in the analysis and the number of case is critical to assess the performance of CATS. Consequently, the MB, RMSE and Correlation Coefficient are all subject to both the number and the characteristics of the CATS-EARLINET cases used. In the study the authors use the maximum number of available EARLINET cases, to avoid any possible selection effect resulting from a poor sample of correlative cases, when strict collocation filters are applied. Using the maximum number of available correlative cases, i.e. twenty six (26) and twenty one (21) for nighttime and daytime respectively, for ISS overpasses within 50km radius from the EARLINET stations, the authors envisage to quantitatively address the question of CATS performance and the representativeness of the aerosol backscatter coefficient profiles, over various atmospheric, illumination and ISS overpass conditions.

[Figure]

*Figure 1: CATS backscatter coefficient at 1064nm with respect to EARLINET ground-based measurements, as a function of distance (km) between the closest CATS Level 2 Aerosol Profile and the respective "collocated" EARLINET station, for daytime (red line) and nighttime (blue line) ISS overpasses. Left: Mean Bias [$Mm^{-1}sr^{-1}$], center: RMSE [$Mm^{-1}sr^{-1}$] and right: Correlation Coefficient. Dashed lines correspond to the number of CATS-EARLINET correlative cases considered per each upper distance threshold between the CATS footprint and the locations of EARLINET stations.*

*(ii) Effect of Feature Type Score*

The main objective of the CATS Cloud Aerosol Discrimination (CAD) score, or Feature Type Score, is to provide to the Feature Type classification a level of confidence. In the case of CATS, the Feature Type score is an integer number ranging between -10 and 10. The values of CATS Feature Type score correspond to classified aerosol atmospheric layers (negative values) and cloud atmospheric layers (positive values), while the magnitude of the Feature Type score corresponds to the confidence level of the classification. A value of -10 indicates complete confidence that the layer is an aerosol layer, while Feature Type score equal to 0, indicates an atmospherics layer with equal probability to be cloud or aerosol.

Figure 2 shows the effect of Feature Type Score, for different values, between -8 and 0 (i.e. for atmospheric layers classified as aerosol layers). The Mean Bias (MB; [$Mm^{-1}sr^{-1}$]) - (Fig.2a), Root Mean Square Error (RMSE; [$Mm^{-1}sr^{-1}$]) - (Fig.2b) and Correlation Coefficient (Fig.2c) are shown per each Feature Type Score. For each Feature Type score, cases of lower classification confidence level are not considered in the assessment of CATS performance and representativity, indicating the effect of the selected Feature Type thresholds.

Based on the MB, RMSE and Correlation Coefficient, a similar tendency is observed for different Feature Type Scores. To be more specific, not considerable changes are observed for different Feature Type Scores, regardless of the selected Feature Type threshold. This effect is due to the atmospheric characteristics of the CATS-EARLINET cases considered in the analysis. In the framework of the study, to account for contamination effects of multiple-scattering and specular reflection in the intercomparison process, only cloud-free atmospheric scenes are used. Furthermore, cases with detected cirrus, either at the EARLINET Range-Corrected-Signal quicklooks or at the ISS-CATS backscatter coefficient profiles or the feature type profiles, are not considered in the study. Initially, the presence of clouds was investigated through the implementation of CATS backscatter coefficient and depolarization time-height images and EARLINET range-corrected-signal. Cases for which the retrieval of EARLINET temporally-averaged profile was not feasible due to the presence of clouds, and/or CATS cases that the presence of clouds propagated into the CATS spatial-averaged profile were discarded from the analysis. Consequently, the lack of dependence shown in Figure 2 (a-c) is the result from the a priory selection of cloud free conditions selected in the analysis. However, a notably characteristic is the nighttime performance of CATS, which as shown from the lower absolute MB and lower RMSE, but in addition from the higher Correlation Coefficient values, due to higher SNR, is more representative than the corresponding daytime performance.

[Figure]

*Figure 2: CATS backscatter coefficient at 1064nm with respect to EARLINET ground-based measurements, as a function of Feature Type score, for daytime (red line) and nighttime (blue line) ISS overpasses. Left: Mean Bias [Mm$^{-1}$sr$^{-1}$], center: RMSE [Mm$^{-1}$sr$^{-1}$] and right: Correlation Coefficient.*

*(iii) Effect of number of CATS-ISS L2 aerosol profiles used in the spatial averaging*

Similarly to the analysis presented and discussed above, Figure 3 shows the effect of different number of aerosol profiles used when spatially averaging to retrieve the CATS aerosol profiles used in the framework of the study. In Figure 3, the acronym "CPro" corresponds to the closest CATS profiles to the corresponding EARLINET station. Accordingly, the Mean Bias (MB; [Mm$^{-1}$sr$^{-1}$]) - (Fig.3a), Root Mean Square Error (RMSE; [Mm$^{-1}$sr$^{-1}$]) - (Fig.3b), Correlation Coefficient (Fig.3c), are computed for different number of profiles used (i.e. CPro±1Profile, CPro±2Profiles, …).

Based on the MB, RMSE and Correlation Coefficient, the representativeness of CATS spatial profile is increasing with increasing number of aerosol profiles used in the horizontal averaging. To be more specific nighttime MB is almost constant, showing a low dependence on the number of profiles used, while for daytime CATS cases the opposite effect is observed, with improvement of CATS performance though increasing number of profiles used. Regarding RMSE no significant changes are observed, though a slight decreasing tendency in the RMSE is observed for both daytime and nighttime cases. Regarding the Correlation Coefficient, increasing in the values is also observed, with increasing number of profiles used, both for daytime and nighttime cases, denoting the improvement of the representativeness with increasing number of CATS profiles used in the spatial averaging.

[Figure]

*(iv) Effect of EARLINET stations topography*

In order to study the effect of topography on the CATS profiles the authors separated the participating EARLINET stations into 3 clusters: Continental (Case I – Belsk, Bucharest, Leipzig, and Warsaw), Coastal (Case II – NOA, Athens NTUA, Barcelona, Cabauw, Thessaloniki and Lecce) and Mountainous (Case III – Dushanbe, Evora, Observatory Hohenpeissenberg, Potenza). The three clusters and the characteristics of the stations are given in Table 2. In addition, Figure 4 shows the locations of the participating stations; green circles denote Continental stations, blue circles denote Coastal stations and brown circles denote Mountainous stations. Figure 4 shows, additionally to the geographical distribution of the active EARLINET stations, the daytime/nighttime overpasses of ISS within the evaluation period, between 02/2015 and 09/2016, encompassing the first twenty months of CATS operation. Due to the limited available dataset of CATS-EARLINET cases, the daytime/nighttime approach was not followed in the case of the analysis regarding the effect of topography.

*Table 2: Clustering of EARLINET stations with respect to topographical features.*

| Case I - Continental | | | | |
|---|---|---|---|---|
| EARLINET Station | Identification Code | Latitude ($^{\circ}$N) | Longitude ($^{\circ}$E) | Altitude a.s.l. (m) |
| Belsk | be | 51.83 | 20.78 | 180 |
| Bucharest | bu | 44.35 | 26.03 | 93 |
| Leipzig | le | 51.35 | 12.43 | 90 |
| Warsaw | wa | 52.21 | 20.98 | 112 |

| Case II - Coastal | | | | |
|---|---|---|---|---|
| EARLINET Station | Identification Code | Latitude ($^{\circ}$N) | Longitude ($^{\circ}$E) | Altitude a.s.l. (m) |
| Athens-NOA | no | 37.97 | 23.72 | 86 |
| Athens-NTUA | at | 37.96 | 23.78 | 212 |
| Barcelona | ba | 41.39 | 2.12 | 115 |
| Cabauw | ca | 51.97 | 4.93 | 0 |
| Thessaloniki | th | 40.63 | 22.95 | 50 |
| Lecce | lc | 40.33 | 18.10 | 30 |

| Case III - Mountainous | | | | |
|---|---|---|---|---|
| EARLINET Station | Identification Code | Latitude ($^{\circ}$N) | Longitude ($^{\circ}$E) | Altitude a.s.l. (m) |
| Dushanbe | du | 38.56 | 68.86 | 864 |
| Évora | ev | 38.57 | -7.91 | 293 |
| Observatory Hohenpeissenberg | oh | 47.8 | 11.01 | 974 |
| Potenza | po | 40.60 | 15.72 | 760 |

[Figure]

*Figure 4: Distribution of EARLINET lidar stations over Europe and West Asia. Green dots: Continental stations used in the inter-comparison. Blue dots: Coastal stations used in the inter-comparison. Brown dots: Mountainous stations used in the inter-comparison. ISS orbits between 02/2015 and 09/2016 are overlaid in red for daytime and in blue for nighttime overpasses.*

Figure 5 shows the effect of Topography, for three different clusters of station characteristics, as introduced above (Case I: Continental, Case II: Coastal and Case III: Mountainous). In Figure 5a, the Box and Whisker plot on the CATS$_i$-EARLINET$_i$ residuals is shown, including the lower and upper whiskers which indicate the 10$^{th}$ and 90$^{th}$ percentiles respectively, and the 25$^{th}$ and the 75$^{th}$ quantiles indicated by the lower and upper box boundaries respectively. The horizontal line and the red dot indicate the statistical mean and median values respectively while outliers are indicated by red crosses. According to the results, it is evident that the correlative measurements between the Mountainous EARLINET stations and the ISS overpasses are characterized by higher variability, more extreme differences, higher absolute mean and median biases and higher RMSE than in the Continental and Maritime cases. Complex topography, in terms of geographical characteristics, erroneous mean backscatter coefficient profiles due to the high variability of aerosol load in the Planetary Boundary Layer, the horizontal distance between the CATS lidar footprint and the ground-based lidar stations and surface returns enhance the discrepancies, especially in the lowermost part of the profiles, resulting in higher differences between the EARLINET profiles and CATS profiles. Due to the lack of the aforementioned effects arising from complex topography, CATS representativeness and performance is higher over the Continental cases, while CATS performance over the Coastal stations is characterized by slightly lower absolute value of mean bias and at the same time by lower Correlation Coefficient than in the case of Continental cases. However, it has to be taken into consideration the important factor related to the presented results that is the number of CATS-EARLIENT correlative cases used in the analysis, 23 for Case I - Continental, 10 for Case II - Coastal and 14 for Case III - Mountainous. Analytical evaluation metrics on the effect of topography are given in Table 3.

[Figure]

*Figure 5: CATS backscatter coefficient at 1064nm with respect to EARLINET ground-based measurements, as a function of different topography of EARLINET stations for three different clusters of station topographical characteristics (Case I: Continental, Case II: Coastal and Case III: Mountainous). In Fig.5a, the Box and Whisker plot on the CATS$_i$-EARLINET$_i$ residuals is shown, including the lower and upper whiskers which indicate the 10$^{th}$ and 90$^{th}$ percentiles respectively, and the 25$^{th}$ and the 75$^{th}$*

*quantiles indicated by the lower and upper box boundaries respectively. The horizontal line and the red dot indicate the statistical mean and median values respectively while outliers are indicated by red crosses. Fig.5b and Fig.5c show the RMSE and Correlation Coefficient as a function of the different clusters, including the number of available cases per cluster.*

*Table 3: Clusters of EARLINET stations and CATS evaluation metrics.*

| | Continental stations | Coastal stations | Mountainous stations |
|---|---|---|---|
| Median | -0.053 [Mm$^{-1}$sr$^{-1}$] | -0.076 [Mm$^{-1}$sr$^{-1}$] | -0.106 [Mm$^{-1}$sr$^{-1}$] |
| Mean | -0.016 [Mm$^{-1}$sr$^{-1}$] | -0.058 [Mm$^{-1}$sr$^{-1}$] | -0.151[Mm$^{-1}$sr$^{-1}$] |
| RMSE | 0.367 [Mm$^{-1}$sr$^{-1}$] | 0.293 [Mm$^{-1}$sr$^{-1}$] | 0.434 [Mm$^{-1}$sr$^{-1}$] |
| Correlation Coefficient | 0.673 | 0.499 | 0.591 |
| Number of cases | 23 | 10 | 14 |

**8. Page 9, line 18: "here considered"-> "considered here".**

The text is modified according to the reviewer's recommendation.

**9. Page 9, lines 32-33: I cannot understand this conclusive statement. How "the absence of significant biases, both daytime and nighttime" is obvious from figure 3c.**

The reviewer is right, that Figure 3c corresponds to a nighttime atmospheric scene, therefore the statement, referring not only to nighttime but also to daytime conclusions, may be confusing for the reader. The authors, have inspected of all available cases one-by-one, and wanted to provide the information through this section, that when the atmospheric scene is homogeneous and the scattering characteristics of the aerosol layers are above the MDB thresholds of CATS sensor (i.e. sufficient SNR for detection and classification), the overall CATS performance under such homogeneous conditions is good, with absence of significant biases. This conclusion holds both for daytime and nighttime. For this reason the "*representative case*" was used.

However, since the authors agree with the reviewer that the sentence may be confusing, the sentence was reformulated from:

*"The intercomparison presented in Figure 3c is a representative case, indicating the overall high performance of CATS and the absence of significant biases, during both daytime and nighttime, under relative homogeneous and cloud free conditions."*

to:

*"Although the case presented and discussed in Figure 3 corresponds to a nighttime ISS overpass, the case is representative for cloud free and relative homogeneous atmospheric scenes in terms of aerosols, for both daytime and nighttime solar background illumination, demonstrating the overall high performance of CATS under such conditions."*

**10. Page 10, lines 9-10: "due to the different SNR:::": I think that indeed this is the case. But this contradicts to the author statement of no significant bias between day and night conditions stated earlier (page 9, lines 32-33).**

The reviewer is right on the high importance and effect of SNR is CATS retrievals and algorithms. Statement of page 9, lines 32-33 has been reformulated to avoid possible confusions, according to the reviewer's comment.

**11. Page 10, lines 24-29: I have the feeling that this information should be moved to section 2.2 where the description of CATS data level product is already given. At that section, the authors can present a detailed description of their methodology followed for could screening.**

According to reviewer's recommendation the suggested part of the manuscript was moved (and slightly modified to fit better to the paragraph), to Section 2.1 (former Section 2.2 in the ACPD discussion version).
To be more specific, the suggested part was modified from:

*"In addition to the backscatter coefficient, CATS Level 2 data provide the feature classification of the detected layers (namely: clear air, cloud, aerosol and totally attenuated) and the numerical confidence level of the classification, similar to the CALIOP Cloud-Aerosol-Discrimination (CAD) algorithm (Liu et al., 2004; Liu et al., 2009). CATS Feature Type Score is a multidimensional probability density function (PDF) developed based on multiyear CPL observations, that discriminates cloud and aerosol features, assigning an integer between -10 and 10 for each detected atmospheric layer."*

to:

*"In addition to CATS Level 2 Feature Type (namely: clear air, cloud, aerosol and totally attenuated), the algorithm provides the confidence level of the Feature Type classification, similar to the CALIOP Cloud-Aerosol-Discrimination (CAD) algorithm (Liu et al., 2004; Liu et al., 2009). CATS Feature Type Score is a multidimensional probability density function (PDF) developed based on multiyear CPL observations, that discriminates cloud and aerosol features, assigning an integer between -10 and 10 for each detected atmospheric layer."*

**12. Page 11, line 23: "end of 2018:::" -> Maybe "end of 2019" ?**

The manuscript was modified to:

"*Based on this analysis and comparisons with CALIPSO, the CATS cloud-aerosol discrimination algorithm was updated for the V3-00 Level 2 data products (released in the end of 2018) to improve the accuracy of the Feature Type and Feature Type Score, especially during daytime.*"

**13. Section 3.2: I wonder why the authors constrained their study only to the comparison of aerosol backscatter and they did not proceed with comparison of other aerosol related properties as well (e.g. physical and not properties such as integrated backscatter, AOD, lidar ratio, layer center of mass-thickness). I have the feeling that by taking into account more properties in their comparison will improve the manuscript and will enhance the arguments (i.e. argument of tenuous layer, argument of lidar ratio assumption) for the discrepancies shown here. In addition to that the information provided by each station individually is lost in the analysis demonstrated here. For example, a figure showing the differences between CATS-EARLINET for day and night time conditions per station along with the mean value may explain some of the discrepancies shown in this section (e.g. the argument of topography) or it may reveal other discrepancy patterns if any (i.e. latitudinal).**

CATS products and processing algorithms are provided in different levels of processing. CATS Level 1B (L1B) data include vertical profiles of total and perpendicular attenuated backscatter signals, range-corrected, calibrated and annotated with ancillary meteorological parameters (McGill et al., 2007; Powell et al., 2009; Vaughan et al., 2010). CATS Level 2 (L2) products provide the vertical distribution of aerosol and cloud properties (depolarization ratio, backscatter and extinction coefficient profiles at 1064 nm – FFOV), with a horizontal and vertical resolution of 5 km and 60 m respectively. In addition, L2 data include geophysical parameters of the identified atmospheric layers (vertical feature mask - feature type, aerosol subtype), the required horizontal averaging and information on the feature type classification confidence (Yorks et al., 2019).

Regarding CATS L1B, the validation is a study led by NASA GSFC Team, and more specific by Dr. Rebecca Pauly (Science Systems and Applications Inc., Lanham, 20706, United States Science Systems and Applications Inc., Lanham, 20706, United States), member of the CATS Team. The study is already submitted on AMT journal:

"*Pauly, R. M., Yorks, J. E., Hlavka, D. L., McGill, M. J., Amiridis, V., Palm, S. P., Rodier, S. D., Vaughan, M. A., Selmer, P. A., Kupchock, A. W., Baars, H., and Gialitaki, A.: Cloud Aerosol Transport System (CATS) 1064 nm Calibration and Validation, Atmos. Meas. Tech. Discuss., https://doi.org/10.5194/amt-2019-172, in review, 2019*".

In this study, the EARLINET authors in collaboration with the CATS Team evaluate CATS Level 2 Mode 7.2 v2.01 backscatter profiles at 1064nm (Palm et al., 2016). The reason of focusing to the evaluation of backscatter coefficient is the operation wavelength of CATS, i.e. the 1064nm wavelength. Since EARLINET lidar systems do not provide depolarization ratio measurements at 1064nm the particulate depolarization ratio parameter could not be evaluated, included in the analyssis. In addition, since CATS is a satellite-based elastic backscatter lidar (McGill et al., 2015), in order to provide vertically resolved extinction coefficient profiles (km$^{-1}$) of aerosols and clouds in the Earth's atmosphere, the computation algorithm implements a number of intermediate parameters (i.e. lidar ratio, feature type classification, aerosol subtype classification, among others). Due to the reason that the profiles of

extinction coefficient are a computed product and not included in the direct measurements, extinction coefficient profiles were also not included in the analysis. The authors have focused on particulate backscatter coefficients ($km^{-1}sr^{-1}$), since this is the product directly derived from measurements, the sum of the parallel and perpendicular backscatter measurements (i.e., $\beta_{1064nm\ total} = \beta_{1064nm\ parallel} + \beta_{1064\ perpendicular}$). Future study will include high collocated analysis on the CATS performance and representativeness, including the issues mentioned by the reviewer, based on high temporally and spatially collocated measurements between airborne FAAM Bae-146 research aircraft and ISS measurements, performed in the framework of the AER-D/ICE-Dcampaign, over Cape-Verde (Santiago island), on August 6-25, 2015, as introduced by Marenco et al. (2018) – Figure 5.

[Figure]

*Figure 5: B920 flight on August 7$^{th}$, 2015 over Cape Verde, high collocated with ISS-CATS overpass. Left: B920 flight and ISS footprint (left), and CATS backscatter coefficient 1064nm scene (right).*

Regarding the comment of the reviewer of explicitly addressing the differences between CATS-EARLINET for day and night time conditions per station, along with the mean value to explain some of the discrepancies, it has to be noted that the sample of collocated profiles in many stations does not permit an analysis with strong "per-station" conclusions. For instance, we mention here that Barcelona (ba), Athens_NTUA (at), and Bucharest (bu) stations are participating with only one available case of CATS-EARLINET collocated measurements. In addition, certain number of station happens to contribute with either only nighttime or daytime correlative cases, i.e. Athens_NOA (no) and Lecce (le) with only nighttime cases (three and two cases respectively) and Evora (ev) with only daytime cases (two cases), not allowing to follow the per-station approach.

The undervalue of EARLINET is relying to the approach of the participating community treats EARLINET as a single entity, with the main objective to obtain an extended, coordinated and of continental scale network of sophisticated ground-based Raman lidars and eventually, to foster a quantitative, comprehensive, and statistically significant database of the distribution of aerosol on a continental scale (Bösenberg et al., 2003; Pappalardo et al., 2014). The quality assurance and improvement of the performance of the EARLINET systems is tested through the intercomparison of both the infrastructure (Wandinger et al., 2015) and the optical products (Böckmann et al., 2004; Pappalardo et al., 2004). In addition, the homogenization of the lidar data in a standardized output format is facilitated and an automatic algorithm is developed to further address the quality assurance of the lidar measurements (the Single Calculus Chain (SCC), D'Amico et al, 2015; D'Amico et al, 2016; Mattis et al., 2016).

In order to clarify and demonstrate the sample issue, not allowing to follow a per-station approach, the authors have included here (but also in the manuscript) the following "Table 4", where the cases used in the intercomparison are given.

Table 4: ISS-CATS and EARLINET cases considered in the evaluation process of CATS backscatter coefficient profiles at 1064 nm.

[revised manuscript text omitted]

**14. The pair of observation "i" refer to the vertical height of each case study or to each case study individually? This a general comment related to the comparison methodology followed by the authors: I speculate that the initial vertical resolution of the two profiles is not the same. For example, the L1 data products obtained by CATS are within 60 m vertical resolution (Yorks et al., 2011). On the other hand, the data products obtained by EARLINET (especially the Raman retrievals) are processed (application of low-pass filter on the signal) leading to range-resolution loss. A concept of effective resolution is already discussed in the literature (e.g. Iarlori et al., 2015). Therefore, it is not so clear to the reader how the authors managed to compare values obtained from**

**different atmospheric heights? Did they interpolate their values or they used mean values in specific vertical height windows? In any case the authors are kindly suggested to comment their approach on this. (Iarlori, M., Madonna, F., Rizi, V., Trickl, T., Amodeo, A., Effective resolution concepts for lidar observations, Atmos. Meas. Tech., 8, 5157–5176, 2015 www.atmos-meas-tech.net/8/5157/2015/ doi:10.5194/amt-8-5157-2015).**

The authors agree with the reviewer regarding not properly commenting on the respective aspect. Regarding CATS L2 profiles, the product provides the vertical distribution of aerosol and cloud properties (depolarization ratio, backscatter and extinction coefficient profiles at 1064 nm – FFOV), with a horizontal and vertical resolution of 5km and 60m respectively. On the contrary, EARLINET profiles were provided by the EARLINET community with higher vertical resolution. Towards the assessment of CATS performance, for the comparison of CATS against EARLINET, we implemented the $CATS_i$-$EARLINET_i$ residuals for each pair of observations "i", as a statistical indicator of CATS average overestimation or underestimation of the aerosol load, in terms of backscatter coefficient values. Since the vertical resolution of the two profiles was not the same and in order to compute the $CATS_i$-$EARLINET_i$ residuals, the EARLINET profiles were reduced in resolution to obtain 1-1 datasets, characterized by the same vertical resolution. This was achieved by computing the EARLINET mean backscatter coefficient value from all EARLINET bins within each CATS 60m backscatter coefficient range. Thus, indeed the speculation of the reviewer on the methodology, through computing mean values in specific vertical height windows, is right.

The aforementioned approach indeed led to loss of vertical resolution in the EARLINET profiles (Iarlori et al., 2015). For this reason, the authors (in the initial steps of the study) performed an exercise, to investigate the magnitude of the effect of the selected approach and the significance of loss of resolution in the EARLINET profiles, since the opposite approach (i.e. to increase the resolution of CATS profiles to match the EARLINET resolution), was not feasible.

Figure 6 shows an example of the exercise, corresponding to a nighttime ISS orbit, on September 30, 2015 (blue line), at a minimum distance of 12.9km from the EARLINET Leipzig – Germany PollyXT lidar system (indicated by a white dot), at 22:21 UTC (Fig. 3a). CATS particulate backscatter coefficient cross section at 1064 nm (Fig.6-right) shows the presence of aerosols up to 2.2 km (a.s.l.). CATS spatial-averaged and Leipzig temporal-averaged profiles were derived from CATS profiles within horizontal distance below of 50 km, between the Leipzig station and the ISS footprint.

[Figure]

*Figure 6: (left) Nighttime ISS orbit over EARLINET Leipzig station on the 30th of September 2015 (blue line). The white dot denotes the location of Leipzig lidar system, (b) CATS Backscatter Coefficient at 1064 nm.*

Figure 7 shows the direct comparison between the backscatter coefficient profiles, measured from the EARLINET Leipzig station (red line) and CATS (blue line), along with their standard deviations (horizontal error bars). The profiles indicate the presence of aerosol up to 2.6 km height (a.s.l.). The intercompared profiles between ISS-CATS and EARLINET-Leipzig station are characterized by adequate agreement, although significant discrepancies were also present, especially to the lowermost part of the profiles, as discussed in the manuscript.

The intercomparison presented in Figure 7 is shown to provide to the reviewer a quantitative response to the specific comment. Figure 7 shows the CATS averaged backscatter coefficient profile in blue color, while with respect to EARLINET both the initial (high resolution) and final (reduced in resolution to match the CATS profile resolution)

are provided in black and red colors. As was observed the necessary loss resolution in the EARLINET profiles for achieving vertical match between the two datasets is very low, with final EARLINET profile following with high accuracy the characterizes and tendencies, both qualitative and quantitative, of the initial EARLINET profiles.

[Figure]

*Figure 7: CATS and EARLINET-Leipzig backscatter coefficient profiles (1064 nm) for the nighttime ISS orbit over EARLINET Leipzig station on the 30th of September 2015. CATS backscatter coefficient profile at 1064nm is shown in blue line. EARLINET-Leipzig initial and final profiles, are shown is black and red respectively.*

However, the authors agree with the reviewer on the absence of properly addressing the vertical match between the two datasets. For this reason, the following part was added on "*Section 2.3.2 - Particle backscatter coefficient retrievals from ground based lidars at 1064 nm*":

"*Finally, in order to perform the intercomparison between CATS and EARLINET profiles, the high resolution of EARLINET profiles was lowered to match the vertical resolution of CATS profiles (i.e. 60m). The objective of obtaining profiles of similar vertical resolution was addressed through computing the EARLINET mean backscatter coefficient value from all EARLINET bins within each CATS 60m backscatter coefficient height range. The computed EARLINET profiles of similar vertical resolution with CATS followed with high accuracy the characterizes and tendencies, both qualitative and quantitative, of the initial EARLINET profiles, despite the loss of vertical resolution (Iarlori et al., 2015).*".

**15. Page 13, line 30: "CALIOP" -> Maybe "CATS" instead of CALIOP?**

CATS calibration is performed by normalizing the NRB signal in the altitude regime between 23 and 27 km. Although the region is used to normalize the NRB signal to the molecular backscatter, the region between 23 and 27 km is not aerosol free. According to the ATBD, the scattering ratios (e.g. total backscatter to molecular backscatter) at 532 nm are estimated based on CALIPSO CALIOP V4 L1 data. The 532 nm scattering ratios are used to estimate the 1064 nm scattering ratios and accordingly to the calibration of CATS. Consequently, a source of systematic errors in the CATS calibration is related to errors in the stratospheric scattering ratios provided by CALIPSO (ΔR). The scattering ratio values in CATS calibration are determined as outlined in section 3.3.4. of the CATS ATBD (https://cats.gsfc.nasa.gov/media/docs/CATS_ATBD.pdf; last visit: 29/05/2019).

**16. Page 15, line 18, lines 24-25: "slight underestimations of the total AOD in climatic studies." "results in large AOD biases and unrealistic AOD values." I agree with these statements. However, in the current state of the manuscript there is no straight forward comparison of AOD but only backscatter coefficient. See also my previous specific comment No. 14.**

The authors agree with the reviewer. Although not a CATS extinction coefficient 1064nm and AOD 1064 nm analysis were not included, the authors in order to provide a more detailed overview of CATS capabilities and

representativeness have included literature review on studies investigating the performance of CATS. To be more specific, the following paragraph was added to the manuscript (Section 1 - Introduction), in line to the comment of the reviewer and in order to justify the statement mentioned ny the reviewer:

"*CATS performance has been validated against ground-based AErosol RObotic NETwork (AERONET; Holben et al., 1998) measurements and evaluated against satellite-based Atmospheric Optical Depth (AOD) retrievals of Aqua and Terra Moderate Imaging Spectroradiometer (MODIS; Levy et al., 2013) and active CPL (McGill et al., 2002) and CALIPSO CALIOP (Winker et al., 2009) profiles of extinction coefficient and AOD at 1064 nm. Lee et al. (2018) compared daytime quality-assured CATS V2-01 vertically integrated extinction coefficient profiles (1064 nm) and AERONET AOD (1020 nm) values, spatially (within 0.4º Longitude and Latitude) and temporally (±30 minutes) collocated, and found a reasonable agreement with a correlation of 0.64. A comparative analysis of CATS and MODIS C6.1 Dark Target (DT) AOD retrievals, through spectral interpolation between 0.87 and 1.24 µm channels, reported correlation of 0.75 and slope of 0.79, over ocean. In addition, Lee et al., (2019) evaluated AOD and extinction coefficient profiles from CATS through intercomparison with CALIOP. With regard to AOD, analysis a total of 2681 CATS and CALIOP collocated observation cases (within 0.4º Longitude/Latitude and ±30 minutes ISS and CALIPSO overpass difference), showed correlation of 0.62 and 0.52 over land and ocean respectively during daytime (1342 cases), and 0.84 and 0.81 over land and ocean respectively during nighttime (1339 cases). Comparison of CATS and CALIOP collocated extinction coefficient profiles based on the closest Euclidian distance on the earth's surface, shows also good shape agreement, despite an apparent CALIOP underestimation in the lowest 2 km height. CATS and CALIOP observations were used by Rajapakshe et al. (2017) to study the seasonally transported aerosol layers over the SE Atlantic Ocean. The performed comparative analysis reported on similar geographical patterns regarding Above Cloud Aerosols (ACA), Cloud Fraction (CF) and ACA occurrence frequency (ACA_F) between CATS and CALIOP retrievals. However, the authors reported also on differences between CATS and CALIOP vertical aerosol distributions, with ACA bottom height identified by CATS lower than the respective of CALIOP. CATS retrievals were used to document the diurnal cycle and variations of clouds, with CALIOP complementarily used. Noel et al. (2018) showed that both CATS and CALIOP profiles of CF agree well on both the vertical patterns and values at 01:30 and 13:30 LT, over both land and ocean, with minor differences of the order of 2-7% throughout the entire profiles of cloud fraction. CATS depolarization measurements, which are critical in the processing algorithms of aerosol subtype classification, were investigated in the case of desert dust, smoke from biomass burning and cirrus clouds (Yorks et al., 2016), and were found consistent and in good agreement with depolarization measurements from previous studies and historical datasets implementing CPL (Yorks et al., 2011) and CALIOP (Liu et al., 2015).*

**17. Page 29, line 13: "The white circle" -> "The white dot denotes the location". The white circle refers to points at various distances from the lidar station as stated by the authors in Figure 2. Please consider correcting this minor typo in figures 3, 4, and 5.**

The text is modified according to the reviewer's recommendation.

**18. Figure 7: For the night time mean profiles the discrepancies are negligible but for the day time and specifically for the height region from 1-2 km large differences are observed. What is the main reason behind this? The significant influence of the topography? In that case why this difference is not shown also in the nigh-time profiles, considering this as a bias from one or more stations. The low daytime CATS SNR? In that case I would expect to see higher discrepancies than sown inside the PBL (longer atmospheric path), compared to 1-2 km. The calibration region of CATS? In any case, I think that a solid and quantitative explanation on this is missing.**

The effect of signal-to-noise ratio (SNR) and the associated Minimum Detection Backscatter (MDB) are the critical factors determining the performance of CATS. However along with the technical capabilities of CATS there are different factors with effect on the final CATS profiles (i.e. topography, as mentioned by the reviewer). Regarding the quantitative and qualitative explanation exercises under different cases are presented and discussed in the reviewer's question #7.

**Interactive comment on "EARLINET evaluation of the CATS L2 aerosol backscatter coefficient product"**
**by Emmanouil Proestakis et al.**
**Anonymous Referee #2**

The authors would like to thank the reviewer for the interesting and at the same time substantial comments and suggestions. We tried, and did our best, to incorporate the most suitable proposed changes and corrections in the revised manuscript, aiming at improving the presented paper. Following, you will find our responses, one by one to the comments addressed, in the uploaded supplement pdf file.
Kind regards,
Emmanouil Proestakis

**General comments:**
**This manuscript compares EARLINET (ground-based) and CATS (onboard the international spatial station) retrievals of the aerosol backscatter coefficient over 12 European sites and 1 Asian site. The paper is well written, however, I did miss some explanation in the introduction about the importance of CATS product. I believe this could be easily achieved by modifying the order of some paragraphs and including extra information. In particular, I suggest moving the second paragraph of Section 2.2 (page 4, line 30 to page 5, line 12) to the introduction, with the due adjustments.**

The authors agree with the reviewer. The science goals of CATS, indeed, were not mentioned in the introduction, leading to issues in the understanding of the scientific importance of the project in the early stages of the manuscript. For this reason the authors have followed the referee's recommendation to rearrange the manuscript, making at the same time all the appropriate modifications to ensure that the adjustments did not have a negative impact to the understanding of the manuscript context. To be more specific, the following section was added to the introduction:

*"CATS was developed to meet three main science goals. The primary objective was to measure and characterize aerosols and clouds on a global scale. The space-borne lidar orbited the Earth at an altitude of approximately 405 km and 51-degree inclination. The use of the ISS as an observation platform facilitated for the first time global lidar-based climatic studies of aerosols and clouds at various local times (Noel et al., 2018, Lee et al., 2018). In addition, near-real-time data acquisition of the CATS observations was developed towards the improvement of aerosol forecast models (Hughes et al, 2016). A secondary objective was related to the need of long-term and continuous satellite-based lidar observations to be available for climatic studies. The first spaceborne lidar mission, the Lidar In-space Technology Experiment (LITE; McCormick et al., 1993) in 1994, was succeeded by the joint NASA and Centre National d'Études Spatiales (CNES) Cloud-Aerosol Lidar and Infrared Pathfinder Satellite Observation (CALIPSO) mission in June, 2006 (Winker et al., 2007). Since 2009 the Cloud-Aerosol Lidar with Orthogonal Polarization (CALIOP) instrument (Winker et al., 2009) onboard CALIPSO operates on the secondary backup laser. The launch of the post-CALIPSO missions, the joint European Space Agency (ESA) and JAXA satellite Earth Cloud Aerosol and Radiation Explorer (EarthCARE; Illingworth et al., 2015) and the NASA's Aerosols, Clouds, and Ecosystems (ACE) are planned for 2021 and post-2020 respectively. The CATS project was partially intended to fill a potential gap on global lidar observations of vertical aerosols and clouds profiling. The third scientific objective of CATS was to serve as a low-cost technological demonstration for future satellite lidar missions (McGill et al., 2015). Its science goal to explore different technologies was fulfilled through the use of photon-counting detectors and of two low energy (1-2 mJ) and high repetition rate (4-5 kHz) Nd:YVO4 lasers (Multi-Beam and HSRL - UV demonstrations), aiming to provide simultaneous multiwavelength observations (355, 532 and 1064 nm). Additional gains of the CATS were related to the exploitation and risk reduction of newly applied laser technologies, to pave the way for future spaceborne lidar missions (high repetition rate, injection seeding, wavelength tripling at 355 nm)."*

**I also suggest comparing some scenes of coincident vertical profiles of CATS and CALIOP. Would that be possible? I believe this would dramatically improve the visibility of the paper. Also, it wasn't clear to me**

**whether CATS should only be used to fill a gap in space-based lidar observations or if it is as reliable as CALIOP. I believe this should be further clarified in the text.**

The suggested evaluation study between CATS and CALIOP has been already performed by Lee et al. (2018), Rajapakshe et al. (2017), Noel et al. (2018) and Yorks et al., 2016, reporting also on the good agreement of the intercomparison studies. However, the authors agree with the reviewer regarding the necessity of including the findings of the aforementioned studies and have adjusted the manuscript accordingly. To be more specific, the following paragraph was added to the manuscript (Section 1 - Introduction):

*"CATS performance has been validated against ground-based AErosol RObotic NETwork (AERONET; Holben et al., 1998) measurements and evaluated against satellite-based Atmospheric Optical Depth (AOD) retrievals of Aqua and Terra Moderate Imaging Spectroradiometer (MODIS; Levy et al., 2013) and active CPL (McGill et al., 2002) and CALIPSO CALIOP (Winker et al., 2009) profiles of extinction coefficient and AOD at 1064 nm. Lee et al. (2018) compared daytime quality-assured CATS V2-01 vertically integrated extinction coefficient profiles (1064 nm) and AERONET AOD (1020 nm) values, spatially (within 0.4° Longitude and Latitude) and temporally (±30 minutes) collocated, and found a reasonable agreement with a correlation of 0.64. A comparative analysis of CATS and MODIS C6.1 Dark Target (DT) AOD retrievals, through spectral interpolation between 0.87 and 1.24 µm channels, reported correlation of 0.75 and slope of 0.79, over ocean. In addition, Lee et al., (2019) evaluated AOD and extinction coefficient profiles from CATS through intercomparison with CALIOP. With regard to AOD, analysis a total of 2681 CATS and CALIOP collocated observation cases (within 0.4° Longitude/Latitude and ±30 minutes ISS and CALIPSO overpass difference), showed correlation of 0.62 and 0.52 over land and ocean respectively during daytime (1342 cases), and 0.84 and 0.81 over land and ocean respectively during nighttime (1339 cases). Comparison of CATS and CALIOP collocated extinction coefficient profiles based on the closest Euclidian distance on the earth's surface, shows also good shape agreement, despite an apparent CALIOP underestimation in the lowest 2 km height. CATS and CALIOP observations were used by Rajapakshe et al. (2017) to study the seasonally transported aerosol layers over the SE Atlantic Ocean. The performed comparative analysis reported on similar geographical patterns regarding Above Cloud Aerosols (ACA), Cloud Fraction (CF) and ACA occurrence frequency (ACA_F) between CATS and CALIOP retrievals. However, the authors reported also on differences between CATS and CALIOP vertical aerosol distributions, with ACA bottom height identified by CATS lower than the respective of CALIOP. CATS retrievals were used to document the diurnal cycle and variations of clouds, with CALIOP complementarily used. Noel et al. (2018) showed that both CATS and CALIOP profiles of CF agree well on both the vertical patterns and values at 01:30 and 13:30 LT, over both land and ocean, with minor differences of the order of 2-7% throughout the entire profiles of cloud fraction. CATS depolarization measurements, which are critical in the processing algorithms of aerosol subtype classification, were investigated in the case of desert dust, smoke from biomass burning and cirrus clouds (Yorks et al., 2016), and were found consistent and in good agreement with depolarization measurements from previous studies and historical datasets implementing CPL (Yorks et al., 2011) and CALIOP (Liu et al., 2015).*

**I also believe a final paragraph stating the main conclusion is needed (that is, what are your suggestions for future studies: should we use CATS or not, under which conditions these retrievals are reliable, what are their advantages and disadvantages and how could future studies benefit - or not - from CATS).**

The authors agree with the reviewer and a final paragraph stating suggestions related the use of the unique CAST dataset was included. To be more specific, the following section was added to the "Summary and Conclusions section":

*"The qualitative and quantitative agreement between CATS and EARLINET reported in this study is encouraging, especially during nighttime, agreement that will hopefully facilitate further studies implementing CATS observations in the future. CATS, for a period of almost three years, provided an unprecedented global dataset of vertical profiles of aerosols and clouds, much like CALIOP, taking though advantage of the unique orbital characteristics of the ISS. ISS enabled CATS to provide for the first time satellite-based lidar measurements of the diurnal evolution of aerosols and clouds over the tropics and midlatitudes, and to be more specific to latitudes below 52°. Since CALIPSO and Aeolus (and in the future also EarthCARE) are polar sun-synchronous satellites of fixed equatorial crossing time (01:30 and 13:30 LT for CALIOP, 06:00 and 18:00 for ALADIN), it is expected that, at least for the near future, CATS dataset will remain the only available satellite-based lidar source of nearly global diurnal measurements of*

*atmospheric aerosols and clouds. In addition, while CALIOP is a two-wavelength lidar system operating at 532 nm and 1064 nm with depolarization capabilities at 532 nm, CATS provided satellite-based aerosol and cloud depolarization profiles at 1064 nm, thus in a different wavelength. This dataset, much like CALIOP dataset, is especially useful for studies of the three-dimensional distribution of non-spherical aerosol particles in the atmosphere (e.g. mineral dust and volcanic ash), and especially since it is an active sensor, over regions of high reflectivity (e.g. deserts, ice). Future studies including the exploitations of CATS unique observations may help the scientific community to shed new light on physical processes of aerosols and clouds in the Earth's atmosphere."*

**Specific comments:**
**page 2, line 3 - Please modify "Physic" to "Physics".**

The text is modified according to the reviewer's recommendation.

**page 2, line 20 - Please reformulate the sentence (suggestion: "Quality assessment of CATS...").**

The text is modified according to the reviewer's recommendation.

**page 2, line 24 - Please modify "consists" to "consists of".**

The text is modified according to the reviewer's recommendation.

**page 3, line 15 - What is the difference between capacity and capability?**

The text is reformulated according to the reviewer's recommendation:

*"Since the beginning of the initiative in 2000, EARLINET has significantly increased its observing and operational capacity"*

**page 3, line 16 - Please reformulate or remove the sentence "EARLINET stations are classified as active on condition of...".**

According to the reviewer's comment, the sentence was reformulated to:

*"EARLINET stations are classified between "active", "not permanent", "joining" and "not active". An EARLINET station is classified as active when on condition of performing regularly and simultaneously measurements with the other stations composing the lidar network, and accordingly, contributing with uploading the performed measurements to the EARLINET database (https://www.earlinet.org/, last access: 20 December 2018)."*

**page 4, line 32 - Please modify "space-borne" to "spaceborne".**

The text is modified according to the reviewer's recommendation.

**page 6, line 16 - It's not clear to me if observations more than 90 minutes apart were compared or not. Could you clarify this?**

The study follows the CALIPSO CALIOP validation methodology developed in the framework of a collaboration between ESA and EARLINET collaboration (Pappalardo et al., 2010). The ESA dedicated program of collocated and concurrent EARLINET observations with CALIOP observations was developed prior to the launch of CALIPSO and is planned with a duration until the end-of-mission of the mission. On the contrary of the well-established CALIPSO-EARLINET validation activity, but also to the ESA-Aeolus and to the upcoming ESA-EarthCARE satellite missions, a similar CATS-EARLINET validation strategy was not established.
The participating EARLIENT stations in the study contributed to the evaluation of CATS through measurements performed during the fixed-scheduled program of EARLINET operation. As described in Pappalardo et al (2014), the EARLINET scheduled program of measurements includes three measurements per week, one during daytime around local noon (Monday, 14:00 ± 1h) and two during nighttime (Monday/Thursday, sunset + 2/3h), to enable

Raman extinction retrievals. In addition, EARLIENT operates a small number of lidar systems capable for 24/7 continuous measurements (Engelmann et al., 2016).

The absence of an established dedicated validation activity between NASA and EARLINET prior to the operation of CATS, in combination with the fixed measurements schedule of EARLINET, the high variable overpass-time of CATS (bounded by the orbital characteristics of ISS) and the frequently cloud-contaminated cases led to a low number of collocated and concurrent EARLINET-CATS cases to be available for the study. Eventually, this obstacle was tackled through the cooperative effort of a large number of EARLINET stations, contributing through the already performed measurements. The increasing number of EARLINET stations showing interest to contribute to the study led to an overall of forty-seven (47) available cloud-free EARLINET-CATS collocated cases to implement for the evaluation of CATS.

The EARLINET-CATS correlative study considers the collocation criteria established in the validation plan of CALIPSO. Regarding the spatial collocation, EARLINET participating stations contributed with measurements when the ISS overpass was within 50 km horizontal radius from their location.

Regarding the temporal collocation, the study implemented ground-based measurements with a temporal window of EARLINET performed measurements with starting time, or stop time as close in time as possible to the ISS overpass. Accordingly, all the identified EARLINET cases where studied, through case-by-case inspection of the Range-Corrected-Signal quicklooks, for atmospheric homogeneity was of high importance, and additionally for other constrains (e.g. cirrus-clouds). During the first twenty months of CATS operation, based on thirteen EARLINET contributing stations, only 47 cases were found suitable to be used in the comparison. From the total of 47 cases, 44 where performed with "*starting time*", or "*stop time*" within 90 minutes of the ISS overpass. For this reason why the phrase *"typically within 90 minutes of the ISS overpass"* was used in the manuscript. In addition, it has to be mentioned that in the majority of the EARLINET cases encompasses the ISS overpass. The length of the temporal window was variable, based-on the expertise of the EARLINET teams, the homogeneity of the atmospheric scenes and the unique cloud constrains of each case, in order to allow retrievals of high-quality EARLINET backscatter coefficient profiles.

The authors agree though with the reviewer that this part of the manuscript was not clear, therefore the manuscript was revised in the 2.3.1 section referring to the "Comparison methodology", and in addition the manuscript was updated with the following table (*"Table 2"* in the manuscript) that includes information on the correlative cases used in the study. The table provides the "Day-Night Flag" of the study case, "Date" and "Time" of the ISS overpass, the corresponding EARLINET station and the minimum distance between the ISS orbit-track and the station location, and finally the EARLINET temporal window of measurements.

In Section 2.3.1, the following part of the manuscript was reformulated according to the reviewer's recommendation, from:

[revised manuscript text omitted]

**page 6, line 24 - What does "including cirrus clouds" mean? Cirrus clouds scenes were used or not?**

The authors acknowledge that the sentence was not clearly written, thus the sentence was reformulated from:
*"In addition, to account for contamination effects of multiple-scattering and specular reflection in the intercomparison process, only cloud-free (including cirrus clouds) atmospheric scenes are used."*
to:
*"In addition, to account for contamination effects of multiple-scattering and specular reflection in the intercomparison process, only cloud-free atmospheric scenes are used. Cases with detected cirrus either at the EARLINET Range-Corrected-Signal quicklooks or at the ISS-CATS backscatter coefficient profiles or the feature type profiles are not considered in the study."*

**page 7, line 19 - Please modify "participated" to "participating".**

The text is modified according to the reviewer's recommendation.

**page 7, line 20 - "exited". Did you mean "excited"?**

The reviewer is correct, the text is modified according to the reviewer's comment.

**page 9, line 14 – Please modify "in details" to "in detail".**

The text is modified according to the reviewer's recommendation.

**page 9, line 24 - Please modify "below" to "of".**

The text is modified according to the reviewer's recommendation.

**Page 9, line 37 - Please modify "over-lying" to "overlaying".**

The text is modified according to the reviewer's recommendation.

**page 11, line 22 - Has the new product already been released? How does the new algorithm differ from the previous one? What kind of improvements does it present?**

CATS_V3-00 replaced CATS_V2-05 on October 1$^{st}$, 2018. Initially, the changes in CATS Level 1 and Level 2 algorithms corresponding to CATS Version 3-00 data was planned to be the final algorithm release for the CATS project, though observed issues in the CATS products led to the modifications of V3-00 and the release of the V3-01 later on in the beginning of 2019. Since CATS products are provided in different levels of processing, the made changes in the algorithms correspond to both L1B and L2O products.
To be more specific, the changes in the L1B algorithms include:
(1) improvement of the nighttime attenuated total backscatter (ATB) profiles due to improvements in the calibration of CATS, thus improvement also in the daytime ATB profiles, since nighttime ATB is implemented in the calculations of the daytime calibration.
(2) changes to the "Depolarization_Quality_Flag", and
(3) implementation of MERRA-2 Reanalysis data instead of GMAO forecasts, for the meteorology in V3-00 and V3-01.
The changes made in the algorithms of CATS L1B reflect on improvements on CATS L2O products. Though additional changes in CATS L2O algorithms include also:
(1) updates in number of profiles in the L2O datasets
(2) improvements in the calculations of uncertainties in the L2O layer-integrated parameters
(3) changes to the "Depolarization_Quality_Flag"
(4) improvements of the Cloud Aerosol Discrimination (CAD) through the implementation of an additional parameter, namely the "Cloud_350m_Fraction_XXX_FOV", to report of the number of 350 L1B profiles within each 5 km L2O bin of the L2O layer product with attenuated total backscatter values greater than 0.03 km$^{-1}$sr$^{-1}$, thus atmospheric features of high probability of being a cloud. In addition, the parameter "Num_Profs_Avg_LRatio_XXX_FOV" was added to the L2O Layer data product.
(5) improvements in CATS Feature Type and Feature Type Score variables, but also in the Aerosol Subtype classification (replace of "volcanic" with "UTLS Aerosol") and addition of the parameters "Opaque_Feature_Optical_Depth_1064_XXX_FOV" and "Opaque_Feature_Optical_Depth_Uncertainty_1064_XXX_FOV" in Mode 7.2 L2O datasets.
(6) Updates in the Lidar Ratio (LR) values for cirrus clouds
(7) update of the effective multiple scattering factor for ice clouds values to 0.52.
The above changes in the CATS V3-00 and V3-01 algorithms and the respective products are extensively presented and in-depth discussed in the CATS official website (; last visit on: 22/05/2019), in the "Publications" section.

**pages 11 and 12, Section 3.2 and Table 2: It would be interesting to show the mean relative bias (that is bias over mean value).**

According to the referee's comment we have computed and included in the table of comparison statistics between CATS and EARLINET the Mean Relative Bias (MRB), calculated as follows:

$$MRB = \left(\frac{1}{n}\sum_{1}^{n}\frac{(b_{CATS} - b_{EAR})}{b_{EAR}}\right) * 100$$

The MRB were found equal to -24.06% and -19.84% for daytime and nighttime CATS observations respectively, and the results were included to the table.

**page 14, line 10, Please modify "discrepancies" to "discrepancy".**

The text is modified according to the reviewer's recommendation.

**page 15, line 1, Please modify "based to" to "based on".**

The text is modified according to the reviewer's recommendation.

**Figs. 3 to 5: Please use "b) CATS backscatter coefficient at 1064 nm", or "(1064 nm)".**

The text is modified according to the reviewer's recommendation.

**Fig. 5: I would guess topography influence CATS coefficient quite significantly. Could it be causing the spykes shown in this figure? Could you provide a quantitative estimate of the contributing effect of topography on the discrepancy observed in this figure? What about an estimate of the other contributing effects?**

Regarding the question of the reviewer on the discussed constrains on the dataset, Figures 1-4 show quantitatively the effects of (i) distance between the EARLINET station and the closest profile of the CATS-ISS overpass for each correlative case, (ii) CATS Feature Type, (iii) number of CATS Level 2 (L2) Aerosol Profiles (APro) used in the CATS horizontal average, and the effect of (iv) topography of EARLINET stations. The comparison exercise examines the effect of one discussed constrain at a time, while keeping all the other parameters in the methodology constant, and considers various evaluation metrics, as discussed in the following sections.

*(i) Effect of distance between the EARLINET station and the closest profile of the CATS-ISS overpass*

Figure 1 shows the effect of distance between the closest CATS L2 APro and the respective EARLINET station matchup, for different upper Euclidean distance thresholds (i.e.: 5n km, $n\in N=\{1,10\}$). To be more specific, the Mean Bias (MB; [Mm$^{-1}$sr$^{-1}$]) - (Fig.1a), Root Mean Square Error (RMSE; [Mm$^{-1}$sr$^{-1}$]) - (Fig.1b), Correlation Coefficient (Fig.1c), and the number of CATS-EARLINET correlative cases per each upper distance threshold are considered. For each upper distance threshold, all the available CATS-EARLINET cases of Euclidean distance lower or equal to the respective upper limit are considered in the computation of the aforementioned evaluation metrics. This cumulative approach is selected due to the limited number of CATS-EARLINET correlative cases, and is applied separately for daytime and nighttime ISS overpasses, due to the different CATS measurement conditions.
Based on the analysis, during nighttime (daytime), the CATS-EARLINET MB is increasing (decreasing) starting from the 5 km upper distance threshold, to reach -0.0300 (-0.123) Mm$^{-1}$sr$^{-1}$, for the radius threshold of 50km shown in the study. The computed RMSE values are in the range between 0.447 and 0.343 Mm$^{-1}$sr$^{-1}$ for nighttime and between 0.357 and 0.448 Mm$^{-1}$sr$^{-1}$ for daytime, for the distance thresholds of 5km and 50km respectively. The minimum RMSE values are observed when considering ISS overpass cases of closer than 40 km distance to the EARLINET stations during nighttime, corresponding to MB of 0.018 Mm$^{-1}$sr$^{-1}$. The Correlation Coefficient is decreasing with increasing distance between the ISS overpass and the EARLINET stations. Notably, the Correlation Coefficient is not changing considerably for thresholds between 15 and 40 km for nighttime (~0.8) and between 15 and 30 km for daytime (~ 0.7). Sharp decreases in the Correlation Coefficient are observed during daytime (0.547), for distances closer to the EARLINET stations than during nighttime (0.693), for 35 and 40 km distance respectively.

The observed tendencies can be explained in terms of the distance thresholds and number of available cases, since the distance thresholds define the number of cases that are used in the analysis and the number of case is critical to assess the performance of CATS. Consequently, the MB, RMSE and Correlation Coefficient are all subject to both the number and the characteristics of the CATS-EARLINET cases used. In the study the authors use the maximum number of available EARLINET cases, to avoid any possible selection effect resulting from a poor sample of correlative cases, when strict collocation filters are applied. Using the maximum number of available correlative cases, i.e. twenty six (26) and twenty one (21) for nighttime and daytime respectively, for ISS overpasses within 50km radius from the EARLINET stations, the authors envisage to quantitatively address the question of CATS performance and the representativeness of the aerosol backscatter coefficient profiles, over various atmospheric, illumination and ISS overpass conditions.

[Figure]

*Figure 1: CATS backscatter coefficient at 1064nm with respect to EARLINET ground-based measurements, as a function of distance (km) between the closest CATS Level 2 Aerosol Profile and the respective "collocated" EARLINET station, for daytime (red line) and nighttime (blue line) ISS overpasses. Left: Mean Bias [Mm$^{-1}$sr$^{-1}$], center: RMSE [Mm$^{-1}$sr$^{-1}$] and right: Correlation Coefficient. Dashed lines correspond to the number of CATS-EARLINET correlative cases considered per each upper distance threshold between the CATS footprint and the locations of EARLINET stations.*

**(ii) Effect of Feature Type Score**

The main objective of the CATS Cloud Aerosol Discrimination (CAD) score, or Feature Type Score, is to provide to the Feature Type classification a level of confidence. In the case of CATS, the Feature Type score is an integer number ranging between -10 and 10. The values of CATS Feature Type score correspond to classified aerosol atmospheric layers (negative values) and cloud atmospheric layers (positive values), while the magnitude of the Feature Type score corresponds to the confidence level of the classification. A value of -10 indicates complete confidence that the layer is an aerosol layer, while Feature Type score equal to 0, indicates an atmospherics layer with equal probability to be cloud or aerosol.

Figure 2 shows the effect of Feature Type Score, for different values, between -8 and 0 (i.e. for atmospheric layers classified as aerosol layers). The Mean Bias (MB; [Mm$^{-1}$sr$^{-1}$]) - (Fig.2a), Root Mean Square Error (RMSE; [Mm$^{-1}$sr$^{-1}$]) - (Fig.2b) and Correlation Coefficient (Fig.2c) are shown per each Feature Type Score. For each Feature Type score, cases of lower classification confidence level are not considered in the assessment of CATS performance and representativity, indicating the effect of the selected Feature Type thresholds.

Based on the MB, RMSE and Correlation Coefficient, a similar tendency is observed for different Feature Type Scores. To be more specific, not considerable changes are observed for different Feature Type Scores, regardless of the selected Feature Type threshold. This effect is due to the atmospheric characteristics of the CATS-EARLINET cases considered in the analysis. In the framework of the study, to account for contamination effects of multiple-scattering and specular reflection in the intercomparison process, only cloud-free atmospheric scenes are used. Furthermore, cases with detected cirrus, either at the EARLINET Range-Corrected-Signal quicklooks or at the ISS-CATS backscatter coefficient profiles or the feature type profiles, are not considered in the study. Initially, the presence of clouds was investigated through the implementation of CATS backscatter coefficient and depolarization time-height images and EARLINET range-corrected-signal. Cases for which the retrieval of EARLINET temporally-averaged profile was not feasible due to the presence of clouds, and/or CATS cases that the presence of clouds propagated into the CATS spatial-averaged profile were discarded from the analysis. Consequently, the lack of dependence shown in Figure 2 (a-c) is the result from the a priory selection of cloud free conditions selected in the analysis. However, a notably characteristic is the nighttime performance of CATS, which as shown from the lower

absolute MB and lower RMSE, but in addition from the higher Correlation Coefficient values, due to higher SNR, is more representative than the corresponding daytime performance.

[Figure]

*Figure 2: CATS backscatter coefficient at 1064nm with respect to EARLINET ground-based measurements, as a function of Feature Type score, for daytime (red line) and nighttime (blue line) ISS overpasses. Left: Mean Bias [Mm$^{-1}$sr$^{-1}$], center: RMSE [Mm$^{-1}$sr$^{-1}$] and right: Correlation Coefficient.*

*(iii) Effect of number of CATS-ISS L2 aerosol profiles used in the spatial averaging*

Similarly to the analysis presented and discussed above, Figure 3 shows the effect of different number of aerosol profiles used when spatially averaging to retrieve the CATS aerosol profiles used in the framework of the study. In Figure 3, the acronym "CPro" corresponds to the closest CATS profiles to the corresponding EARLINET station. Accordingly, the Mean Bias (MB; [Mm$^{-1}$sr$^{-1}$]) - (Fig.3a), Root Mean Square Error (RMSE; [Mm$^{-1}$sr$^{-1}$]) - (Fig.3b), Correlation Coefficient (Fig.3c), are computed for different number of profiles used (i.e. CPro±1Profile, CPro±2Profiles, …).

Based on the MB, RMSE and Correlation Coefficient, the representativeness of CATS spatial profile is increasing with increasing number of aerosol profiles used in the horizontal averaging. To be more specific nighttime MB is almost constant, showing a low dependence on the number of profiles used, while for daytime CATS cases the opposite effect is observed, with improvement of CATS performance though increasing number of profiles used. Regarding RMSE no significant changes are observed, though a slight decreasing tendency in the RMSE is observed for both daytime and nighttime cases. Regarding the Correlation Coefficient, increasing in the values is also observed, with increasing number of profiles used, both for daytime and nighttime cases, denoting the improvement of the representativeness with increasing number of CATS profiles used in the spatial averaging.

[Figure]

*Figure 3: CATS backscatter coefficient at 1064nm with respect to EARLINET ground-based measurements, as a function of the number of L2 Aerosol Profiles used in the CATS spatial averaging, for daytime (red line) and nighttime (blue line) ISS overpasses. Left: Mean Bias [Mm$^{-1}$sr$^{-1}$], center: RMSE [Mm$^{-1}$sr$^{-1}$] and right: Correlation Coefficient. "CPro" corresponds to the closest CATS profile to the EARLINET station.*

*(iv) Effect of EARLINET stations topography*

In order to study the effect of topography on the CATS profiles the authors separated the participating EARLINET stations into 3 clusters: Continental (Case I – Belsk, Bucharest, Leipzig, and Warsaw), Coastal (Case II – NOA, Athens

NTUA, Barcelona, Cabauw, Thessaloniki and Lecce) and Mountainous (Case III – Dushanbe, Evora, Observatory Hohenpeissenberg, Potenza). The three clusters and the characteristics of the stations are given in Table 1. In addition, Figure 4 shows the locations of the participating stations; green circles denote Continental stations, blue circles denote Coastal stations and brown circles denote Mountainous stations. Figure 4 shows, additionally to the geographical distribution of the active EARLINET stations, the daytime/nighttime overpasses of ISS within the evaluation period, between 02/2015 and 09/2016, encompassing the first twenty months of CATS operation. Due to the limited available dataset of CATS-EARLINET cases, the daytime/nighttime approach was not followed in the case of the analysis regarding the effect of topography.

*Table 1: Clustering of EARLINET stations with respect to topographical features.*

| Case I - Continental | | | | |
|---|---|---|---|---|
| EARLINET Station | Identification Code | Latitude (°N) | Longitude (°E) | Altitude a.s.l. (m) |
| Belsk | be | 51.83 | 20.78 | 180 |
| Bucharest | bu | 44.35 | 26.03 | 93 |
| Leipzig | le | 51.35 | 12.43 | 90 |
| Warsaw | wa | 52.21 | 20.98 | 112 |

| Case II - Coastal | | | | |
|---|---|---|---|---|
| EARLINET Station | Identification Code | Latitude (°N) | Longitude (°E) | Altitude a.s.l. (m) |
| Athens-NOA | no | 37.97 | 23.72 | 86 |
| Athens-NTUA | at | 37.96 | 23.78 | 212 |
| Barcelona | ba | 41.39 | 2.12 | 115 |
| Cabauw | ca | 51.97 | 4.93 | 0 |
| Thessaloniki | th | 40.63 | 22.95 | 50 |
| Lecce | lc | 40.33 | 18.10 | 30 |

| Case III - Mountainous | | | | |
|---|---|---|---|---|
| EARLINET Station | Identification Code | Latitude (°N) | Longitude (°E) | Altitude a.s.l. (m) |
| Dushanbe | du | 38.56 | 68.86 | 864 |
| Évora | ev | 38.57 | -7.91 | 293 |
| Observatory Hohenpeissenberg | oh | 47.8 | 11.01 | 974 |
| Potenza | po | 40.60 | 15.72 | 760 |

[Figure]

*Figure 4: Distribution of EARLINET lidar stations over Europe and West Asia. Green dots: Continental stations used in the inter-comparison. Blue dots: Coastal stations used in the inter-comparison. Brown dots: Mountainous stations used in the inter-comparison. ISS orbits between 02/2015 and 09/2016 are overlaid in red for daytime and in blue for nighttime overpasses.*

Figure 5 shows the effect of Topography, for three different clusters of station characteristics, as introduced above (Case I: Continental, Case II: Coastal and Case III: Mountainous). In Figure 5a, the Box and Whisker plot on the $CATS_i$-$EARLINET_i$ residuals is shown, including the lower and upper whiskers which indicate the 10th and 90th percentiles respectively, and the 25th and the 75th quantiles indicated by the lower and upper box boundaries respectively. The horizontal line and the red dot indicate the statistical mean and median values respectively while outliers are indicated by red crosses. According to the results, it is evident that the correlative measurements between the Mountainous EARLINET stations and the ISS overpasses are characterized by higher variability, more extreme differences, higher absolute mean and median biases and higher RMSE than in the Continental and Maritime cases. Complex topography, in terms of geographical characteristics, erroneous mean backscatter coefficient profiles due to the high variability of aerosol load in the Planetary Boundary Layer, the horizontal distance between the CATS lidar footprint and the ground-based lidar stations and surface returns enhance the discrepancies, especially in the lowermost part of the profiles, resulting in higher differences between the EARLINET profiles and CATS profiles. Due to the lack of the aforementioned effects arising from complex topography, CATS representativeness and performance is higher over the Continental cases, while CATS performance over the Coastal stations is characterized by slightly lower absolute value of mean bias and at the same time by lower Correlation Coefficient than in the case of Continental cases. However, it has to be taken into consideration the important factor related to the presented resultsm that is the number of CATS-EARLIENT correlative cases used in the analysis, 23 for Case I - Continental, 10 for Case II - Coastal and 14 for Case III - Mountainous. Analytical evaluation metrics on the effect of topography are given in Table 2.

[Figure]

*Figure 5: CATS backscatter coefficient at 1064nm with respect to EARLINET ground-based measurements, as a function of different topography of EARLINET stations for three different clusters of station topographical characteristics (Case I: Continental, Case II: Coastal and Case III: Mountainous).*

*In Fig.5a, the Box and Whisker plot on the CATS$_i$-EARLINET$_i$ residuals is shown, including the lower and upper whiskers which indicate the 10$^{th}$ and 90$^{th}$ percentiles respectively, and the 25$^{th}$ and the 75$^{th}$ quantiles indicated by the lower and upper box boundaries respectively. The horizontal line and the red dot indicate the statistical mean and median values respectively while outliers are indicated by red crosses. Fig.5b and Fig.5c show the RMSE and Correlation Coefficient as a function of the different clusters, including the number of available cases per cluster.*

Table 2: Clusters of EARLINET stations and CATS evaluation metrics.

|  | Continental stations | Coastal stations | Mountainous stations |
|---|---|---|---|
| Median | -0.053 [Mm$^{-1}$sr$^{-1}$] | -0.076 [Mm$^{-1}$sr$^{-1}$] | -0.106 [Mm$^{-1}$sr$^{-1}$] |
| Mean | -0.016 [Mm$^{-1}$sr$^{-1}$] | -0.058 [Mm$^{-1}$sr$^{-1}$] | -0.151[Mm$^{-1}$sr$^{-1}$] |
| RMSE | 0.367 [Mm$^{-1}$sr$^{-1}$] | 0.293 [Mm$^{-1}$sr$^{-1}$] | 0.434 [Mm$^{-1}$sr$^{-1}$] |
| Correlation Coefficient | 0.673 | 0.499 | 0.591 |
| Number of cases | 23 | 10 | 14 |
**General Comments: This manuscript evaluates the Level 2 aerosol backscatter coefficient retrieved by the spaceborne backscatter lidar CATS (Cloud-Aerosol Transport System) using collocated ground-based**
15 **measurements from 14 EARLINET (European Aerosol Research Lidar Network) stations. The manuscript is well written and its contribution to the scientific aerosol community is valuable. I believe that the paper is adequate for publication under the special issue "EARLINET aerosol profiling: contributions to atmospheric and climate research" of the Atmospheric Chemistry and Physics journal after minor revision.**

20 The authors would like to thank the reviewer for the interesting and at the same time substantial comments and suggestions. We tried, and did our best, to incorporate the proposed changes and corrections in the revised manuscript, aiming at improving the presented paper. Following, you will find our responses, one by one to the comments addressed.
Kind regards,
25 Emmanouil Proestakis

**Specific Comments:**
**Page 2, Line 5: "…underestimations of the total Aerosol Optical Depth (AOD)". Please reframe this sentence. The way it is currently written it gives the impression that the AOD exploration is part of this study.**

The authors agree with the reviewer regarding CATS AOD at 1064nm. CATS AOD at 1064 nm has been investigated by a significant number of research groups, towards the assessment of CATS performance (e.g. Rajapakshe et al., 2017; Lee et al., 2018; Noel et al., 2018), with the findings of the aforementioned studies to the manuscript. To be more specific, CATS performance has been validated against ground-based AERONET (Holben et al., 1998)
35 measurements and evaluated against satellite-based AOD retrievals of MODIS (Levy et al., 2013) Aqua and Terra and active CPL (McGill et al., 2002) and CALIPSO CALIOP (Winker et al., 2009) profiles of extinction coefficient and AOD at 1064 nm. Lee et al. (2018) based on AERONET AOD (1020 nm) found a reasonable agreement with CATS AOD with a correlation of 0.64. A comparative analysis of CATS and MODIS C6.1 DT AOD retrievals reported correlation of 0.75 and slope of 0.79, over ocean. In addition, Lee et al., (2019) evaluated AOD and extinction
40 coefficient profiles from CATS through intercomparison with CALIOP and showed correlation of 0.62 and 0.52 over land and ocean respectively during daytime, and 0.84 and 0.81 over land and ocean respectively during nighttime.

Comparison of CATS and CALIOP collocated extinction coefficient profiles shows also good shape agreement. Rajapakshe et al. (2017) reported on similar geographical patterns regarding Above Cloud Aerosols and Cloud Fraction between CATS and CALIOP retrievals. Furthermore, CATS retrievals were used to document the diurnal cycle and variations of clouds. Noel et al. (2018) showed that both CATS and CALIOP profiles of Clouds agree well, with minor differences of the order of 2-7% throughout the entire profiles. In addition, CATS depolarization measurements were investigated in the case of desert dust, smoke from biomass burning and cirrus clouds (Yorks et al., 2016), and were found consistent and in good agreement with depolarization measurements from previous studies and historical datasets implementing CPL (Yorks et al., 2011) and CALIOP (Liu et al., 2015). The studies report in general on the good performance of CATS, despite apparent underestimations. Similarly, to the aforementioned studies, the present study also reports on the good performance of CATS, especially during nighttime, despite underestimations in the backscatter coefficients. The performed studies, not only the present study, report on CATS underestimations, starting from CATS L1 to CATS L3 data, for this reason the sentence includes the phrase: "CATS low negative biases, partially attributed to the deficiency of lidar systems to detect tenuous aerosol layers of backscatter signal below the minimum detection thresholds, *may* lead to systematic deviations and slight underestimations of the total Aerosol Optical Depth (AOD) in climate studies.".

**Page 3, Line 29: "CATS retrievals…..complementarily used". This sentence is incomplete as written. Please reframe.**

According to the reviewer's recommendation, the commented sentences of the manuscript were modified from:
*"CATS retrievals were used to document the diurnal cycle and variations of clouds, with CALIOP complementarily used. Noel et al. (2018) showed that both CATS and CALIOP profiles of CF agree well on both the vertical patterns and values at 01:30 and 13:30 LT, over both land and ocean, with minor differences of the order of 2-7% throughout the entire profiles of cloud fraction"*
to:
*"Noel et al. (2018), implemented measurements from CATS to investigate the diurnal cycle and variations of clouds over land and ocean. The authors showed that both CATS and CALIOP profiles and CF agree well on both the vertical patterns and values at 01:30 and 13:30 LT, over both land and ocean, with minor differences of the order of 2-7% throughout the entire cloud profiles"*.

**Page 4, Line 3: Could the authors specify the reason behind choosing the aerosol backscatter at 1064 nm as the only parameter in the comparison? As explained in Section 2.1, Mode 2 gives the opportunity to include the aerosol backscatter coefficient at 532 nm in the comparison which would have enabled a better evaluation of CATS products spectrally and would have also enabled an error estimation for secondary derived lidar parameters such as the backscatter-related Ångström exponent.**

The reviewer is right regarding the potential value of the Mode 7.2 532nm product. However as mentioned in the CATS "Data Release Notes" and the CATS "Algorithm Theoretical Basis Document" (ATBD), unlike the M7.1 data, where the 532 and 1064 nm signals are comparable, the M7.2 532 and 1064 nm signals are very different. Mode 7.2 data at 532 nm is noisy due to issues with stabilizing the seeded laser (laser 2). Since the frequency stability is poor on laser 2, the laser is not aligned properly with the CATS etalon, causing very weak signal transmission. Therefore, it is highly recommended by the NASA CATS team not to use the M7.2 532 nm data for any application, especially for daytime. On the contrary, the use of the 1064 nm data is recommended, though only for studies that are wavelength-independent (i.e. layer detection, relative backscatter intensity).
 (https://cats.gsfc.nasa.gov/media/docs/CATS_Release_Notes7.pdf)

**Page 4, Line 30: "CATS was a….to three years". Difficult to read sentence. Please rephrase.**

According to the reviewer's recommendation, the commented sentences of the manuscript were modified from:
 *"CATS was a technology demonstration designed to operate on-orbit for a minimum of six months and up to three years"*

to:

*"CATS was a technology demonstration designed to operate on-orbit between six months and three years"*.

**Page 4, Line 35: "CATS products and….of processing". Please rephrase the sentence. The part "…and provided in …" is not in the correct tense or it is not a continuation of the previous text.**

According to the reviewer's recommendation, the commented sentences of the manuscript were modified from:
 *"CATS products and processing algorithms (Pauly et al., 2019) rely heavily on the processing algorithms developed in the framework of the CPL, ACATS and CALIPSO lidar systems (Palm et al., 2002; Yorks et al., 2011; Hlavka et al., 2012) and provided in different levels of processing."*
to:
*"CATS processing algorithms (Pauly et al., 2019) rely heavily on the processing algorithms developed in the framework of the CPL, ACATS and CALIPSO lidar systems (Palm et al., 2002; Yorks et al., 2011; Hlavka et al., 2012), while CATS products are provided in different levels of processing"*.

**Page 5, Line 1: What is the error in CATS aerosol backscatter retrievals?**

The primary sources of uncertainties in the CATS attenuated backscatter signal are the calibration constant and signal noise. Thus if the calibration constant is accurate, the CATS The source of the systematic error in the CATS ATB is the uncertainty in the calibration constant and is estimated at 5-10% for 1064 nm at night (10-20% for daytime data). The random error in the ATB is dominated by noise in the lidar signal. The total uncertainty, sum of the systematic and random errors, in the CATS ATB 1064 nm is estimated at 10-20% for nighttime data and 20-30% for daytime data.
According to the reviewer's observation and comment, the manuscript was expanded to include also the phrase:
*"The total uncertainty, the sum of the systematic and random errors, in the CATS ATB at 1064nm is estimated at 10-20% for nighttime data and 20-30% for daytime data."*

**Page 7, Line 4: Hohenpeissenberg site is not listed here. Please add it.**

The authors would like the reviewer for observing that Hohenpeissenberg EARLINET station was not included in the list. The list is updated to include Hohenpeissenberg station.

**Page 7, Line 22: "widow" -> window**

The text is corrected according to the reviewer's comment.

**Page 8, Line 11-13: The authors used two different processing algorithms for the retrieval of the ground-based aerosol backscatters namely the SCC and PollyXT specified retrieval algorithms. Under the SCC, all measurements could have been processed/treated in the same way. Could you comment on this decision not to process all measurements in the same way and whether these two algorithms can introduce discrepancies in the reported CATS comparison?**

Indeed, in case of Polly[XT] systems, an algorithm designed specifically for these systems is used for the retrieval of aerosol backscatter coefficient profiles. The reason why we decided not to process these data with the Single Calculus Chain (SCC) is that up to now, the SCC is able to support very efficiently the processing of lidar signals for systems employing two different acquisition receivers (i.e. one in analogue and one in photon counting mode or both in photon counting mode) for the acquisition of far range and near range signals, but only one receiving telescope. However, it does not yet support the simultaneous processing of lidar signals in case of systems employing two different telescopes for the acquisition of far range and near range signals (i.e. Polly[XT] systems). Therefore, for Polly[XT] systems in the SCC, the only way in order to correct for the incomplete overlap effect at lower altitudes would be to use an overlap function profile. On the contrary, the Polly[XT] specified retrieval algorithm

enables the processing of the two signals (collected from far and near range telescopes) simultaneously, for this reason we decided not ignore the useful information collected by these two different receiving units and use this algorithm instead. The aforementioned algorithm has been excessively studied in the past (Baars et al., 2016) and proven to provide accurate results in an efficient manner. We thus believe that in this way the comparison of Polly$^{XT}$
5   profiles with profiles from other lidar systems, processed with SCC is more accurate since for the former an assumption of the overlap profile is not needed.

**Page 9, Line 28: CATS has an overpass over Athens-NTUA at the same day even closer to the measurement site than Athens-NOA but at different time frame (a bit later). As the authors explain, the atmospheric conditions were rather stable at that day. To authors' discretion, I find it valuable/informative if the profiles from that**
10  **station would be added to Figure 2d and discuss further on the possible differences or similarities.**

According to the reviewer's observation and comment, Figure 2d was adapted in order to include not only the CATS aerosol profile (denoted by blue line) and the observations of the EARLINET-NOA (denoted by red line) station but in addition the aerosol profile of the EARLINET-NTUA station (denoted by black line). The different of the EARLINET-
15  NOA and EARLINET-NTUA profiles are related to the significant different elevation of the two stations (212m and 95m respectively), to the differences of the two systems, including the implementation of a near field at the Polly$^{XT}$ lidar system of the National Observatory of Athens, in contrast to the NTUA system.

[revised manuscript text omitted]

**Page 10, Line 30. I suggest to put each of the cases into a different section giving a short title indicating the complexity of the example.**

According to reviewer's recommendation, the suggested part of the manuscript is modified to include the following headers, indicating the different EARLINET-CATS correlative cases:

3.1.1 Case I: ISS-CATS over Leipzig - 13/09/2016 03:37 UTC

3.1.2 Case II: ISS-CATS over Évora - 31/05/2016 19:43 UTC

3.1.3 Case III: ISS-CATS over Dushanbe - 25/05/2015 18:53 UTC

**Page 12, Line 1: I suggest to report the numbers with the same accuracy, for example the numbers at the end of this sentence are inconsistent. Similarly to Table 4.**

The authors agree with the reviewer, therefore the sentence was modified to include the Minimum Detectable Backscatter (MDB) with the same accuracy, as shown in the paper of Yorks et al. (2016) regarding CATS level 1 processing algorithms and data products, as follows:

*"CATS M7.2 Minimum Detectable Backscatter 1064 nm: Night: 5.00E-5 ± 77E-5 $km^{-1}sr^{-1}$ / Day: 1.30E-3 ± 0.24E-3 $km^{-1}sr^{-1}$ - for cirrus clouds; Yorks et al., 2016".*

In addition, and according to the reviewer's comment, Table 4 is corrected, to show numbers with the same accuracy, as follows:

| Metric | Daytime | Nighttime |
|---|---|---|
| Mean Bias [$Mm^{-1}sr^{-1}$] | -0.123 | -0.031 |
| Median Differences | -0.094 | -0.065 |

| | [Mm$^{-1}$sr$^{-1}$] | | |
|---|---|---|---|
| Mean Absolute Bias [Mm$^{-1}$sr$^{-1}$] | 0.323 | 0.249 |
| Mean Relative Bias [%] | -24.062 | -19.843 |
| SD [Mm$^{-1}$sr$^{-1}$] | 0.431 | 0.342 |
| (min / max Differences) [Mm$^{-1}$sr$^{-1}$] | (-1.802 / 1.189) | (-1.348 / 1.149) |
| RMSE [Mm$^{-1}$sr$^{-1}$] | 0.448 | 0.343 |
| Correlation Coefficient | 0.547 | 0.694 |
| Fractional Bias | -0.773 | -0.676 |
| Fractional Gross Error | 0.999 | 1.061 |
| Number of Cases (#) | 21 | 26 |

**Page 14, Line 10. To my understanding, the authors mention at Page 10, lines 19-21 that cases where CATS backscatter coefficient is zero or it is at its minimum detection limit have been eliminated from the study yet they are present in this figure for altitudes higher than 6km. Could you clarify?**

In the framework of the comparison methodology, cases of EARLINET backscatter coefficient values below the CATS minimum detectable backscatter limit at 1064 nm are not included in the comparison, when the corresponding CATS backscatter coefficient is reported to be zero (Fig. 2d - shaded area i). This constrain is applied to account for very thin detected layers from ground-based Lidar systems with backscatter values below the CATS minimum detection limit due to the low Signal-to-Noise Ratio values (SNR). The discussed constrains are employed because of our basic idea to quantitatively assess the representativeness and accuracy of the detected by CATS aerosol features, while preventing possible contaminations (e.g. presence of clouds) to propagate into the CATS-EARLINET dataset. It is applied in the comparison shown for instance in Figure 6, to address quantitatively the accuracy and representativeness of the satellite-based lidar retrievals and to estimate possible biases in the CATS backscatter coefficient. It is applied in the comparison of CATS against EARLINET, to the implementation of the CATSi-EARLINETi residuals for each pair of observations "i", to be used as statistical indicators of CATS average overestimation or underestimation of the aerosol load, in terms of backscatter coefficient values. In this term, since the analysis if focusing to the possible CATS overestimation/underestimation of the aerosol load, the authors compare cases where aerosols are detected by both EARLINET and CATS, or by at least one of the two systems. The comparison statistics on the efficiency of CATS to detect atmospheric features detected by EARLINET systems refers to the aforementioned discussion. However, the study of the evaluation discussion of the mean aerosol backscatter coefficient profiles at 1064 nm as provided by CATS and EARLINET, without further processing, for both daytime

(Fig. 7a) and nighttime (Fig. 7b) lidar observations, to investigate the characteristics, similarities and discrepancies between CATS and EARLINET.

**Page 16, Line 4: The authors have used the Level 2 v2.01 for the evaluation. Nonetheless the latest available version is the v3.01. How this versioning is going to change the associations reported here? Could you correct the versioning to the latest available as in here and line 1 at page 17, for consistency?**

The authors would like to thank the reviewer for this comment, the versioning at page 17 was corrected as should be.

Regarding CATS algorithms and the different versions, CATS V3-00 replaced CATS V2-05 on October 1$^{st}$, 2018. Accordingly, due to algorithm issues present in CATS V3-00 was shortly after replaced by CATS V3-01. Initially, the changes in CATS Level 1 and Level 2 algorithms corresponding to CATS Version 3-00 data was planned to be the final algorithm release for the CATS project, though observed issues in the CATS products led to the modifications of V3-00 and the release of the V3-01 later on in the beginning of 2019. Since CATS products are provided in different levels of processing, the made changes in the algorithms correspond to both L1B and L2O products.

To be more specific, the changes in the L1B algorithms include:

(1) improvement of the nighttime attenuated total backscatter (ATB) profiles due to improvements in the calibration of CATS, thus improvement also in the daytime ATB profiles, since nighttime ATB is implemented in the calculations of the daytime calibration.

(2) changes to the "Depolarization_Quality_Flag", and

(3) implementation of MERRA-2 Reanalysis data instead of GMAO forecasts, for the meteorology in V3-00 and V3-01.

The changes made in the algorithms of CATS L1B reflect on improvements on CATS L2O products. Though additional changes in CATS L2O algorithms include also:

(1) updates in number of profiles in the L2O datasets

(2) improvements in the calculations of uncertainties in the L2O layer-integrated parameters

(3) changes to the "Depolarization_Quality_Flag"

(4) improvements of the Cloud Aerosol Discrimination (CAD) through the implementation of an additional parameter, namely the "Cloud_350m_Fraction_XXX_FOV", to report of the number of 350 L1B profiles within each 5 km L2O bin of the L2O layer product with attenuated total backscatter values greater than 0.03 km$^{-1}$sr$^{-1}$, thus atmospheric features of high probability of being a cloud. In addition, the parameter "Num_Profs_Avg_LRatio_XXX_FOV" was added to the L2O Layer data product.

(5) improvements in CATS Feature Type and Feature Type Score variables, but also in the Aerosol Subtype classification (replace of "volcanic" with "UTLS Aerosol") and addition of the parameters "Opaque_Feature_Optical_Depth_1064_XXX_FOV" and "Opaque_Feature_Optical_Depth_Uncertainty_1064_XXX_FOV" in Mode 7.2 L2O datasets.

(6) Updates in the Lidar Ratio (LR) values for cirrus clouds

(7) update of the effective multiple scattering factor for ice clouds values to 0.52.

The above changes in the CATS V3-00 and V3-01 algorithms and the respective products are extensively presented and in-depth discussed in the CATS official website (https://cats.gsfc.nasa.gov/; last visit on: 22/05/2019), in the "Publications" section.

CATS products and processing algorithms are provided in different levels of processing. CATS Level 1B (L1B) data include vertical profiles of total and perpendicular attenuated backscatter signals, range-corrected, calibrated and annotated with ancillary meteorological parameters (McGill et al., 2007; Powell et al., 2009; Vaughan et al., 2010). CATS Level 2 (L2) products provide the vertical distribution of aerosol and cloud properties (depolarization ratio, backscatter and extinction coefficient profiles at 1064 nm – FFOV), with a horizontal and vertical resolution of 5 km and 60 m respectively. In addition, L2 data include geophysical parameters of the identified atmospheric layers (vertical feature mask - feature type, aerosol subtype), the required horizontal averaging and information on the feature type classification confidence (Yorks et al., 2019).

Regarding the way this versioning is change the associations reported in the present manuscript, more information are included in the CATS validation led by NASA GSFC Team, and more specific by Dr. Rebecca Pauly (Science

Systems and Applications Inc., Lanham, 20706, United States Science Systems and Applications Inc., Lanham, 20706, United States), member of the CATS Team. The study is already submitted on AMT journal:

*"Pauly, R. M., Yorks, J. E., Hlavka, D. L., McGill, M. J., Amiridis, V., Palm, S. P., Rodier, S. D., Vaughan, M. A., Selmer, P. A., Kupchock, A. W., Baars, H., and Gialitaki, A.: Cloud Aerosol Transport System (CATS) 1064 nm Calibration and Validation, Atmos. Meas. Tech. Discuss., https://doi.org/10.5194/amt-2019-172, in review, 2019"*.

This study, similarly to the present study, implements specific lidar systems of EARLINET. However, the study of CATS performance will continue and not end with the present manuscript. In this study, the EARLINET authors in collaboration with the CATS Team evaluate CATS Level 2 Mode 7.2 v2-05 backscatter profiles at 1064nm (Palm et al., 2016). The authors have focused on particulate backscatter coefficients (km$^{-1}$sr$^{-1}$), since this is the product directly derived from measurements, the sum of the parallel and perpendicular backscatter measurements (i.e., $\beta_{1064nm\ total} = \beta_{1064nm\ parallel} + \beta_{1064\ perpendicular}$).

An already ongoing study includes high collocated analysis on the CATS performance (V3-01) and representativeness, including the issues mentioned by the reviewer. The study is based on high temporally and spatially collocated measurements between airborne FAAM Bae-146 research aircraft and ISS measurements, performed in the framework of the AER-D/ICE-D campaign, over Cape-Verde (Santiago island), on August 6-25, 2015, as introduced by Marenco et al. (2018) – Figure 5.

[Figure]

*Figure 5: B920 flight on August 7$^{th}$, 2015 over Cape Verde, high collocated with ISS-CATS overpass. Left: B920 flight and ISS footprint (left), and CATS backscatter coefficient 1064nm scene (right).*

**Page 17, Line, 11: "52o" -> "52°"**

The text is corrected according to the reviewer's comment.

**Page 17, Line 20: "explotations" -> "explotation"**

The text is modified according to the reviewer's comment.

**Page 34, Line 2: Please include a complete explanation of the figure, e.g the time frame of the ground-based lidar retrieval and the overpass of CATS along with explanation on the error bar in Figure c. The same applies to the rest of the cases (Figures 4 and 5).**

Figure 3
From:

*"Figure 3: (a) Nighttime ISS orbit over EARLINET Leipzig station on the 13th of September 2016 (blue line). The white dot denotes the location of Leipzig lidar system, (b) CATS Backscatter Coefficient at 1064 nm and (c) CATS (blue line) and EARLINET-Leipzig (red line) backscatter coefficient profiles (1064 nm)".*

to:

*Figure 3: (a) Nighttime ISS orbit over the EARLINET Leipzig station on the 13th of September 2016 at 03:37:49 UTC and of closest distance between the footprint of CATS and the EARLINET- Leipzig station of 3.79km. The white dot denotes the location of Leipzig lidar system while the blue line shows the lidar footprint of CATS. (b) CATS Backscatter Coefficient at 1064 nm. (c) CATS (blue line) spatially and EARLINET-Leipzig (red line) temporarily averaged backscatter coefficient profiles (1064 nm). The implemented EARLINET-Leipzig time window of cloud-free measurements was between 00:00:00 and 02:30:00 UTC. The horizontal blue and red lines denote the variability (one standard deviation) of the CATS and EARLINET measured atmospheric scenes, respectively.".*

Figure 4
From:
*"Figure 4: (a) Daytime ISS orbit over Évora EARLINET station on the 31th of May 2016 (red line). The white dot denotes the location of Évora lidar system, (b) CATS Backscatter Coefficient at 1064 nm and (c) CATS (blue line) and EARLINET-Évora (red line) backscatter coefficient profiles (1064 nm)".*

to:

*"Figure 4: (a) Daytime ISS orbit over the EARLINET Évora station on the 31th of May 2016 at 19:43:31 UTC and of closest distance between the footprint of CATS and the EARLINET- Évora station of 39.42km. The white dot denotes the location of Évora lidar system while the red line shows the lidar footprint of CATS. (b) CATS Backscatter Coefficient at 1064 nm. (c) CATS (blue line) spatially and EARLINET-Évora (red line) temporarily averaged backscatter coefficient profiles (1064 nm). The implemented EARLINET-Évora time window of cloud-free measurements was between 19:29:56 and 19:59:35 UTC. The horizontal blue and red lines denote the variability (one standard deviation) of the CATS and EARLINET measured atmospheric scenes, respectively.".*

Figure 5
From:
*"Figure 5: (a) Nighttime ISS orbit over Dushanbe EARLINET station on the 25th of May 2015 (blue line). The white dot denotes the location of Dushanbe lidar system, (b) CATS Backscatter Coefficient at 1064 nm and (c) CATS (blue line) and EARLINET-Dushanbe (red line) backscatter coefficient profiles (1064 nm)."*

to:

*"Figure 5: (a) Nighttime ISS orbit over the EARLINET Dushanbe station on the 25th of May 2015 at 18:53:19 UTC and of closest distance between the footprint of CATS and the EARLINET- Dushanbe station of 24.3km. The white dot denotes the location of Dushanbe lidar system while the blue line shows the lidar footprint of CATS. (b) CATS Backscatter Coefficient at 1064 nm. (c) CATS (blue line) spatially and EARLINET-Dushanbe (red line) temporarily averaged backscatter coefficient profiles (1064 nm). The implemented EARLINET-Dushanbe time window of cloud-free measurements was between 18:00:00 and 20:00:00 UTC. The horizontal blue and red lines denote the variability (one standard deviation) of the CATS and EARLINET measured atmospheric scenes, respectively.".*

**Page 34, Line 12: In Figure 5a the overpass, although nighttime, it is colored as red. To my understanding nighttime overpasses are blue colored. Please ignore this comment if not relevant.**

The reviewer is right regarding the Dushanbe station, regarding the time of ISS overpass and the frequently type of color used, at least in CALIPSO CALIOP quicklooks. Although in the case of ISS-CATS orbits are not presented in the same colors, blue and red for nighttime and daytime orbits respectively, we have used the same color here, thus Figure 5a was adapted accordingly, as follows:

[Figure]

*Figure 5: (a) Nighttime ISS orbit over the EARLINET Dushanbe station on the 25th of May 2015 at 18:53:19 UTC and of closest distance between the footprint of CATS and the EARLINET- Dushanbe station of 24.3km. The white dot denotes the location of Dushanbe lidar system while the blue line shows the lidar footprint of CATS. (b) CATS Backscatter Coefficient at 1064 nm. (c) CATS (blue line) spatially and EARLINET-Dushanbe (red line) temporarily averaged backscatter coefficient profiles (1064 nm). The implemented EARLINET-Dushanbe time window of cloud-free measurements was between 18:00:00 and 20:00:00 UTC. The horizontal blue and red lines denote the variability (one standard deviation) of the CATS and EARLINET measured atmospheric scenes, respectively.*